# On the Edge of Memorization in Diffusion Models

**Sam Buchanan**[*]        **Druv Pai**[*]        **Yi Ma**        **Valentin De Bortoli**
TTIC        UC Berkeley        UC Berkeley, HKU        Google DeepMind

## Abstract

When do diffusion models reproduce their training data, and when are they able to generate samples beyond it? A practically relevant theoretical understanding of this interplay between memorization and generalization may significantly impact real-world deployments of diffusion models with respect to issues such as copyright infringement and data privacy. In this work, to disentangle the different factors that influence memorization and generalization in practical diffusion models, we introduce a scientific and mathematical "laboratory" for investigating these phenomena in diffusion models trained on fully synthetic or natural image-like structured data. Within this setting, we hypothesize that the memorization or generalization behavior of an underparameterized trained model is determined by the *difference in training loss* between an associated memorizing model and a generalizing model. To probe this hypothesis, we theoretically characterize a *crossover point* wherein the weighted training loss of a fully generalizing model becomes greater than that of an underparameterized memorizing model at a critical value of model (under)parameterization. We then demonstrate via carefully-designed experiments that the location of this crossover predicts a phase transition in diffusion models trained via gradient descent, validating our hypothesis. Ultimately, our theory enables us to analytically predict the model size at which memorization becomes predominant. Our work provides an analytically tractable and practically meaningful setting for future theoretical and empirical investigations. Code for our experiments is available at `https://github.com/DruvPai/diffusion_mem_gen`.

## 1 Introduction

Diffusion models are one of the premier methodologies for deep generative modeling. They exhibit great capabilities across modalities and are state-of-the-art at synthesizing images [Saharia et al., 2022, Podell et al., 2023], videos [Ho et al., 2022, Blattmann et al., 2023], or proteins [Watson et al., 2023]. Despite significant success when used in practice, in the context of large-scale commercial deployment [Ramesh et al., 2022] diffusion models are often plagued with data privacy and copyright infringement issues [Ghalebikesabi et al., 2023, Carlini et al., 2023, Nasr et al., 2023, Cui et al., 2023, Wang et al., 2024a, Vyas et al., 2023, Franceschelli and Musolesi, 2022], which may have significant long-term consequences. These issues stem from the possibility and (sometimes) propensity for diffusion models to *memorize* their training data. Namely, in some cases, trained diffusion models generate outputs which are verbatim copies of the training samples [Zhang et al., 2023, Kadkhodaie et al., 2023]. Yet, in many other cases, these models can generate outputs which appear natural but are not present in the training set; this behavior is often described as *creativity* [Kamb and Ganguli, 2024] or *generalization* [Zhang et al., 2023, Kadkhodaie et al., 2023, Niedoba et al., 2024].

In general, memorization and generalization are difficult to disambiguate. There has been significant *empirical* work focusing on building heuristic approaches to detect and study memorization in large-scale diffusion models [Zhang et al., 2023, Gu et al., 2023, Yoon et al., 2023, Carlini et al., 2023,

---

[*]These authors contributed equally to this work.

39th Conference on Neural Information Processing Systems (NeurIPS 2025).

Somepalli et al., 2023, Wen et al., 2024, Ross et al., 2024, Wang et al., 2024b,a, Chen et al., 2024a]. Most such approaches use ad-hoc definitions of memorization which compare the features of the generated samples (w.r.t. some deep neural network encoder) to features of samples from the training data [Pizzi et al., 2022]. This choice effectively reduces the question of *how exactly can we define memorization?* to *how are the features of the chosen encoder related to the input data?* which is also a difficult problem [Papyan et al., 2020, Yu et al., 2023]. While such methods appear to perform reasonably well in some practical cases, we emphasize that such measures of memorization are ultimately heuristic and there is still scientific disagreement about their efficacy [Stein et al., 2024]. This is unsuitable for building a scientific understanding of diffusion models that could, among other things, potentially suggest resolutions to the aforementioned legal and societal issues.

Thus, building a theoretical and scientific understanding of memorization in diffusion models is critical. The central problem that such a theory needs to contend with is that the training loss promotes memorization: given a fixed sample set, a sufficiently powerful and perfectly-optimized diffusion model will *always* reproduce the training data exactly, i.e., every single one of its generations will be exactly a training point [Peluchetti, 2023, Biroli et al., 2024, Kamb and Ganguli, 2024]. Any theory of memorization must therefore explain why well-trained diffusion models do not always memorize. Several previous works have proposed different explanations for this issue in terms of certain aspects of the training procedure, such as the landscape of the stochastic optimization problem which minimizes the training loss [Wu et al., 2025, Vastola, 2025] and the parameterization of the backbone denoiser in the diffusion model [Yoon et al., 2023, Zhang et al., 2023, Wang et al., 2024c, Kamb and Ganguli, 2024, Niedoba et al., 2024, George et al., 2025]. However, a precise and predictive theoretical characterization of memorization remains elusive.

**Our contributions.** In this work, we theoretically investigate memorization and generalization in diffusion models. We first introduce a *memorization laboratory*, a natural setting for investigating memorization and generalization in diffusion models trained on synthetic data. We justify this setting by proving that within it, we may distinguish a target distribution from its empirical version for many configurations of problem parameters. To explore the behavior of trained models, we hypothesize that *denoisers trained via gradient descent-like methods memorize or generalize depending on whether the empirical (training) loss is lower for parameter-matched memorizing denoisers or generalizing denoisers.* This hypothesis, to the best of our knowledge, has not been formally explored by prior work, and can only be probed here due to the controlled laboratory setting. To formalize and eventually attempt to falsify this hypothesis, we introduce a *partially memorizing denoiser*, an underparameterized denoiser whose output distribution's samples are always memorized. Within a simple setting of our laboratory, we characterize the critical level of model (under)parameterization ("crossover point") at which the training loss of the partially memorizing denoiser first becomes lower than the idealized generalizing denoiser, by deriving and using tight theoretical approximations to these losses. We show that, if our hypothesis is true, then this crossover characterization naturally provides the location of the *phase transition* from generalization to memorization in trained denoisers. Ultimately, we use these theoretically-derived tools to build a *predictive model for the phase transition* from generalization to memorization in terms of a minimal set of problem parameters, and show via experiments that our model achieves extremely low error in practice, validating our hypothesis. We finish our experiments by examining another setting of our laboratory which captures more of the complexities involved in training diffusion models on natural images, showing that despite its additional complexity it is qualitatively very similar to the first case. Beyond the current investigation, the framework we propose provides an analytically tractable yet rich setting for further investigation of memorization and generalization in diffusion models.

All proofs are included in the supplementary material. Our notation is presented in Tables 1 and 2.

## 2 A Memorization/Generalization Laboratory

Diffusion models show a complex tradeoff between model capacity, training compute, and dataset size with respect to key behaviors such as memorization and generalization. We present a framework ("laboratory") to disentangle these different factors. The class of models we study is sufficiently expressive to admit a rich family of behaviors while remaining tractable for theoretical analysis.

**Diffusion models.** Given a target probability distribution $\pi_\star$ and a training dataset $(\boldsymbol{x}^i)_{i=1}^N$ from $\pi_\star$, the goal of generative modeling is to define an *output probability distribution* $\hat{\pi}$ which approximates the underlying target $\pi_\star$. Diffusion models [Sohl-Dickstein et al., 2015, Song and Ermon, 2019, Ho et al., 2020] define such an output distribution $\hat{\pi}$ as the result of a stochastic process. More precisely, we consider, for $t \in [0, 1]$, a noising process with marginals as follows:

$$X_t \stackrel{d}{=} \alpha_t X_0 + \sigma_t Z, \quad Z \sim \mathcal{N}(\boldsymbol{0}, \boldsymbol{I}), \quad X_0 \sim \pi_\star, \tag{1}$$

with $\alpha_0 = \sigma_1 = 1$ and $\alpha_1 = \sigma_0 = 0$. This noising process can be associated with the dynamic

$$\mathrm{d}X_t = f_t X_t \mathrm{d}t + g_t \mathrm{d}B_t, \quad X_0 \sim \pi_\star. \tag{2}$$

where $f_t$ and $g_t$ can be obtained in closed form, see Gao et al. [2024] for instance, and $(B_t)_{t \in [0,1]}$ is a $d$-dimensional Brownian motion. We recall that the backward process associated with (2) is given by

$$\mathrm{d}Y_t = \{-f_{1-t}Y_t + (g_{1-t}^2/2)\nabla \log p_{1-t}(Y_t)\}\mathrm{d}t + g_{1-t}\mathrm{d}B_t, \quad Y_0 \sim \mathcal{N}(\boldsymbol{0}, \boldsymbol{I}), \tag{3}$$

where $p_t$ is the density of $X_t$ at time $t$. Using Tweedie's identity [Robbins, 1956], we have

$$\nabla \log p_t(\boldsymbol{x}_t) = (\alpha_t \bar{\boldsymbol{x}}(t, \boldsymbol{x}_t) - \boldsymbol{x}_t)/\sigma_t^2, \tag{4}$$

where $\bar{\boldsymbol{x}}(t, \boldsymbol{x}_t) = \mathbb{E}[X_0 \mid X_t = \boldsymbol{x}_t]$. Thus, using the samples $(\boldsymbol{x}^i)_{i=1}^N$, practical diffusion models define a denoiser that approximates $\bar{\boldsymbol{x}}$ by solving the training loss minimization problem

$$\min_\theta \mathcal{L}_N(\bar{\boldsymbol{x}}_\theta, \lambda) \tag{5}$$

where

$$\mathcal{L}_N(\bar{\boldsymbol{x}}, \lambda) = \mathbb{E}_t[\lambda(t)\mathcal{L}_{N,t}(\bar{\boldsymbol{x}})], \qquad \mathcal{L}_{N,t}(\bar{\boldsymbol{x}}) = \frac{1}{N}\sum_{i=1}^N \mathbb{E}_{X_t^i}\left[\left\|\bar{\boldsymbol{x}}(t, X_t^i) - \boldsymbol{x}^i\right\|^2\right] \tag{6}$$

over a parametric class of denoisers $\bar{\boldsymbol{x}}_\theta$, with $\lambda(t)$ a weighting function, $t$ distributed as $\mathrm{Unif}([0, 1])$, and $X_t^i$ distributed according to the noising process (1) with $X_0 = \boldsymbol{x}^i$, i.e., $X_t^i \stackrel{d}{=} \alpha_t \boldsymbol{x}^i + \sigma_t Z$. We then substitute the learned denoiser $\bar{\boldsymbol{x}}_\theta(t, \boldsymbol{x}_t)$ into a suitable discretization of the backward process (3), via (4), which defines the *output distribution* $\hat{\pi}$ as the marginal distribution of $Y_1$.

As has been well noted in the literature, the training objective (5) is at odds with the stated goal of diffusion models. This is because the non-parametric minimizer of (5) memorizes the training data:

$$\arg\min_{\bar{\boldsymbol{x}}} \mathcal{L}_N(\bar{\boldsymbol{x}}, \lambda) = \sum_{i=1}^N \boldsymbol{x}^i \, \mathrm{softmax}(\boldsymbol{w}(\,\cdot\,))_i, \tag{7}$$

where $\bar{\boldsymbol{x}}(t, \cdot)$ is square-integrable, for any $\boldsymbol{v} \in \mathbb{R}^N$ we have $\mathrm{softmax}(\boldsymbol{v})_i = \mathrm{e}^{v_i}/\sum_{j=1}^N \mathrm{e}^{v_j}$, and

$$w_i(\boldsymbol{x}_t) = -\frac{1}{2\sigma_t^2}\|\alpha_t \boldsymbol{x}^i - \boldsymbol{x}_t\|^2, \quad i = 1, \ldots, N.$$

We denote the memorizing denoiser in (7) as $\bar{\boldsymbol{x}}_{\mathrm{mem}}(t, \boldsymbol{x}_t)$. Towards theoretically studying the memorization/generalization trade-off in diffusion models, we first specify our main assumptions regarding the data and our models which define our memorization/generalization laboratory.

**Data and model assumptions.** We set $\pi_\star$ to be an equally-weighted mixture of $K$ Gaussians in $\mathbb{R}^d$ with means $\boldsymbol{\mu}_\star^k \in \mathbb{R}^d$ and covariances $\boldsymbol{\Sigma}_\star^k \succeq \boldsymbol{0}$:

$$\pi_\star = \frac{1}{K}\sum_{k=1}^K \mathcal{N}(\boldsymbol{\mu}_\star^k, \boldsymbol{\Sigma}_\star^k \boldsymbol{I}). \tag{8}$$

Gaussian mixture models are flexible enough to represent a large class of datasets while being amenable to theoretical investigation, see [Wang and Vastola, 2024, Shah et al., 2023, Wang et al., 2024c] for instance, Section 4.2 for an example, and Appendix A for a further discussion.

Let $M \in \mathbb{N}$ denote the number of equally-weighted mixture components in a generic Gaussian mixture $\pi_\theta$, with $\theta = (\boldsymbol{\mu}^1, \boldsymbol{\Sigma}^1, \ldots, \boldsymbol{\mu}^M, \boldsymbol{\Sigma}^M)$. Tweedie's identity (4) links the statistical model $\pi_\theta$ and its associated denoiser $\bar{\boldsymbol{x}}_\theta$, which we recall below in Lemma 2.1 and prove in Appendix A.

**Lemma 2.1 (Gaussian mixture model denoiser):** *Assume that $\pi_\theta = (1/M) \sum_{i=1}^{M} \mathcal{N}(\boldsymbol{\mu}^i, \boldsymbol{\Sigma}^i)$. Then, we have that*

$$\bar{\boldsymbol{x}}_\theta(t, \boldsymbol{x}_t) = \frac{1}{\alpha_t} \left( \boldsymbol{x}_t - \sigma_t^2 \sum_{i=1}^{M} (\alpha_t^2 \boldsymbol{\Sigma}^i + \sigma_t^2 \boldsymbol{I})^{-1}(\boldsymbol{x}_t - \alpha_t \boldsymbol{\mu}^i) \operatorname{softmax}(\boldsymbol{w}(t, \boldsymbol{x}_t))_i \right), \quad (9)$$

*where for all $i \in [M]$*

$$w_i(t, \boldsymbol{x}_t) = -\frac{1}{2} \log \det(\alpha_t^2 \boldsymbol{\Sigma}^i + \sigma_t^2 \boldsymbol{I}) - \frac{1}{2}(\boldsymbol{x}_t - \alpha_t \boldsymbol{\mu}^i)^\top (\alpha_t^2 \boldsymbol{\Sigma}^i + \sigma_t^2 \boldsymbol{I})^{-1}(\boldsymbol{x}_t - \alpha_t \boldsymbol{\mu}^i).$$

The class of $M$-parameter Gaussian mixture models constitutes a rich framework for investigating memorization and generalization. For at one extreme, where $M = K$, $\boldsymbol{\Sigma}^k = \boldsymbol{\Sigma}_\star^k$ and $\boldsymbol{\mu}^k = \boldsymbol{\mu}_\star^k$ for $k \in [K]$ we recover the *generalizing denoiser* associated to the target distribution (8), which we will denote as $\bar{\boldsymbol{x}}_\star$; and at the other extreme, with $M = N$, $\boldsymbol{\Sigma}^i = \boldsymbol{0}$, and $\boldsymbol{\mu}^i = \boldsymbol{x}^i$ for $i \in [N]$ we recover the *memorizing denoiser* $\bar{\boldsymbol{x}}_{\mathrm{mem}}$ as in (7). We will therefore consider the training loss (5) with the model class specified by Lemma 2.1, and study the role played by the model capacity $M$ as a function of the data complexity $K$, the dimension $d$, and the number of training samples $N$.

**Memorization and generalization.** We now aim to quantify our notions of memorization and generalization. Quantifying memorization in generative models is a complicated issue and many metrics have been proposed to evaluate it in practice. We adopt a popular, relatively strict metric for memorization, proposed by Yoon et al. [2023].

**Definition 2.2 (Memorization):** *Given a dataset $(\boldsymbol{x}^i)_{i=1}^N$, a small absolute constant $c \in (0,1)$, and the output distribution $\hat{\pi}$ of a diffusion model, we say that a sample $\hat{\boldsymbol{x}} \sim \hat{\pi}$ is* memorized *if $\|\hat{\boldsymbol{x}} - \boldsymbol{x}^{(1)}\|^2 \leq c\|\hat{\boldsymbol{x}} - \boldsymbol{x}^{(2)}\|^2$, where $\boldsymbol{x}^{(k)}$ is the $k$-th nearest neighbor in $\ell_2$ norm to $\hat{\boldsymbol{x}}$ in $(\boldsymbol{x}^i)_{i=1}^N$.*

On the other hand, generalization is easily formalized in statistical learning terms (in contrast to, e.g., assessing creativity of practical image diffusion models), see Appendix A. In experiments, we estimate the generalization error on a held-out set of samples from $\pi_\star$, which we can freely generate.

Given these quantitative definitions, the key question that we investigate in our laboratory is:

> *For a trained denoiser $\bar{\boldsymbol{x}}_\theta$, can we quantitatively predict, based on problem parameters $M$, $N$, $d$, and $K$, whether it memorizes or generalizes?*

By the phrases "the denoiser memorizes/generalizes" we mean that its associated output distribution $\hat{\pi}$ produces memorized samples or samples (approximately) from the true distribution, as defined above. As we will immediately see, it is also fruitful to understand these behaviors as $\hat{\pi}$ being "close" to the empirical distribution $\pi_\star^N$ of the training set $(\boldsymbol{x}^i)_{i=1}^N$ or the target distribution $\pi_\star$ respectively.

**Scaling the number of samples.** Under the Gaussian mixture model $\pi_\star$ that generates the training data $(\boldsymbol{x}^i)_{i=1}^N$, the parameters $K$ and $d$ (together with the means $\boldsymbol{\mu}_\star^k$ and covariances $\boldsymbol{\Sigma}_\star^k$) control the geometric complexity of the data. Relative to these measures of complexity, the scaling of the number of samples $N$ plays a fundamental role in the dichotomy between memorization and generalization behavior we seek to establish. We will focus on the case where $N = \operatorname{poly}(d)$, which we argue below is a correct scaling in which to study memorization and generalization.

For assessing the similarity of probability distributions, it is standard to use the 2-Wasserstein distance $W_2$ [Blau and Michaeli, 2017]. We will argue that when $N = \exp[d \log d]$, there is no meaningful distinction between memorization and generalization in sufficiently high dimensions. For simplicity, assume all covariance matrices $\boldsymbol{\Sigma}_\star^k$ in the definition of $\pi_\star$ in (8) are full rank. Then by [Weed and Bach, 2019, Theorem 1, Proposition 7], we have for all $d \geq 4$ that $\mathbb{E}[W_2(\pi_\star, \pi_\star^N)] \leq C_0 N^{-1/2d} \leq C_0/\sqrt{d}$, for a constant $C_0 \geq 0$. Therefore, $W_2(\pi_\star, \pi_\star^N) \to 0$ and we cannot distinguish the true distribution $\pi_\star$ from its samples $\pi_\star^N$: memorization and generalization are equivalent. On the other hand, using [Weed and Bach, 2019, Theorem 1, Proposition 2], one has for *any* draw of the empirical measure that $W_2(\pi_\star, \pi_\star^N) \geq C_1 N^{-2/d}$ for a constant $C_1 \geq 0$. In particular, if $N = \operatorname{poly}(d)$ then the 2-Wasserstein distance is lower bounded by a constant for any dimension, implying a meaningful distinction between memorization and generalization.

| Notation | Interpretation |
|----------|----------------|
| $N$ | Number of samples |
| $M$ | Capacity of the model |
| $K$ | Number of modes |
| $d$ | Problem dimension |

**Table 1:** Data and model scalings.

| Notation | Interpretation |
|----------|----------------|
| $(\boldsymbol{\mu}_\star^i)_{i=1}^K, \sigma_\star^2$ | Data generating process parameters |
| $(\boldsymbol{\mu}^i)_{i=1}^M, \sigma^2$ | Learned model parameters |
| $\bar{\boldsymbol{x}}_\star(t, \boldsymbol{x}_t)$ | MMSE denoiser, data generating process |
| $\bar{\boldsymbol{x}}_\theta(t, \boldsymbol{x}_t)$ | MMSE denoiser, learned model |
| $\bar{\boldsymbol{x}}_{\text{mem}}(t, \boldsymbol{x}_t)$ | MMSE denoiser, empirical distribution |
| $\bar{\boldsymbol{x}}_{\text{pmem},M}(t, \boldsymbol{x}_t)$ | Partially memorizing denoiser |
| $\boldsymbol{w}$ | Softmax weight vector |

**Table 2:** Data and model parameters.

## 3  Training Losses and Memorization

Now that we have set up the framework, we turn to answering the following fundamental question: *when do trained denoisers (i.e., approximate optimizers of* (5)*) memorize, and when do they generalize*? One first approach to answering this question would be to directly examine the critical points of the diffusion model training optimization problem (5) w.r.t. the denoiser parameterization (9). However, this is not straightforward; the problem is non-convex and there are many spurious critical points. Therefore, we posit a hypothesis which, roughly speaking, says that *the training losses of surrogate memorizing denoisers with $M$ parameters and generalizing denoisers is all that matters* for predicting the behavior of trained models with $M$ parameters.

To formalize this hypothesis, we will define two simple surrogate denoisers whose training losses will help us identify transitions between memorization and generalization as a function of the models' underparameterization. The first of these two summary models is the generalizing denoiser $\bar{\boldsymbol{x}}_\star$. The second is a *partially memorizing* denoiser $\bar{\boldsymbol{x}}_{\text{pmem},M}$, which memorizes the first $M$ training samples:[2]

$$\bar{\boldsymbol{x}}_{\text{pmem},M}(t, \boldsymbol{x}_t) = \sum_{i=1}^M \boldsymbol{x}^i \, \text{softmax}(\boldsymbol{w}(t, \boldsymbol{x}_t))_i, \quad \text{where} \quad w_i(t, \boldsymbol{x}_t) = -\frac{1}{2}\|\alpha_t \boldsymbol{x}^i - \boldsymbol{x}_t\|^2/\sigma_t^2. \quad (10)$$

Note that here $\boldsymbol{w} \colon \mathbb{R} \times \mathbb{R}^d \to \mathbb{R}^M$, and in the case where $M = N$, it holds $\bar{\boldsymbol{x}}_{\text{pmem},M} = \bar{\boldsymbol{x}}_{\text{mem}}$. Therefore, we can formulate our hypothesis as follows:

> *There exists a loss weighting $\lambda$ such that a trained denoiser $\bar{\boldsymbol{x}}_\theta$ with $M$ parameters memorizes if and only if $\mathcal{L}_N(\bar{\boldsymbol{x}}_{\text{pmem},M}, \lambda) \leq \mathcal{L}_N(\bar{\boldsymbol{x}}_\star, \lambda)$.*

To attempt to verify this hypothesis, we will estimate the (excess) training losses of the two denoisers $\bar{\boldsymbol{x}}_{\text{pmem},M}$ and $\bar{\boldsymbol{x}}_\star$ and compare them to develop a criterion for when a trained denoiser memorizes. For tractability, we focus on a special case of the general mixture of Gaussians model (8) where each mixture component has identical covariance, equal to a constant multiple of the identity matrix (i.e., $\boldsymbol{\Sigma}_\star^k = \sigma_\star^2 \boldsymbol{I}$ for each $k \in [K]$ for the ground truth model (8), and $\boldsymbol{\Sigma}^i = \sigma^2 \boldsymbol{I}$ for each $i \in [M]$ for general denoisers $\bar{\boldsymbol{x}}_\theta$), as reflected in Table 2. This is a common assumption in the theoretical literature on diffusion models [Shah et al., 2023, Gatmiry et al., 2024]. The structure of the learned denoiser $\bar{\boldsymbol{x}}_\theta$ implied by Lemma 2.1 simplifies to:

$$\bar{\boldsymbol{x}}_\theta(t, \boldsymbol{x}_t) = \frac{\alpha_t \sigma^2}{\alpha_t^2 \sigma^2 + \sigma_t^2} \boldsymbol{x}_t + \frac{\sigma_t^2}{\alpha_t^2 \sigma^2 + \sigma_t^2} \sum_{i=1}^M \boldsymbol{\mu}^i \, \text{softmax}(\boldsymbol{w}(\boldsymbol{x}_t))_i, \quad (11)$$

where

$$w_i(\boldsymbol{x}_t) = -\frac{1}{2}\|\alpha_t \boldsymbol{\mu}^i - \boldsymbol{x}_t\|^2/(\alpha_t^2 \sigma^2 + \sigma_t^2), \quad i = 1, \ldots, M, \quad (12)$$

with the analogous structure for $\bar{\boldsymbol{x}}_\star = \bar{\boldsymbol{x}}_{(\boldsymbol{\mu}_\star^1, \ldots, \boldsymbol{\mu}_\star^K, \sigma_\star^2)}$. In contrast to (11), the form of the memorizing and partially memorizing denoisers (7) and (10) does not change. Note that when $M \geq K$, the denoiser $\bar{\boldsymbol{x}}_\theta$ with $M$ parameters can generalize, and when $M \geq N$, it can memorize.[3]

---

[2] For our theory, the choice of the subset of samples we use in the partial memorizing denoiser is irrelevant, as long as its size is $M$ and it is chosen independently of the samples.

[3] As we show rigorously in Appendix B using properties of the softmax function, the loss (5) of any $M$-parameter denoiser $\bar{\boldsymbol{x}}_{(\boldsymbol{\mu}^1, \ldots, \boldsymbol{\mu}^M, \sigma^2)}$ can be achieved by a sequence of $M + 1$ parameter denoisers. Therefore for a fixed value of $M \geq K$, a denoiser with $M$ parameters can be (arbitrarily close to) the generalizing denoiser, i.e., denoisers with arbitrarily many parameters can generalize, and similarly for memorization with $M \geq N$.

**Training losses of memorizing and generalizing denoisers.** In what follows, we are going to state our main theoretical results which characterize the behavior of two denoisers of interest. More precisely, we compare the training loss (5) computed for the $K$-parameter generalizing denoiser $\bar{\boldsymbol{x}}_\star$ with the training loss obtained by an $M$-parameter partially memorizing denoiser (10), with $M \geq K$. To do this we compute the high dimensional asymptotics of these denoisers' excess training losses. Note that these asymptotics provide approximations (annotated with hats, i.e., $\hat{\mathcal{L}}_{N,t}$) which agree well with experiments at even moderate dimensions (see Figure 1 and Section 4).

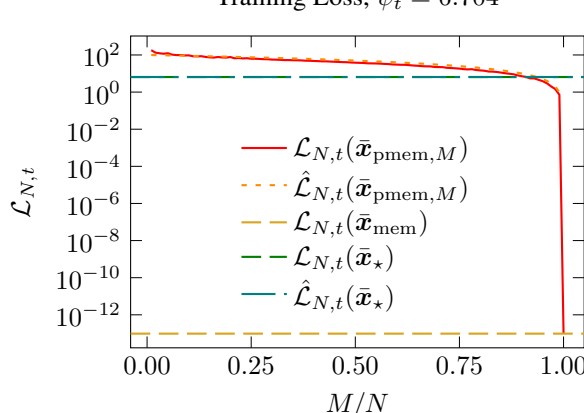

Figure 1: **We observe a remarkable degree of agreement between our loss approximations and the empirical losses.** A simulation of the loss of the partially memorizing and generalizing denoisers $\bar{\boldsymbol{x}}_\star$ and $\bar{\boldsymbol{x}}_{\mathrm{pmem},M}$ at a high-SNR value of $t$, with $d = 50$, $K = 12$, and $N = 200$, and compare them to the approximations introduced in Theorems 3.1 and 3.2. We use a constant 2 for the $\Theta(\cdot)$ expression appearing in Theorem 3.2, which we can prove is also an upper-bound.

We recall the standard re-expression of the objective in (5) via the orthogonality principle: for any $t$,

$$\mathcal{L}_{N,t}(\bar{\boldsymbol{x}}) = \frac{1}{N}\sum_{i=1}^{N} \mathbb{E}_{X_t^i}\left[\left\|\bar{\boldsymbol{x}}(t, X_t^i) - \bar{\boldsymbol{x}}_{\mathrm{mem}}(t, X_t^i)\right\|^2\right] + \frac{1}{N}\sum_{i=1}^{N} \mathbb{E}_{X_t^i}\left[\left\|\bar{\boldsymbol{x}}_{\mathrm{mem}}(t, X_t^i) - \boldsymbol{x}^i\right\|^2\right], \quad (13)$$

$$= \frac{1}{N}\sum_{i=1}^{N} \mathbb{E}_{X_t^i}\left[\left\|\bar{\boldsymbol{x}}(t, X_t^i) - \bar{\boldsymbol{x}}_{\mathrm{mem}}(t, X_t^i)\right\|^2\right] + \mathcal{L}_{N,t}(\bar{\boldsymbol{x}}_{\mathrm{mem}}),$$

where $\bar{\boldsymbol{x}}_{\mathrm{mem}}$ is the memorizing denoiser (7) and $X_t^i$ is distributed as (1) initialized with $\boldsymbol{x}^i$, i.e., $X_t^i \stackrel{d}{=} \alpha_t \boldsymbol{x}^i + \sigma_t Z$ where $Z \sim \mathcal{N}(\boldsymbol{0}, \boldsymbol{I})$.

Our first result fully characterizes the expected excess training loss of the generalizing denoiser $\bar{\boldsymbol{x}}_\star$, given by (11) with $\theta = (\boldsymbol{\mu}_\star^1, \ldots, \boldsymbol{\mu}_\star^K, \sigma_\star^2)$, over a draw of the random i.i.d. sample $(\boldsymbol{x}^1, \ldots, \boldsymbol{x}^N)$ from $\pi_\star$, under an assumption that the cluster centers $\boldsymbol{\mu}_\star^k$ are well-separated in $\ell^2$ distance. For simplicity, we state our results in the regime where $\max_k \|\boldsymbol{\mu}_\star^k\|^2 = O(d)$ and $\sigma_\star^2 = \Theta(1)$.[4] Denote the signal-to-noise ratio (SNR) of the noising process (1) by $\psi: (0,1) \to (0,+\infty)$, where $\psi_t = \alpha_t^2/\sigma_t^2$, which is assumed to be decreasing, and its inverse by $\varsigma: (0,+\infty) \to (0,1)$.

**Theorem 3.1:** *Assume that $N = \mathrm{poly}(d)$, $\min_{k \neq k'} \|\boldsymbol{\mu}_\star^k - \boldsymbol{\mu}_\star^{k'}\|^2 = \Theta(d)$, $\max_k \|\boldsymbol{\mu}_\star^k\|^2 = \Theta(d)$ and $\sigma_\star^2 = \Theta(1)$. Let $\kappa(d) = \varsigma(\Theta(\log(d)^2/d))$. We have that uniformly on $t \in [0, \kappa(d)]$*

$$\mathbb{E}_{(\boldsymbol{x}^i)_{i=1}^N} \left[\mathcal{L}_{N,t}(\bar{\boldsymbol{x}}_\star) - \mathcal{L}_{N,t}(\bar{\boldsymbol{x}}_{\mathrm{mem}})\right] = \Theta\left(\frac{d\sigma_\star^2}{\psi_t \sigma_\star^2 + 1}\right).$$

*In particular, the leading-order coefficient for the right-hand side is 1.*

Although Theorem 3.1 is stated for the excess training loss, our proofs further establish high-probability bounds on the behavior of the excess loss of the same order of magnitude. Moreover, in the scaling regime for $\sigma_\star^2$ and $(\boldsymbol{\mu}_\star^k)_{k=1}^K$ treated in Theorem 3.1, it is easily shown that for practically-relevant choices of $\alpha_t$ and $\sigma_t$, for example the scheme $\alpha_t = \sqrt{1-t^2}$, $\sigma_t = t$ that we use in our experiments in Section 4, $\kappa(d) \to 1$ as $d \to \infty$. As a consequence, for sufficiently large $d$, the uniform control of the risk for $t \in [0, \kappa(d)]$ that is established in Theorem 3.1 implies uniform control of the weighted integrated loss (5), via (13).

---

[4]To motivate this regime, note that in intuitive terms, images sampled from such a $\pi_\star$ are centered at a nominal image whose pixel intensities are normalized to $[0,1]$, and the variability due to the Gaussian noise can change each pixel by a constant amount.

Our second result estimates the excess training loss of the partially memorizing denoiser $\bar{\boldsymbol{x}}_{\mathrm{pmem},M}$.

> **Theorem 3.2:** *Assume that* $N = \mathrm{poly}(d)$, $\min_{k \neq k'} \|\boldsymbol{\mu}_\star^k - \boldsymbol{\mu}_\star^{k'}\|^2 = \Theta(d)$, $\max_k \|\boldsymbol{\mu}_\star^k\|^2 = \Theta(d)$ *and* $\sigma_\star^2 = \Theta(1)$. *Let* $\kappa(d) = \varsigma(\Theta(\log(d)^2/d))$. *We have that uniformly on* $t \in [0, \kappa(d)]$
> $$\mathbb{E}_{(\boldsymbol{x}^i)_{i=1}^N} [\mathcal{L}_{N,t}(\bar{\boldsymbol{x}}_{\mathrm{pmem},M}) - \mathcal{L}_{N,t}(\bar{\boldsymbol{x}}_{\mathrm{mem}})] = \Theta\left(\left(1 - \frac{M}{N}\right) d\sigma_\star^2\right).$$
>
> *In particular, the leading-order coefficient for the right-hand side is between* 1 *and* 2.

Note that by following the proof, we can also derive a corresponding high-probability bound.

**When does each denoiser have lower training loss?** Comparing Theorem 3.1 and Theorem 3.2, in high dimensions ($d \to \infty$) we estimate the training loss difference as

$$\mathbb{E}_{(\boldsymbol{x}^i)_{i=1}^N} [\mathcal{L}_N(\bar{\boldsymbol{x}}_{\mathrm{pmem},M}, \lambda) - \mathcal{L}_N(\bar{\boldsymbol{x}}_\star, \lambda)] \approx \mathbb{E}_t\left[\lambda(t)\left\{C\left(1 - \frac{M}{N}\right)d\sigma_\star^2 - \frac{d\sigma_\star^2}{\psi_t\sigma_\star^2 + 1}\right\}\right],$$

where $C \in [1, 2]$ is a constant. Notice that this expression is monotonically decreasing as $M \to N$, so there exists a "crossover point" $M_\star$ such that the (approximate) training loss of the partially memorizing denoiser is higher than that of the generalizing denoiser for $M \leq M_\star$ and lower for $M \geq M_\star$. Solving for $M_\star$, we obtain

$$\mathbb{E}_{(\boldsymbol{x}^i)_{i=1}^N} [\mathcal{L}_N(\bar{\boldsymbol{x}}_{\mathrm{pmem},M_\star}, \lambda) - \mathcal{L}_N(\bar{\boldsymbol{x}}_\star, \lambda)] \approx 0 \implies M_\star \approx N\left\{1 - \frac{\mathbb{E}_t\left[\lambda(t)/(\psi_t\sigma_\star^2 + 1)\right]}{C\,\mathbb{E}_t[\lambda(t)]}\right\}, \quad (14)$$

which is a *linear* function of $N$. This criterion provides a *test* for our hypothesis: if we believe that there exists a loss weighting $\lambda$ such that the memorization and generalization properties of trained denoisers are controlled by the location of the crossover point w.r.t. $\lambda$, then we should see an approximately linear relationship between the location of memorization and the number of samples. We conduct this test in Section 4, in the immediate sequel.

## 4 Experiments

In this section, we illustrate the versatility and efficacy of our laboratory through experimental analysis. First, we show that inside the setting of our laboratory, *trained* models exhibit a *phase transition from generalization to memorization*. Specifically in the isotropic Gaussian mixture setting, our main result builds a *predictive model* for the onset of memorization, and shows that it resolves to a simple linear fit as in (14), which *validates our hypothesis* from Section 3. Finally, we showcase the flexibility of our laboratory by studying a low-rank Gaussian mixture setting which is constructed to imitate the structure of training a denoiser on natural images, yet still belongs to the setting of our theoretical laboratory and exhibits similar phase transition behavior as in the simple isotropic case. We provide further details and more results, including mechanistic analyses, in Appendix H.

**High-level experiment details, and memorization criterion.** For training, we use the training loss (5) with the loss weighting $\lambda(t) = (\alpha_t/\sigma_t)^2$ (i.e., "noise prediction"). For sampling, we use the DDIM sampler [Song et al., 2020], more precisely the implementation suggested by [De Bortoli et al., 2025], with the "variance preserving" coefficients $\alpha_t = \sqrt{1 - t^2}$ and $\sigma_t = t$. We discretize the time interval $[\epsilon, 1 - \epsilon]$ uniformly into $L + 1$ timesteps $(t_\ell)_{\ell=0}^L$, where $\epsilon = 10^{-3}$, and use these timesteps for both training and sampling. For completeness we formally describe the end-to-end procedure in Appendix H.2. We consider the memorization criterion introduced in Definition 2.2 with constant $c = 1/9$, i.e., $\hat{\boldsymbol{x}}$ is memorized if $\|\hat{\boldsymbol{x}} - \boldsymbol{x}^{(1)}\|^2 \leq (1/9)\|\hat{\boldsymbol{x}} - \boldsymbol{x}^{(2)}\|^2$ where $\hat{\boldsymbol{x}}$ is the generated sample, and $\boldsymbol{x}^{(k)}$ is the $k^{\mathrm{th}}$ closest point to $\hat{\boldsymbol{x}}$ in the training dataset. Then, we say that a denoiser $\bar{\boldsymbol{x}}$ is *memorizing on average* if its *memorization ratio* is $\geq 1/2$, i.e., at least 50% of the samples $\hat{\boldsymbol{x}}$ drawn from the associated output measure $\hat{\pi}$ are memorized; we say it has *started the phase transition* if its memorization ratio is $\geq 1/10$ and *ended the phase transition* if its memorization ratio is $\geq 9/10$.

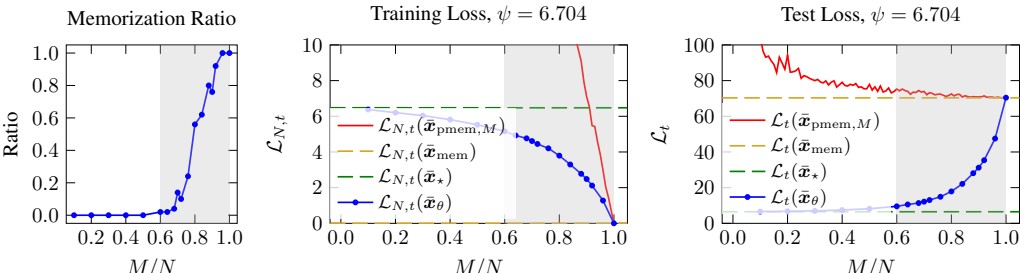

**Figure 2: We observe a clear phase transition between from generalization to memorization in trained models.** *Left:* A plot of the "memorization ratio" — the ratio of generated samples that are memorized over a set of $N_{\text{eval}} = 50$ generations — as the number of components of the trained denoiser increases. The start and end of the phase transition, as formally defined at the beginning of Section 4, are bracketed by the gray shaded region. *Middle:* A plot of the training loss of the trained model in terms of the number of components, compared to partially memorizing, fully memorizing, and ground truth denoisers, at a representative $t \approx 0.3$ with SNR $\psi_t \approx 6.704$. *Right:* A plot of the test loss $\mathcal{L}_t$, estimated over a hold-out set, in the same setting. *We observe that there is a phase transition from generalization to memorization, visible through the behavior of the memorization ratio and loss plots*; the model stops generalizing (i.e., having an acceptable test loss) only when it starts memorizing (i.e., generating samples which are approximately contained in the training data).

## 4.1 Experiments for an Isotropic Gaussian Mixture Model

First, as in Section 3, we consider the target measure $\pi_\star$ to be an isotropic Gaussian mixture model, namely, $\pi_\star = (1/K) \sum_{i=1}^{K} \mathcal{N}(\boldsymbol{\mu}_\star^i, \sigma_\star^2 \boldsymbol{I})$. As emphasized in the rest of the work, we will study what happens when we use a denoiser $\bar{\boldsymbol{x}}_\theta$ corresponding to an isotropic Gaussian mixture model with a different number of components $M$, with the parameterization given by (11).

**Existence of a phase transition.** The first and arguably most critical property of the trained denoisers in our laboratory is that *there exists a phase transition from generalization to memorization as the model size increases*, which we show in Figure 2. Namely, we observe in the rightmost panel that initially, heavily underparameterized trained denoisers exhibit statistical generalization (i.e., similar train and test loss to the ground truth denoiser $\bar{\boldsymbol{x}}_\star$); then, as the model size $M$ increases, we observe a rapid transition to *memorization* where the generated samples are memorized (nearly) 100% of the time (center and left panels). We emphasize that this replicates the central observations of many empirical studies of memorization in diffusion models, such as Zhang et al. [2023], Kadkhodaie et al. [2023].

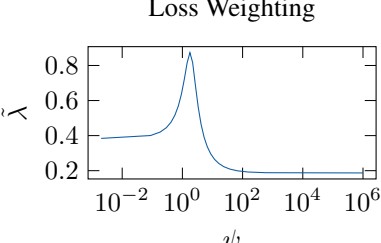

**Figure 3: We can accurately predict the phase transition using our loss approximations.** The optimal loss weighting as per (15) using normalized *approximate losses*. Train and test errors are $\leq 2 \times 10^{-4}$ when the regression targets are $\approx 10^0$.

**Predicting the phase transition.** Next, we show (via Figure 3) that the phase transition is *predictable* using *only the approximate training losses derived in Section 3*; namely, it serves as the "crossover point" of the integrated approximate training loss $\hat{\mathcal{L}}_N(\cdot, \tilde{\lambda})$ for a particular timestep weighting $\tilde{\lambda}(t)$, and this crossover point is *a linear function* of the number of training samples $N$. To show this, we compute $\tilde{\lambda}(t)$ as follows. First, we create a grid of $(N, d, K)$ tuples and train several denoisers in each setting in order to estimate the location of the phase transition $M_{\text{pt}}(N, d, K)$, the first $M$ such that the trained denoiser $\bar{\boldsymbol{x}}_\theta$ is memorizing on average. Then, we solve the following optimization problem in the loss weighting $\tilde{\lambda}$ to make $M_{\text{pt}}(N, d, K)$ close to the crossover point:

$$\min_{\tilde{\lambda}} \sum_{(N,d,K)} \left( \frac{\tilde{M}_{\text{pt}}(N, d, K, \tilde{\lambda})}{N} - \frac{M_{\text{pt}}(N, d, K)}{N} \right)^2, \tag{15}$$

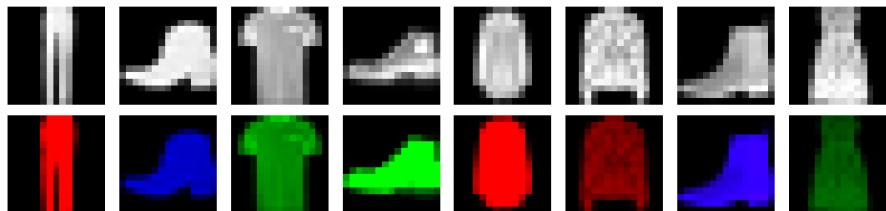

**Figure 4: Our memorization laboratory enables modeling of natural image distributions with latent low-dimensional structure.** Some sample synthetic images from our simple image model (*bottom*) visualized alongside their corresponding monochromatic image templates (*top*).

where $\tilde{M}_{\mathrm{pt}}(N, d, K, \tilde{\lambda})$ is the nearest $M$ to the crossover point with loss weighting $\tilde{\lambda}$, i.e.,

$$\tilde{M}_{\mathrm{pt}}(N, d, K, \tilde{\lambda}) = \underset{M}{\arg\min} \left( \sum_{\ell=0}^{L} \tilde{\lambda}(t_\ell) \{ \hat{\mathcal{L}}_{N,t}(\bar{\boldsymbol{x}}_{\mathrm{pmem},M}(N, d, K)) - \hat{\mathcal{L}}_{N,t}(\bar{\boldsymbol{x}}_\star(N, d, K)) \} \right)^2.$$

The optimization and normalization details are postponed to Appendix H.3. In Figure 3, we report that the train error and test error (evaluated on a holdout set) are less than $2 \times 10^{-4}$, signifying that we are able to compute the location of the memorization phase transition within one or two $M$'s on average, making our predictive model extremely accurate. Moreover, the recovered $\tilde{M}_{\mathrm{pt}}$ is always a *linear function* of $N$, namely $\tilde{M}_{\mathrm{pt}}(N, d, K, \tilde{\lambda}) = (4/5)N$. Therefore, $M_{\mathrm{pt}}(N, d, K) \approx \tilde{M}_{\mathrm{pt}}(N, d, K, \tilde{\lambda}) = (4/5)N$, demonstrating experimentally that the consequences of our hypothesis (e.g., (14)) do indeed hold. Overall, *the generalization-memorization phase transition is effectively predictable via our hypothesis and subsequent theoretical characterization of the loss*.

### 4.2 Experiments for a Simple Image Model

In the previous Section 4.1, we considered data which belonged to an isotropic Gaussian mixture model. To showcase the versatility of our Gaussian mixture model formulation, we construct a (low-rank) Gaussian mixture model whose samples resemble natural images. Namely, we consider a flattened and monochromatic $d \times d$ image ("template") $\boldsymbol{x}_\star \in \mathbb{R}^{d^2}$. Now, we endow $\boldsymbol{x}_\star$ with a *color* $\boldsymbol{c}$, sampled randomly as $\boldsymbol{c} \sim \mathcal{N}(\boldsymbol{u}_\star, \sigma_\star^2 \boldsymbol{I}) \in \mathbb{R}^c$. The colored output $\boldsymbol{y} \in \mathbb{R}^{cd^2}$ is given by $\boldsymbol{y} = \boldsymbol{c} \otimes \boldsymbol{x}_\star$ where $\otimes$ is the Kronecker product. Defining the matrix $\boldsymbol{A}_{\boldsymbol{x}} := \boldsymbol{I} \otimes \boldsymbol{x}$ we have $\boldsymbol{y} = \boldsymbol{A}_{\boldsymbol{x}_\star} \boldsymbol{c}$. The colored image $\boldsymbol{Y} \in \mathbb{R}^{c \times d \times d}$ is obtained by reshaping $\boldsymbol{y}$. Since $\boldsymbol{c} \sim \mathcal{N}(\boldsymbol{u}_\star, \sigma_\star^2 \boldsymbol{I})$ and $\boldsymbol{y} = \boldsymbol{A}_{\boldsymbol{x}_\star} \boldsymbol{c}$, it holds that $\boldsymbol{y} \sim \mathcal{N}(\boldsymbol{A}_{\boldsymbol{x}_\star} \boldsymbol{u}_\star, \sigma_\star^2 \boldsymbol{A}_{\boldsymbol{x}_\star} \boldsymbol{A}_{\boldsymbol{x}_\star}^\top)$. Note, in particular, that $\boldsymbol{y}$ is a low-rank Gaussian random variable. By combining multiple templates and color distributions, we obtain a mixture of low-rank Gaussians for our data distribution, namely, $\boldsymbol{y} \sim \pi_\star := (1/K) \sum_{i=1}^{K} \mathcal{N}(\boldsymbol{A}_\star^i \boldsymbol{u}_\star^i, \sigma_\star^2 \boldsymbol{A}_\star^i (\boldsymbol{A}_\star^i)^\top)$. This is an instance of Equation (8), and so we can compute its denoiser via Lemma 2.1. In Appendix H.4 we discuss some ways to efficiently implement this class of low-rank denoisers. We visualize some samples in Figure 4 with FashionMNIST templates [Xiao et al., 2017].

Our main result demonstrates that our framework is still expressive enough to model this facsimile of real-world image distributions, and that it exhibits similar phenomena to the isotropic case, enabling it to be studied in detail via our laboratory. Notably, we show in Figure 5 that we can still identify a phase transition phenomenon, which looks qualitatively similar to the isotropic Gaussian mixture model studied in Section 4.1 and throughout the work.

## 5 Related Works

Understanding memorization and generalization properties of diffusion models is crucial for practitioners [Somepalli et al., 2023, Ren et al., 2024, Rahman et al., 2024, Wang et al., 2024a, Chen et al., 2024b, Stein et al., 2024]. Indeed, concerns about privacy [Ghalebikesabi et al., 2023, Carlini et al., 2023, Nasr et al., 2023] and copyright infringement [Cui et al., 2023, Wang et al., 2024a, Vyas et al., 2023, Franceschelli and Musolesi, 2022] are key issues as these models are deployed. Several factors can influence the memorization capabilities of diffusion models such as duplication [Carlini et al., 2023, Ross et al., 2024, Webster, 2023] or model architecture [Chavhan et al., 2024]. We refer to [Gu et al., 2023, Kadkhodaie et al., 2023] for an in-depth experimental investigation of those issues.

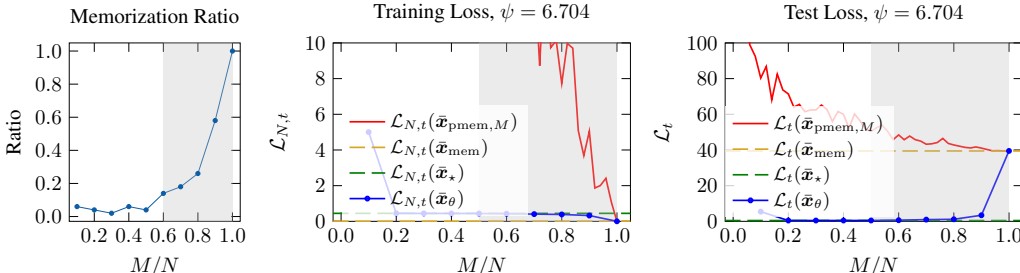

**Figure 5: A phase transition persists in the low-rank Gaussian natural image model.** *Left:* The memorization ratio as the number of components of the trained denoiser increases while everything else is fixed. *Middle:* The training loss of the trained model in terms of the number of components, compared to partially memorizing, fully memorizing, and ground truth denoisers, at a representative $t \approx 0.3$ with SNR $\psi \approx 6.704$. *Right:* The test loss, computed over a hold-out set, in the same setting. We emphasize the identical qualitative picture to Figure 2, wherein we can observe a phase transition from generalization to memorization from the memorization ratio and loss plots. Transient "jaggedness" may be explained by larger variance in the loss estimation.

On the theoretical side, memorization in diffusion models has been investigated through the lens of statistical physics leveraging the concept of *phase transition* [Biroli et al., 2024, Li et al., 2023, Ambrogioni, 2023, Ventura et al., 2024, Raya and Ambrogioni, 2024, Sakamoto et al., 2024, Pavasovic et al., 2025]. In particular in [Biroli et al., 2024], the authors identify three critical transitions for the diffusion model generative trajectories assuming that the score is assumed to be perfectly learned. George et al. [2025] investigated generalization and memorization in trained random features neural network denoisers (nonparametric models) in the case where the data is a standard Gaussian. Our work is complementary to the theories of "creativity" and generalization of Kamb and Ganguli [2024], Niedoba et al. [2024], Vastola [2025], which suggest in different contexts that generalization in diffusion models arises from the implicit bias of an underparameterized denoiser (Vastola [2025] also considers the landscape of the training objective, see [Bertrand et al., 2025] for a rebuttal). Our work disentangles the competing factors in the implicit biases discussed in these works and captures the essential features into our theoretical laboratory. While in this work we only study in detail the simplest and most interpretable instantiation of the laboratory, further connections from our framework to such previous work may be possible and enable us to broaden the scope of practical settings for which we have robust theoretical results.

## 6 Conclusion

In this paper, we have introduced a theoretical laboratory for measuring and predicting memorization and creativity in diffusion models. Focusing on data drawn from $K$-component Gaussian mixture models and Gaussian mixture denoisers with a number of components ranging from $K$ to the number of training samples $N$, our laboratory disentangles different factors contributing to memorization and generalization, enabling us to compute tight theoretical approximations for the losses of representative denoisers within this model class and thereby predict the onset of memorization in trained models at inference time.

While our current framework allows us to study generalization and memorization behavior in a rigorous and testable setting, we highlight several avenues of improvement. First, our model can be extended to capture additional properties of larger and more realistic datasets such as intrinsic dimensionality or partial data replication. In future work, we plan on expanding the memorization/generalization laboratory to cover those cases and refining the asymptotics of our training losses. We envision the laboratory growing to encompass a framework for understanding memorization and generalization purely in terms of the *geometry of the data*, with a robust and extensive experimental apparatus to test hypotheses and verify predictions, and a rich theoretical toolkit that reduces statistical questions of sampling in trained diffusion models to geometric questions about the data itself.

**Acknowledgements.** YM acknowledges support from the joint Simons Foundation-NSF DMS grant #2031899, the ONR grant N00014-22-1-2102, the NSF grant #2402951, and the startup fund from the University of Hong Kong.

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

## Organization of the Appendix

In Appendix A, we recall some basic calculations around Gaussian Mixture Model denoisers and provide additional discussion of our setting. Justifications about our model class are given in Appendix B. Next, we recall some results on Gaussian concentration bounds in Appendix C. In Appendix D, we present key results for approximating softmax operators. We combine our concentration bounds and softmax approximation results in Appendix E in order to provide approximation of the denoisers in generalizing and memorizing scenarios. Our main results regarding the training loss approximations such as Theorem 3.1 are proved in Appendix F. Finally, full experimental details are provided in Appendix H.

## A  Gaussian Mixture Model Denoisers: Basic Calculations

**Justification of Gaussian Mixture Models.**  Gaussian mixture models represent a canonical model for assessing learning and sampling algorithms in theoretical computer science, including in the context of diffusion models [Dasgupta and Schulman, 2007, Ge et al., 2018, Shah et al., 2023, Gatmiry et al., 2024], and they have the appealing ability to model data with *geometric structure*, including hierarchical structure [Li and Chen, 2024] and low-dimensional structure in natural images [Zoran and Weiss, 2011, Wang et al., 2024c] Most importantly, the task of training and sampling with a diffusion model on a Gaussian mixture target $\pi_\star$ via (5) represents an ideal test-bed for investigating issues of memorization and generalization, because as the number of components in the mixture is varied, and in particular is as made as large as the number of training data samples $N$, one can simultaneously represent the true distribution (8) and the memorizing denoiser (7) within the same class of models.

**Discussion around the denoisers.**  The key technical challenges we investigate in our laboratory setting are the characterization of the denoiser $\bar{\boldsymbol{x}}_\theta$ minimizing (6) (or, equivalently its parameters $\theta$), and using its properties to prove that its associated sampling measure $\hat{\pi}$ either memorizes or generalizes. For the first challenge, typical theoretical studies of regression over a parametric class of models (as in (5)) rely on the model class being well-specified with respect to the true model parameters, here those corresponding to $\pi_\star$. In (5), this is the case when $M = K$, but as soon as $M > K$, existing studies of the loss landscape for diffusion model training with Gaussian mixture model data/denoisers break down [Wang et al., 2024c, Shah et al., 2023]. With regards to both challenges, a clear complication that we have alluded to previously is that as $M \to N$, the model class becomes expressive enough to represent the optimal, memorizing denoiser (7), which is suggestive that at intermediate values of $M$, parameters $\theta$ that lead to generalizing-denoiser-like behavior of $\hat{\pi}$ are no longer prevalent. To overcome these challenges, we adopt the hybrid theoretical-empirical methodology outlined in the main body, which our laboratory setting enables.

**Basic denoiser calculations.**  We recall Lemma 2.1.

**Lemma A.1:** *Assume that $\pi_\theta = (1/M) \sum_{i=1}^{M} \mathcal{N}(\boldsymbol{\mu}^i, \boldsymbol{\Sigma}^i)$. Then, we have that*

$$\bar{\boldsymbol{x}}_\theta(t, \boldsymbol{x}_t) = \frac{1}{\alpha_t} \left( \boldsymbol{x}_t - \sigma_t^2 \sum_{i=1}^{M} (\alpha_t^2 \boldsymbol{\Sigma}^i + \sigma_t^2 \boldsymbol{I})^{-1} (\boldsymbol{x}_t - \alpha_t \boldsymbol{\mu}^i) \operatorname{softmax}(\boldsymbol{w}(t, \boldsymbol{x}_t))_i \right),$$

*where for all $i \in [M]$*

$$w_i(t, \boldsymbol{x}_t) = -\frac{1}{2} \log \det(\alpha_t^2 \boldsymbol{\Sigma}^i + \sigma_t^2 \boldsymbol{I}) - \frac{1}{2} (\boldsymbol{x}_t - \alpha_t \boldsymbol{\mu}^i)^\top (\alpha_t^2 \boldsymbol{\Sigma}^i + \sigma_t^2 \boldsymbol{I})^{-1} (\boldsymbol{x}_t - \alpha_t \boldsymbol{\mu}^i).$$

In the rest of this section, we are going to prove Lemma 2.1. This property is well-known and can be found in [Peluchetti, 2023, Biroli et al., 2024, Kamb and Ganguli, 2024] for instance but we include its proof for completeness. First, we compute $p_t$. Recalling the noising process, (1), we get that for any $t \in [0, 1]$ and $\boldsymbol{x}_t \in \mathbb{R}^d$

$$p_t(\boldsymbol{x}_t) = \int_{\mathbb{R}^d} p_{t|0}(\boldsymbol{x}_t|\boldsymbol{x}_0)\mathrm{d}\pi(\boldsymbol{x}_0)$$

$$= (1/M)\sum_{i=1}^{M}\int_{\mathbb{R}^d} p_{t|0}(\boldsymbol{x}_t|\boldsymbol{x}_0)\mathcal{N}(\boldsymbol{\mu}^i,\boldsymbol{\Sigma}^i)(\boldsymbol{x}_0)\mathrm{d}\boldsymbol{x}_0$$

$$= (1/M)\sum_{i=1}^{M}\mathcal{N}(\boldsymbol{x}_t;\alpha_t\boldsymbol{\mu}^i,\sigma_t^2\boldsymbol{I}+\alpha_t^2\boldsymbol{\Sigma}^i).$$

Therefore, we get that for any $t \in [0,1]$ and $\boldsymbol{x}_t \in \mathbb{R}^d$

$$\nabla\log p_t(\boldsymbol{x}_t) = \nabla p_t(\boldsymbol{x}_t)/p_t(\boldsymbol{x}_t)$$

$$= \frac{\sum_{i=1}^{M}(\sigma_t^2\boldsymbol{I}+\alpha_t^2\boldsymbol{\Sigma}^i)^{-1}(\alpha_t\boldsymbol{\mu}^i-\boldsymbol{x}_t)\mathcal{N}(\boldsymbol{x}_t;\alpha_t\boldsymbol{\mu}^i,\sigma_t^2\boldsymbol{I}+\alpha_t^2\boldsymbol{\Sigma}^i)}{\sum_{i=1}^{M}\mathcal{N}(\boldsymbol{x}_t;\alpha_t\boldsymbol{\mu}^i,\sigma_t^2\boldsymbol{I}+\alpha_t^2\boldsymbol{\Sigma}^i)}$$

$$= \sum_{i=1}^{M}(\sigma_t^2\boldsymbol{I}+\alpha_t^2\boldsymbol{\Sigma}^i)^{-1}(\alpha_t\boldsymbol{\mu}^i-\boldsymbol{x}_t)\,\mathrm{softmax}(\boldsymbol{w}(\boldsymbol{x}_t))_i, \qquad (16)$$

with $\boldsymbol{w}$ defined as in the lemma statement. Finally, using Tweedie's identity we get that for any $t \in [0,1]$ and $x_t \in \mathbb{R}^d$

$$\nabla\log p_t(\boldsymbol{x}_t) = (\alpha_t\bar{\boldsymbol{x}}_\theta(t,\boldsymbol{x}_t)-\boldsymbol{x}_t)/\sigma_t^2.$$

Therefore combining this result and (16), we get that for any $t \in [0,1]$ and $\boldsymbol{x}_t \in \mathbb{R}^d$

$$\bar{\boldsymbol{x}}_\theta(t,\boldsymbol{x}_t) = \frac{1}{\alpha_t}\left(\boldsymbol{x}_t - \sigma_t^2\sum_{i=1}^{M}(\alpha_t^2\boldsymbol{\Sigma}^i+\sigma_t^2\boldsymbol{I})^{-1}(\boldsymbol{x}_t-\alpha_t\boldsymbol{\mu}^i)\,\mathrm{softmax}(\boldsymbol{w}(\boldsymbol{x}_t))_i\right),$$

which concludes the proof.

## B  Nesting of Model Classes

Here we will sketch a rigorous proof of the claim that the loss (6) associated to any $M$-parameter GMM denoiser can be represented, in a limiting sense, by that of a $(M+1)$-parameter GMM denoiser. For simplicity, we will treat the case where $\boldsymbol{\Sigma}^i = \boldsymbol{\Sigma}$ for $i \in [M]$, i.e. all components have the same variance. Given such a $M$-parameter GMM denoiser following the form in Lemma 2.1, we show that the $M+1$ parameter model given by the parameters $(\boldsymbol{\mu}^1,\boldsymbol{\Sigma},\ldots,\boldsymbol{\mu}^M,\boldsymbol{\Sigma},m\boldsymbol{\mu}^M,\boldsymbol{\Sigma})$, for $m \in \mathbb{N}$, provides a suitable loss approximation as $m \to \infty$. Comparing the denoisers in Lemma 2.1, we see that it suffices to show a suitable degree of approximation of the softmax autoregression

$$\begin{bmatrix}\boldsymbol{\mu}^1 & \ldots & \boldsymbol{\mu}^M & m\boldsymbol{\mu}^M\end{bmatrix}\mathrm{softmax}(\boldsymbol{w}_{M+1}(\boldsymbol{x}_t)) \approx \begin{bmatrix}\boldsymbol{\mu}^1 & \ldots & \boldsymbol{\mu}^M\end{bmatrix}\mathrm{softmax}(\boldsymbol{w}_M(\boldsymbol{x}_t)),$$

where the weight vectors are subscripted to denote the number of parameters. Because we are concerned with loss approximations, it suffices to show this approximation for $\boldsymbol{x}_t$ given by a noisy sample from the GMM $\pi_\star$, following Equation (1). We have

$$\begin{bmatrix}\boldsymbol{\mu}^1 & \ldots & \boldsymbol{\mu}^M & m\boldsymbol{\mu}^M\end{bmatrix}\mathrm{softmax}(\boldsymbol{w}_{M+1}(\boldsymbol{x}_t))$$

$$= m\boldsymbol{\mu}^M\frac{e^{-\frac{1}{2}(m\alpha_t\boldsymbol{\mu}^M-\boldsymbol{x}_t)^\top(\alpha_t^2\boldsymbol{\Sigma}+\sigma_t^2\boldsymbol{I})^{-1}(m\alpha_t\boldsymbol{\mu}^M-\boldsymbol{x}_t)}}{\sum_{\ell=1}^{M}e^{w_\ell}+e^{-\frac{1}{2}(m\alpha_t\boldsymbol{\mu}^M-\boldsymbol{x}_t)^\top(\alpha_t^2\boldsymbol{\Sigma}+\sigma_t^2\boldsymbol{I})^{-1}(m\alpha_t\boldsymbol{\mu}^M-\boldsymbol{x}_t)}}$$

$$+ \sum_{k=1}^{M}\boldsymbol{\mu}^k\frac{e^{w_k}}{\sum_{\ell=1}^{M}e^{w_\ell}+e^{-\frac{1}{2}(m\alpha_t\boldsymbol{\mu}^M-\boldsymbol{x}_t)^\top(\alpha_t^2\boldsymbol{\Sigma}+\sigma_t^2\boldsymbol{I})^{-1}(m\alpha_t\boldsymbol{\mu}^M-\boldsymbol{x}_t)}},$$

where we write $w_k$ for the $k$-th entry of $\boldsymbol{w}_{M+1}$ above. From here, we can construct a high-probability event (using Gaussian concentration) on which $X_t^i$ is bounded for any sample $\boldsymbol{x}^i$ initializing the noising process (coming from (6)), and then it follows by an application of Gaussian concentration that on this event,

$$\lim_{m\to\infty}e^{-\frac{1}{2}(m\alpha_t\boldsymbol{\mu}^M-\boldsymbol{x}_t)^\top(\alpha_t^2\boldsymbol{\Sigma}+\sigma_t^2\boldsymbol{I})^{-1}(m\alpha_t\boldsymbol{\mu}^M-\boldsymbol{x}_t)} = 0,$$

whence by the previous expression

$$\begin{bmatrix} \boldsymbol{\mu}^1 & \cdots & \boldsymbol{\mu}^M & m\boldsymbol{\mu}^M \end{bmatrix} \operatorname{softmax}(\boldsymbol{w}_{M+1}(\boldsymbol{x}_t)) \to_{m\to\infty} \begin{bmatrix} \boldsymbol{\mu}^1 & \cdots & \boldsymbol{\mu}^M \end{bmatrix} \operatorname{softmax}(\boldsymbol{w}_M(\boldsymbol{x}_t)).$$

Given that loss approximations for (6) only evaluate on $X_t^i$ for which we can construct the aforementioned high-probability event, this establishes the claim that $(M+1)$-parameter models' losses can achieve any $M$-parameter model's loss.

## C Concentration Bounds

### C.1 Chernoff-Cramér bound for $\chi_2$

In this section, we give some basic results regarding the concentration bounds of $\chi_2$ random variables. Those lemmas will be key to establish our main sparsity results in Appendix E. First, we recall the Chernoff-Cramér bound, see [Boucheron et al., 2003, page 21] for instance.

**Theorem C.1:** *Let $X$ be a real-valued random variable. Let $M(t) = \mathbb{E}[\exp[tX]]$. Then, we have that for any $a \in \mathbb{R}$, $\mathbb{P}(X \leq a) \leq \inf_{t>0} M(t) \exp[-ta]$. In particular if $X \sim \chi_2(p)$ with $p \in \mathbb{N}$, we get that for any $\varepsilon \in (0,1)$*

$$\mathbb{P}(X \leq (1-\varepsilon)p) \leq \exp\left[\frac{p}{2}(\varepsilon + \log(1-\varepsilon))\right], \tag{17}$$

*and*

$$\mathbb{P}(X \geq (1+\varepsilon)p) \leq \exp\left[-\frac{p}{2}(\varepsilon - \log(1+\varepsilon))\right]. \tag{18}$$

*As a consequence for any $\varepsilon \in [0,1]$*

$$\mathbb{P}(|X-p| \geq \varepsilon p) \leq 2\exp\left[-\frac{p\varepsilon^2}{8}\right]. \tag{19}$$

*Proof.* We have that (17) and (18) are direct consequences of the Chernoff-Cramér bounds. Then for any $x \in [0,1]$, we have that $\log(1-x) + x \leq -x^2/2$ and $\log(1+x) - x \leq -x^2/2 + x^3/6$. Hence, we have that for any $x \in [0,1]$, we have that $\log(1-x) + x \leq -x^2/8$ and $\log(1+x) - x \leq -x^2/8$ which concludes the proof of (19) using an union bound. $\square$

One of the main application of Theorem C.1 is to establish the concentration of the squared norm of Gaussian random variables.

**Lemma C.2:** *Let $\boldsymbol{g} \sim \mathcal{N}(\boldsymbol{\mu}, \sigma^2 \boldsymbol{I})$ be an isotropic Gaussian with mean $\boldsymbol{\mu}$ and covariance $\sigma^2 \boldsymbol{I}$. Then for any $0 \leq \varepsilon \leq 1$, one has*

$$\mathbb{P}\left[\left|\|\boldsymbol{g}\|^2 - (\|\boldsymbol{\mu}\|^2 + \sigma^2 d)\right| \geq \varepsilon \sigma \sqrt{d}(\sigma \sqrt{d} + \|\boldsymbol{\mu}\|)\right] \leq 4\exp[-d\varepsilon^2/8].$$

*Proof.* We have $\boldsymbol{g} \stackrel{d}{=} \boldsymbol{\mu} + \sigma \boldsymbol{w}$, where $\boldsymbol{w} \sim \mathcal{N}(\boldsymbol{0}, \boldsymbol{I})$, so

$$\|\boldsymbol{g}\|^2 \stackrel{d}{=} \|\boldsymbol{\mu} + \sigma \boldsymbol{w}\|^2 = \|\boldsymbol{\mu}\|^2 + \sigma^2 \|\boldsymbol{w}\|^2 + 2\sigma\langle \boldsymbol{\mu}, \boldsymbol{w}\rangle \stackrel{d}{=} \|\boldsymbol{\mu}\|^2 + \sigma^2 \|\boldsymbol{w}\|^2 + 2\sigma\langle \boldsymbol{\mu}, \boldsymbol{w}\rangle$$

$$\stackrel{d}{=} \|\boldsymbol{\mu}\|^2 + \sigma^2 \|\boldsymbol{w}\|^2 + 2\sigma\|\boldsymbol{\mu}\|w_1 \tag{20}$$

where the last line follows by rotational invariance of the Gaussian distribution, and therefore in particular

$$\mathbb{E}\left[\|\boldsymbol{g}\|^2\right] = \|\boldsymbol{\mu}\|^2 + d\sigma^2.$$

By Theorem C.1, we have for every $0 \leq \varepsilon \leq 1$

$$\mathbb{P}\left[\left|\|\boldsymbol{w}\|^2 - d\right| \geq d\varepsilon\right] \leq 2\exp[-d\varepsilon^2/8],$$

and by Gaussian concentration, for any $t \geq 0$, we have

$$\mathbb{P}[|w_1| \geq t] \leq 2\exp[-t^2/2],$$

so in particular

$$\mathbb{P}\left[|w_1| \geq \frac{\varepsilon\sqrt{d}}{2}\right] \leq 2\exp[-d\varepsilon^2/8].$$

Thus, using a union bound, we have for any $\varepsilon \in [0,1]$

$$\mathbb{P}\left[\big|\|\boldsymbol{g}\|^2 - (\|\boldsymbol{\mu}\|^2 + \sigma^2 d)\big| \geq \varepsilon\sigma\sqrt{d}(\sigma\sqrt{d} + \|\boldsymbol{\mu}\|)\right]$$

$$= \mathbb{P}\left[\big|\sigma^2\|\boldsymbol{w}\|^2 + 2\sigma\|\boldsymbol{\mu}\|w_1 - \sigma^2 d\big| \geq \varepsilon\sigma\sqrt{d}(\sigma\sqrt{d} + \|\boldsymbol{\mu}\|)\right]$$

$$\leq \mathbb{P}\left[\sigma^2\big|\|\boldsymbol{w}\|^2 - d\big| + 2\sigma\|\boldsymbol{\mu}\||w_1| \geq \varepsilon\sigma\sqrt{d}(\sigma\sqrt{d} + \|\boldsymbol{\mu}\|)\right]$$

$$\leq \mathbb{P}\left[\sigma^2\left[\big|\|\boldsymbol{w}\|^2 - d\big| - d\varepsilon\right] + 2\sigma\|\boldsymbol{\mu}\|\left[|w_1| - \frac{\varepsilon\sqrt{d}}{2}\right] \geq 0\right]$$

$$\leq 4\exp[-d\varepsilon^2/8],$$

which concludes the proof. $\qquad\square$

The following result will be used in the proof of Theorem F.3.

**Proposition C.3:** *Let $(\boldsymbol{g}^i)_{i=1}^n$ be i.i.d Gaussian random variables $\mathcal{N}(0, \sigma_\star^2\,\mathrm{Id})$ and $(\boldsymbol{w})_{i=1}^{n-1}$ a collection of positive random variables. Then, for any $\varepsilon \in (0,1)$, we have with probability at least $1 - 6n\exp[-d\varepsilon^2/2]$*

$$d\sigma_\star^2(1 - 3\varepsilon) \leq \left\|\sum_{j=1}^{n-1}\mathrm{softmax}(\boldsymbol{w})_j\boldsymbol{x}^j - \boldsymbol{x}^n\right\|^2 \leq 2d\sigma_\star^2(1 + 3\varepsilon).$$

*Proof.* First, we have that

$$\left\|\sum_{j=1}^{n-1}\mathrm{softmax}(\boldsymbol{w})_j\boldsymbol{x}^j - \boldsymbol{x}^n\right\|^2 = \left\|\sum_{j=1}^{n-1}\mathrm{softmax}(\boldsymbol{w})_j(\boldsymbol{x}^j - \boldsymbol{x}^n)\right\|^2$$

$$\leq \sum_{j=1}^{n-1}\mathrm{softmax}(\boldsymbol{w})_j\|\boldsymbol{x}^j - \boldsymbol{x}^n\|^2.$$

Therefore, using that $\boldsymbol{x}^j - \boldsymbol{x}^n$ is a Gaussian random variable $\mathcal{N}(0, 2\sigma_\star^2\,\mathrm{Id})$, we get using a union bound and Theorem C.1, that with probability $1 - 2n\exp[-\varepsilon^2 d/8]$

$$\left\|\sum_{j=1}^{n-1}\mathrm{softmax}(\boldsymbol{w})_j\boldsymbol{x}^j - \boldsymbol{x}^n\right\|^2 \leq 2d\sigma_\star^2(1 + \varepsilon).$$

For the second part of the proof, we have that

$$\left\|\sum_{j=1}^{n-1}\mathrm{softmax}(\boldsymbol{w})_j\boldsymbol{x}^j - \boldsymbol{x}^n\right\|^2 = \|\boldsymbol{x}^n\|^2 + \left\|\sum_{j=1}^{n-1}\mathrm{softmax}(\boldsymbol{w})_j\boldsymbol{x}^j\right\|^2 - 2\sum_{j=1}^{n-1}\mathrm{softmax}(\boldsymbol{w})_j\langle\boldsymbol{x}^j, \boldsymbol{x}^n\rangle$$

$$\geq \|\boldsymbol{x}^n\|^2 - 2\sum_{j=1}^{n-1}\mathrm{softmax}(\boldsymbol{w})_j\langle\boldsymbol{x}^j, \boldsymbol{x}^n\rangle$$

$$\geq \|\boldsymbol{x}^n\|^2 - 2\sum_{j=1}^{n-1}\mathrm{softmax}(\boldsymbol{w})_j|\boldsymbol{x}_1^j|\|\boldsymbol{x}^n\|$$

$$\geq \sum_{j=1}^{n-1}\mathrm{softmax}(\boldsymbol{w})_j\{\|\boldsymbol{x}^n\|^2 - 2|\boldsymbol{x}_1^j|\|\boldsymbol{x}^n\|\}.$$

We have that with probability at least $1 - 2n\exp[-d\varepsilon^2/8]$ that $|\boldsymbol{x}_1^j| \leq \frac{\sigma_\star \varepsilon \sqrt{d}}{2}$. Combining this result with Theorem C.1 and a union bound we get that, on an event of probability at least $1 - 4n\exp[-d\varepsilon^2/8]$,

$$\left\| \sum_{j=1}^{n-1} \text{softmax}(\boldsymbol{w})_j \boldsymbol{x}^j - \boldsymbol{x}^n \right\|^2 \geq \sum_{j=1}^{n-1} \text{softmax}(\boldsymbol{w})_j \{\|\boldsymbol{x}^n\|^2 - |\boldsymbol{x}_1^j| \|\boldsymbol{x}^n\|\}$$

$$\geq \sum_{j=1}^{n-1} \text{softmax}(\boldsymbol{w})_j \{\|\boldsymbol{x}^n\|^2 - \varepsilon\sqrt{d}\|\boldsymbol{x}^n\|\}$$

$$\geq \sigma_\star^2 d(1-\varepsilon) - \varepsilon\sigma_\star^2 d(1+\varepsilon)^{1/2} \geq d(1-3\varepsilon),$$

which concludes the proof upon using a union bound. $\qquad\square$

### C.2 Coupon Collector Bounds

In order to derive our results in the case of the partial memorizing denoiser, we will consider the following result from the coupon collector's problem.

> **Proposition C.4:** *Let $K \in \mathbb{N}$ be the number of means. Consider the distribution $\pi = (1/K)\sum_{k=1}^{K} \mathcal{N}(\boldsymbol{\mu}_\star^k, \sigma_\star^2 \boldsymbol{I})$ and let $(\boldsymbol{x}^i)_{i=1}^N$ be a collection of i.i.d samples from $\pi$. For any $\boldsymbol{x}^i$ denote $k_i \in [K]$ the index of the associated mean, i.e. $\boldsymbol{x}^i = \boldsymbol{\mu}_\star^{k_i} + \sigma_\star \boldsymbol{w}^i$, with $\boldsymbol{w}^i \sim \mathcal{N}(0, \boldsymbol{I})$. Let $\mathsf{S} = (\boldsymbol{x}^i)_{i=1}^\ell$ such that $\ell \geq (1+\log(d))K\log(K)$. For any $k \in [K]$, denote $\mathsf{A}_k$ the event such that there exists $i_1, i_2 \in [\ell]$ such that $k_{i_1} = k_{i_2} = k$. Finally, denote $\mathsf{A} = \bigcap_{k=1}^K \mathsf{A}_k$. Then*
> 
> $$\mathbb{P}[\mathsf{A}] \geq 1 - K^{-\log(d)}(1 + \log(K)\log(d)).$$

This result is a simple control of the tails of the coupon collector problem. In fact, much more precise estimates could be derived, see [Erdős and Rényi, 1961] for instance.

*Proof.* We only deal with the case $K \geq 2$. The case $K = 1$ is trivial. First, note that we have for any $k \in [K]$

$$\mathbb{P}[\mathsf{A}_k^c] = \left(1 - \frac{1}{K}\right)^\ell + \frac{\ell}{K}\left(1 - \frac{1}{K}\right)^{\ell-1}$$

$$\leq \left(1 - \frac{1}{K}\right)^\ell \left(1 + \frac{\ell}{K-1}\right)$$

$$\leq \exp[-\ell/K]\left(1 + \frac{\ell}{K-1}\right).$$

Note that $t \mapsto \exp[-t/K]\frac{t}{K-1}$ is decreasing on $[K, +\infty)$ and therefore we get that

$$\mathbb{P}[\mathsf{A}_k^c] \leq \exp[-(1+\log(d))\log(K)]\left(1 + \frac{K\log(K)\log(d)}{K-1}\right).$$

Using a union bound and that $K/(K-1) \leq 2$ we get

$$\mathbb{P}[\mathsf{A}] \geq 1 - 2K^{-\log(d)}\left(1 + \log(K)\log(d)\right),$$

which concludes the proof. $\qquad\square$

## D Softmax Approximation

**Low-Temperature Behavior.** The following elementary lemma is useful. It shows that the key quantity controlling 1-sparsity of the softmax is the scale of the temperature $T$ relative to the gap between the largest and second-largest element of the softmax weight vector. For completeness we

recall the definition of the softmax operation. For any $v \in \mathbb{R}^n$ we have that $v \in \mathbb{R}^n$ and for any $i \in \{1, \dots, n\}$

$$\text{softmax}(\boldsymbol{v})_i = \frac{\exp[v_i]}{\sum_{k=1}^n \exp[v_k]}.$$

**Lemma D.1:** *Let $v \in \mathbb{R}^n$ be such that $v_i \neq v_j$ for $i \neq j$. Let $k = \arg\max_{k' \in \{1,\dots,n\}} v_{k'}$ denote the index of the largest element of $v$, and define*

$$\gamma = \min_{i \neq k} (v_k - v_i),$$

*as the "gap" between the largest and second-largest element of $v$. Then for any $p \geq 1$, one has*

$$\|\text{softmax}(\boldsymbol{v}/T) - \boldsymbol{e}_k\|_p \leq 2(n-1)e^{-\gamma/T}.$$

*Proof.* The idea is to notice that taking the ratio between elements of the softmax has a simple expression, namely

$$\frac{\text{softmax}(\boldsymbol{v}/T)_i}{\text{softmax}(\boldsymbol{v}/T)_k} = e^{-(v_k - v_i)/T},$$

where $k$ is as in the statement of the lemma. By definition of $k$, we then have for every $i \neq k$

$$\text{softmax}(\boldsymbol{v}/T)_i \leq e^{-\gamma/T}\text{softmax}(\boldsymbol{v}/T)_k \leq e^{-\gamma/T},$$

and thus

$$\text{softmax}(\boldsymbol{v}/T)_k \geq 1 - (n-1)e^{-\gamma/T}.$$

This gives

$$\|\text{softmax}(\boldsymbol{v}/T) - \boldsymbol{e}_k\|_p^p \leq (n-1)^p e^{-p\gamma/T} + (n-1)e^{-p\gamma/T},$$

so in particular

$$\|\text{softmax}(\boldsymbol{v}/T) - \boldsymbol{e}_k\|_p \leq (n-1)e^{-\gamma/T}\left(1 + (n-1)^{-1+1/p}\right)$$

$$\leq 2(n-1)e^{-\gamma/T}.$$

$\square$

Note that to guarantee approximation in $\ell^p$, in the worst case (reflected in the proof) it is necessary that the temperature depends logarithmically on the number of vector elements $n$.

This proof operates in a worst-case regime where every non-maximal element of the vector $v$ may have the same magnitude. In reality, if there is a more structured distribution of non-maximizers, the estimates improve correspondingly. The proof could be improved to capture this by using a different, more precise measure of the distribution of entries of $v$. For example, if some precise rate of decay of the entry distribution could be asserted, it seems reasonable that this, rather than the vector dimension $n$, would force the ultimate dependence of $T$ for $\ell^p$ approximation. It seems reasonable that something like this should obtain for weight vectors of distances between random vectors.

The following lemma is a slight extension of Lemma D.1.

**Lemma D.2:** *Let $v \in \mathbb{R}^n$ be such that $v_i \neq v_j$ for $i \neq j$. Let $k = \arg\max_{k' \in \{1,\dots,n\}} v_{k'}$ denote the index of the largest element of $v$. Let $\mathsf{S}$ be a subset of $[n]$ such that $k \in \mathsf{S}$ and denote*

$$\gamma_{\mathsf{S}} = \min_{i \notin \mathsf{S}} (v_k - v_i),$$

*as the "gap" between the largest of $v$ and the largest element not in $\mathsf{S}$. Denote*

$$\text{softmax}(\boldsymbol{v}_{|\mathsf{S}}/T)_\ell = \frac{\exp[v_\ell/T]}{\sum_{i \in \mathsf{S}} \exp[v_i/T]},$$

*if $\ell \in \mathsf{S}$ and $\text{softmax}(\boldsymbol{v}_{|\mathsf{S}}/T)_\ell = 0$ otherwise. Then for any $p \geq 1$, one has*

$$\left\|\text{softmax}(\boldsymbol{v}/T) - \text{softmax}(\boldsymbol{v}_{|\mathsf{S}}/T)\right\|_p \leq (1 + |\mathsf{S}|)^{1/p}(n - |\mathsf{S}|)\exp[-\gamma_{\mathsf{S}}/T].$$

Note that Lemma D.1 is a special case of Lemma D.2 where $\mathsf{S} = \{k\}$.

*Proof.* Let $\ell \notin \mathsf{S}$, we have that

$$\text{softmax}(\boldsymbol{v}/T)_\ell = \text{softmax}(\boldsymbol{v}/T)_k \exp[(-v_k + v_i)/T] \leq \exp[-\gamma_\mathsf{S}/T].$$

In addition, we have that for any $\ell \in \mathsf{S}$

$$
\begin{aligned}
\text{softmax}(\boldsymbol{v}_{|\mathsf{S}}/T)_\ell - \text{softmax}(\boldsymbol{v}/T)_\ell &= \frac{\exp[v_\ell/T]}{\sum_{i \in \mathsf{S}} \exp[v_i/T]} - \frac{\exp[v_\ell/T]}{\sum_{i \in \mathsf{S}} \exp[v_i/T] + \sum_{i \notin \mathsf{S}} \exp[v_i/T]} \\
&= \frac{\exp[v_\ell/T]}{\sum_{i \in \mathsf{S}} \exp[v_i/T]} \frac{\sum_{i \notin \mathsf{S}} \exp[v_i/T]}{\sum_{i \in \mathsf{S}} \exp[v_i/T] + \sum_{i \notin \mathsf{S}} \exp[v_i/T]} \\
&\leq \sum_{i \notin \mathsf{S}} \text{softmax}(\boldsymbol{v}/T)_i \leq (n - |\mathsf{S}|) \exp[-\gamma_\mathsf{S}/T].
\end{aligned}
$$

Therefore, we get that for any $p \geq 1$

$$
\begin{aligned}
\left\| \text{softmax}(\boldsymbol{v}/T) - \text{softmax}(\boldsymbol{v}_{|\mathsf{S}}/T) \right\|_p^p &\leq |\mathsf{S}|(n - |\mathsf{S}|)^p \exp[-\gamma_\mathsf{S} p/T] + (n - |\mathsf{S}|) \exp[-\gamma_\mathsf{S} p/T] \\
&\leq (n - |\mathsf{S}|)^p \exp[-\gamma_\mathsf{S} p/T](|\mathsf{S}| + (n - |\mathsf{S}|)^{-1+1/p}) \\
&\leq (n - |\mathsf{S}|)^p \exp[-\gamma_\mathsf{S} p/T](1 + |\mathsf{S}|),
\end{aligned}
$$

which concludes the proof. $\square$

# E Results on High-Dimensional Denoiser Behavior

In Appendix E.1, we present some results which will help us control softmax approximation in Appendix E.2, where we leverage those results to obtain controls on different denoisers.

## E.1 Gaussian Mixtures and Softmax

The following lemmas are the key to establish our main results. They allow us to use our softmax sparsity results in the context of diffusion denoisers. In particular, Lemma E.1 will be used in Lemma E.4 while Lemma E.2 will be used in Lemma E.5.

---

**Lemma E.1:** *Given vectors $(\boldsymbol{\mu}_\star^k)_{k=1}^K$ satisfying*

$$\min_{k \neq k'} \left\| \boldsymbol{\mu}_\star^k - \boldsymbol{\mu}_\star^{k'} \right\| \geq \gamma > 0,$$

*consider the distribution $\pi = (1/K) \sum_{k=1}^K \mathcal{N}(\boldsymbol{\mu}_\star^k, \sigma_\star^2 \boldsymbol{I})$. For $i \in [N]$, let $\boldsymbol{x}^i \sim \pi$, fix $0 \leq t \leq 1$, and let $\boldsymbol{x} \sim \alpha_t \boldsymbol{x}^i + \sigma_t \boldsymbol{g}$, where $\boldsymbol{g} \sim \mathcal{N}(\boldsymbol{0}, \boldsymbol{I})$ is independent from $\boldsymbol{x}^i$. Let $k_i$ denote the index of the (uniquely defined) cluster centroid $\boldsymbol{\mu}_\star^k$ associated to $\boldsymbol{x}^i$. Then for any $0 \leq \varepsilon \leq 1$ satisfying the coupling condition*

$$\varepsilon \leq \frac{\alpha_t \gamma}{4\sqrt{d(\alpha_t^2 \sigma_\star^2 + \sigma_t^2)}},$$

*one has with probability at least $1 - 4K \exp[-d\varepsilon^2/8]$*

$$\min_{k \neq k_i} \left\| \alpha_t \boldsymbol{\mu}_\star^k - \boldsymbol{x} \right\|^2 - \left\| \alpha_t \boldsymbol{\mu}_\star^{k_i} - \boldsymbol{x} \right\|^2 \geq \frac{\gamma^2 \alpha_t^2}{2}.$$

---

*Proof.* Start by writing $\boldsymbol{x}^i \stackrel{d}{=} \boldsymbol{\mu}_\star^{k_i} + \sigma_\star \boldsymbol{w}$, where $k_i \in [K]$ is unique and $\boldsymbol{w} \sim \mathcal{N}(\boldsymbol{0}, \boldsymbol{I})$ is independent, then write, for any $k \in [K]$,

$$\left\| \alpha_t \boldsymbol{\mu}_\star^k - \boldsymbol{x} \right\|^2 = \left\| \alpha_t(\boldsymbol{\mu}_\star^k - \boldsymbol{x}^i) - \sigma_t \boldsymbol{g} \right\|^2 = \left\| \alpha_t(\boldsymbol{\mu}_\star^k - \boldsymbol{\mu}_\star^{k_i} - \sigma_\star \boldsymbol{w}) - \sigma_t \boldsymbol{g} \right\|^2.$$

This is equal in distribution to the squared $\ell^2$ norm of a Gaussian random variable with mean $\alpha_t(\boldsymbol{\mu}_\star^k - \boldsymbol{\mu}_\star^{k_i})$ and covariance $(\alpha_t^2 \sigma_\star^2 + \sigma_t^2)\boldsymbol{I}$. Applying Lemma C.2, it follows

$$\mathbb{P}\left[ \left| \left\| \alpha_t \boldsymbol{\mu}_\star^k - \boldsymbol{x} \right\|^2 - (\alpha_t^2 \left\| \boldsymbol{\mu}_\star^k - \boldsymbol{\mu}_\star^{k_i} \right\|^2 + (\alpha_t^2 \sigma_\star^2 + \sigma_t^2)d) \right| \geq \varepsilon \Xi^k \right] \leq 3 \exp[-d\varepsilon^2/8],$$

where for concision $\Xi^k = \sqrt{d(\alpha_t^2 \sigma_\star^2 + \sigma_t^2)}(\sqrt{d(\alpha_t^2 \sigma_\star^2 + \sigma_t^2)} + \alpha_t \|\boldsymbol{\mu}_\star^k - \boldsymbol{\mu}_\star^{k_i}\|)$. Taking a union bound over $k \in [K]$, the above control holds for all $k$ simultaneously with probability at least $1 - 3K \exp[-d\varepsilon^2/8]$. More precisely, we have that with probability at least $1 - 3K \exp[-d\varepsilon^2/8]$, for any $k \in [K]$ with $k \neq k_i$

$$\|\alpha_t \boldsymbol{\mu}_\star^k - \boldsymbol{x}\|^2 - (\alpha_t^2 \|\boldsymbol{\mu}_\star^k - \boldsymbol{\mu}_\star^{k_i}\|^2 + (\alpha_t^2 \sigma_\star^2 + \sigma_t^2)d) \geq -\varepsilon \Xi^k, \tag{21}$$

and

$$\|\alpha_t \boldsymbol{\mu}_\star^{k_i} - \boldsymbol{x}\|^2 \leq (1 + \varepsilon)d(\alpha_t^2 \sigma_\star^2 + \sigma_t^2),$$

Using (21), we have that for any $k \in [K]$ with $k \neq k_i$

$$\|\alpha_t \boldsymbol{\mu}_\star^k - \boldsymbol{x}\|^2 \geq (1 - \varepsilon)d(\alpha_t^2 \sigma_\star^2 + \sigma_t^2) + \alpha_t^2 \|\boldsymbol{\mu}_\star^k - \boldsymbol{\mu}_\star^{k_i}\|^2 - \varepsilon \alpha_t \sqrt{d(\alpha_t^2 \sigma_\star^2 + \sigma_t^2)} \|\boldsymbol{\mu}_\star^k - \boldsymbol{\mu}_\star^{k_i}\|.$$

Consequently, we have on this event that, for each $k \in [K]$ with $k \neq k_i$,

$$\|\alpha_t \boldsymbol{\mu}_\star^k - \boldsymbol{x}\|^2 - \|\alpha_t \boldsymbol{\mu}_\star^{k_i} - \boldsymbol{x}\|^2 \geq \alpha_t^2 \|\boldsymbol{\mu}_\star^k - \boldsymbol{\mu}_\star^{k_i}\|^2$$
$$- \varepsilon \left( 2d(\alpha_t^2 \sigma_\star^2 + \sigma_t^2) + \alpha_t \sqrt{d(\alpha_t^2 \sigma_\star^2 + \sigma_t^2)} \|\boldsymbol{\mu}_\star^k - \boldsymbol{\mu}_\star^{k_i}\| \right).$$

To simplify this bound, notice that we have for all $k \in [K]$ with $k \neq k_i$

$$\frac{\alpha_t^2}{2} \|\boldsymbol{\mu}_\star^k - \boldsymbol{\mu}_\star^{k_i}\|^2 - \varepsilon \alpha_t \sqrt{d(\alpha_t^2 \sigma_\star^2 + \sigma_t^2)} \|\boldsymbol{\mu}_\star^k - \boldsymbol{\mu}_\star^{k_i}\| \geq \frac{\alpha_t^2}{4} \|\boldsymbol{\mu}_\star^k - \boldsymbol{\mu}_\star^{k_i}\|^2$$

if and only if

$$\alpha_t \gamma \geq 4\varepsilon \sqrt{d(\alpha_t^2 \sigma_\star^2 + \sigma_t^2)}.$$

Similarly, we have for all $k \neq k_i$

$$\frac{\alpha_t^2}{2} \|\boldsymbol{\mu}_\star^k - \boldsymbol{\mu}_\star^{k_i}\|^2 - 2\varepsilon d(\alpha_t^2 \sigma_\star^2 + \sigma_t^2) \geq \frac{\alpha_t^2}{4} \|\boldsymbol{\mu}_\star^k - \boldsymbol{\mu}_\star^{k_i}\|^2$$

if and only if

$$\alpha_t \gamma \geq \sqrt{8\varepsilon d(\alpha_t^2 \sigma_\star^2 + \sigma_t^2)}.$$

So it is enough to enforce

$$\max\{\varepsilon, \sqrt{\varepsilon}\} \leq \frac{\alpha_t \gamma}{4\sqrt{d(\alpha_t^2 \sigma_\star^2 + \sigma_t^2)}}$$

to get that on the aforementioned event, for every $k \in [K]$ with $k \neq k_i$

$$\min_{k \neq k_i} \|\alpha_t \boldsymbol{\mu}_\star^k - \boldsymbol{x}\|^2 - \|\alpha_t \boldsymbol{\mu}_\star^{k_i} - \boldsymbol{x}\|^2 \geq \frac{\gamma^2 \alpha_t^2}{2},$$

which concludes the proof. $\qquad \square$

The following lemma is more involved than Lemma E.1. The main reason is that contrary to Lemma E.1, the softmax weights we are investigating involve not only the means $(\boldsymbol{\mu}_\star^k)_{k=1}^K$ but the datapoints $(\boldsymbol{x}^j)_{j=1}^N$ which are random.

**Lemma E.2:** *Given vectors $(\boldsymbol{\mu}_\star^k)_{k=1}^K$ satisfying*

$$\min_{k \neq k'} \left\| \boldsymbol{\mu}_\star^k - \boldsymbol{\mu}_\star^{k'} \right\| \geq \gamma > 0,$$

*consider the distribution $\pi = (1/K)\sum_{k=1}^K \mathcal{N}(\boldsymbol{\mu}_\star^k, \sigma_\star^2 \boldsymbol{I})$. For each $j \in [N]$, let $\boldsymbol{x}^j \sim \pi$. Fix $i \in [N]$ and $0 \leq t \leq 1$, and let $\boldsymbol{x} \sim \alpha_t \boldsymbol{x}^i + \sigma_t \boldsymbol{g}$, where $\boldsymbol{g} \sim \mathcal{N}(\boldsymbol{0}, \boldsymbol{I})$ is independent from $(x^j)_{j=1}^N$. Then for any $0 \leq \varepsilon \leq 1$ satisfying the coupling conditions*

$$\varepsilon \leq \frac{\alpha_t \gamma}{2\sqrt{d(2\alpha_t^2 \sigma_\star^2 + \sigma_t^2)}}$$

*and*

$$\frac{\varepsilon^2}{1 - \varepsilon} \leq \frac{\alpha_t^2 \sigma_\star^2}{2\sigma_t^2},$$

*one has with probability at least $1 - 4N\exp[-d\varepsilon^2/8]$*

$$\min_{j \neq i} \left\| \alpha_t \boldsymbol{x}^j - \boldsymbol{x} \right\|^2 - \left\| \alpha_t \boldsymbol{x}^i - \boldsymbol{x} \right\|^2 \geq \alpha_t^2 \sigma_\star^2 (1 - \varepsilon)d.$$

*Proof.* We start by writing $\boldsymbol{x}^j \stackrel{d}{=} \boldsymbol{\mu}_\star^{k_j} + \sigma_\star \boldsymbol{w}^j$ for each $j \in [N]$, where $k_j \in [K]$ is unique and $\boldsymbol{w}^j \sim \mathcal{N}(\boldsymbol{0}, \boldsymbol{I})$ is independent. We will first argue that it is enough to consider only those indices $j$ for which the class assignments agree: that is, $k_j = k_i$, where we recall that $\boldsymbol{x} \sim \alpha_t \boldsymbol{x}^i + \sigma_t \boldsymbol{g}$ and therefore $\boldsymbol{x}$ is associated with $k_i$. We have for any $j \in [N]$

$$\alpha_t \boldsymbol{x}^j - \boldsymbol{x} \stackrel{d}{=} \alpha_t \sigma_\star (\boldsymbol{w}^j - \boldsymbol{w}^i) + \alpha_t(\boldsymbol{\mu}_\star^{k_j} - \boldsymbol{\mu}_\star^{k_i}) - \sigma_t \boldsymbol{g},$$

so by rotational invariance

$$\left\| \alpha_t \boldsymbol{x}^j - \boldsymbol{x} \right\|^2 \stackrel{d}{=} \alpha_t^2 \left\| \boldsymbol{\mu}_\star^{k_j} - \boldsymbol{\mu}_\star^{k_i} \right\|^2 + \left\| \alpha_t \sigma_\star (\boldsymbol{w}^j - \boldsymbol{w}^i) - \sigma_t \boldsymbol{g} \right\|^2$$
$$+ 2\alpha_t \left\| \boldsymbol{\mu}_\star^{k_j} - \boldsymbol{\mu}_\star^{k_i} \right\| (\alpha_t \sigma_\star (w_1^j - w_1^i) - \sigma_t g_1). \tag{22}$$

In what follows, we consider the case $j \neq i$.

**A first lower bound in the case of different means.** First, suppose that $k_j \neq k_i$. The random variable $\alpha_t \sigma_\star(w_1^j - w_1^i) - \sigma_t g_1$ is equal in distribution to a Gaussian random variable with mean zero and variance $2\alpha_t^2 \sigma_\star^2 + \sigma_t^2$. Call this random variable $X$; by Gaussian concentration, for any $t \geq 0$, we have

$$\mathbb{P}[|X| \geq t] \leq 2\exp[-t^2/2(2\alpha_t^2 \sigma_\star^2 + \sigma_t^2)],$$

so in particular

$$\mathbb{P}\left[|X| \geq \frac{\varepsilon\sqrt{d(2\alpha_t^2 \sigma_\star^2 + \sigma_t^2)}}{2}\right] \leq 2\exp[-d\varepsilon^2/8].$$

Combining this result with (22), we get that with probability at least $1 - 2|\{j \in [N] \mid k_j \neq k_i\}|\exp[-d\varepsilon^2/8]$, we have for every $j \in [N]$ for which $k_j \neq k_i$ that

$$\left\| \alpha_t \boldsymbol{x}^j - \boldsymbol{x} \right\|^2$$
$$\geq \alpha_t^2 \left\| \boldsymbol{\mu}_\star^{k_j} - \boldsymbol{\mu}_\star^{k_i} \right\|^2 + \left\| \alpha_t \sigma_\star (\boldsymbol{w}^j - \boldsymbol{w}^i) - \sigma_t \boldsymbol{g} \right\|^2 - \varepsilon \alpha_t \left\| \boldsymbol{\mu}_\star^{k_j} - \boldsymbol{\mu}_\star^{k_i} \right\| \sqrt{d(2\alpha_t^2 \sigma_\star^2 + \sigma_t^2)}.$$

Given that

$$\min_{k \neq k'} \left\| \boldsymbol{\mu}_\star^k - \boldsymbol{\mu}_\star^{k'} \right\| \geq \gamma,$$

if $\alpha_t \gamma \geq 2\varepsilon\sqrt{d(2\alpha_t^2 \sigma_\star^2 + \sigma_t^2)}$, we have on the previous event

$$\left\| \alpha_t \boldsymbol{x}^j - \boldsymbol{x} \right\|^2 \geq \frac{\alpha_t^2}{2} \left\| \boldsymbol{\mu}_\star^{k_j} - \boldsymbol{\mu}_\star^{k_i} \right\|^2 + \left\| \alpha_t \sigma_\star (\boldsymbol{w}^j - \boldsymbol{w}^i) - \sigma_t \boldsymbol{g} \right\|^2$$
$$> \left\| \alpha_t \sigma_\star (\boldsymbol{w}^j - \boldsymbol{w}^i) - \sigma_t \boldsymbol{g} \right\|^2.$$

Now, note that for those $j$ for which $k_j = k_i$, we have as above

$$\left\| \alpha_t \boldsymbol{x}^j - \boldsymbol{x} \right\|^2 \overset{d}{=} \left\| \alpha_t \sigma_\star (\boldsymbol{w}^j - \boldsymbol{w}^i) - \sigma_t \boldsymbol{g} \right\|^2.$$

Therefore, we get that with probability $1 - 2N \exp[-d\varepsilon^2/8]$ we have that

$$\left\| \alpha_t \boldsymbol{x}^j - \boldsymbol{x} \right\|^2 \geq \left\| \alpha_t \sigma_\star (\boldsymbol{w}^j - \boldsymbol{w}^i) - \sigma_t \boldsymbol{g} \right\|^2. \tag{23}$$

**Lower bound on the difference (first stochasticity level).**   We are going to give a lower bound for $j \neq i$ on the quantity

$$\left\| \alpha_t \sigma_\star (\boldsymbol{w}^j - \boldsymbol{w}^i) - \sigma_t \boldsymbol{g} \right\|^2 - \left\| \alpha_t \boldsymbol{x}^i - \boldsymbol{x} \right\|^2.$$

We have for $j \neq i$

$$\begin{aligned}
\left\| \alpha_t \sigma_\star (\boldsymbol{w}^j - \boldsymbol{w}^i) - \sigma_t \boldsymbol{g} \right\|^2 - \left\| \alpha_t \boldsymbol{x}^i - \boldsymbol{x} \right\|^2 &\overset{d}{=} \left\| \alpha_t \sigma_\star (\boldsymbol{w}^j - \boldsymbol{w}^i) - \sigma_t \boldsymbol{g} \right\|^2 - \left\| \sigma_t \boldsymbol{g} \right\|^2 \\
&= \alpha_t^2 \sigma_\star^2 \left\| \boldsymbol{w}^j - \boldsymbol{w}^i \right\|^2 - 2\sigma_t \alpha_t \sigma_\star \langle \boldsymbol{w}^j - \boldsymbol{w}^i, \boldsymbol{g} \rangle \\
&\overset{d}{=} \alpha_t^2 \sigma_\star^2 \left\| \boldsymbol{w}^j - \boldsymbol{w}^i \right\|^2 - 2\sigma_t \alpha_t \sigma_\star g_1 \left\| \boldsymbol{w}^j - \boldsymbol{w}^i \right\|
\end{aligned}$$

where the last line uses rotational invariance of the Gaussian distribution. Once again using Gaussian concentration, we have that with probability at least $1 - 2\exp[-d\varepsilon^2/8]$ that $|g_1| \leq \frac{\varepsilon\sqrt{d}}{2}$. It follows from a union bound that, on an event of probability at least $1 - 2N\exp[-d\varepsilon^2/8]$, it holds

$$\left\| \alpha_t \sigma_\star (\boldsymbol{w}^j - \boldsymbol{w}^i) - \sigma_t \boldsymbol{g} \right\|^2 - \left\| \alpha_t \boldsymbol{x}^i - \boldsymbol{x} \right\|^2 \geq \alpha_t^2 \sigma_\star^2 \left\| \boldsymbol{w}^j - \boldsymbol{w}^i \right\|^2 - \varepsilon\sqrt{d}\sigma_t \alpha_t \sigma_\star \left\| \boldsymbol{w}^j - \boldsymbol{w}^i \right\|.$$

**Lower bound on the difference (second stochasticity level).**   Now, as before, if it holds for all such $j$

$$\left\| \boldsymbol{w}^j - \boldsymbol{w}^i \right\| \geq \frac{2\varepsilon\sigma_t\sqrt{d}}{\alpha_t \sigma_\star},$$

then the preceding bound can be simplified to

$$\left\| \alpha_t \sigma_\star (\boldsymbol{w}^j - \boldsymbol{w}^i) - \sigma_t \boldsymbol{g} \right\|^2 - \left\| \alpha_t \boldsymbol{x}^i - \boldsymbol{x} \right\|^2 \geq \frac{\alpha_t^2 \sigma_\star^2}{2} \left\| \boldsymbol{w}^j - \boldsymbol{w}^i \right\|^2.$$

This leads us to consider the lower tail of the random variable $\min_{j \neq i} \left\| \boldsymbol{w}^j - \boldsymbol{w}^i \right\|$, which was studied in Appendix C. A coarser approach will be sufficient for our purposes: when $j \neq i$, the random variable $\frac{1}{2} \left\| \boldsymbol{w}^j - \boldsymbol{w}^i \right\|^2$ is distributed as a $\chi_2(d)$ random variable, so Theorem C.1 implies that for any $0 \leq \varepsilon' \leq 1$,

$$\mathbb{P}\left[ \left\| \boldsymbol{w}^j - \boldsymbol{w}^i \right\|^2 \geq 2(1 - \varepsilon')d \right] \geq 1 - \exp\left[ \frac{-d(\varepsilon')^2}{8} \right].$$

We can for simplicity simply enforce $\varepsilon' = \varepsilon$. In this case, if we add the additional condition

$$\frac{\varepsilon^2}{1 - \varepsilon} \leq \frac{\alpha_t^2 \sigma_\star^2}{2\sigma_t^2},$$

then by a union bound, it holds with probability at least $1 - N\exp[-d\varepsilon^2/8]$ that for all such $j$,

$$\left\| \alpha_t \sigma_\star (\boldsymbol{w}^j - \boldsymbol{w}^i) - \sigma_t \boldsymbol{g} \right\|^2 - \left\| \alpha_t \boldsymbol{x}^i - \boldsymbol{x} \right\|^2 \geq \frac{\alpha_t^2 \sigma_\star^2}{2} \left\| \boldsymbol{w}^j - \boldsymbol{w}^i \right\|^2 \geq \alpha_t^2 \sigma_\star^2 (1 - \varepsilon)d, \tag{24}$$

Finally combining (24), (23) and a union bound, we get that with probability at least $1 - 4N\exp[-d\varepsilon^2/8]$, we have

$$\min_{j \neq i} \left\| \alpha_t \boldsymbol{x}^j - \boldsymbol{x} \right\|^2 - \left\| \alpha_t \boldsymbol{x}^i - \boldsymbol{x} \right\|^2 \geq \alpha_t^2 \sigma_\star^2 (1 - \varepsilon)d,$$

which concludes the proof. $\qquad\square$

**Lemma E.3:** *Given vectors $(\boldsymbol{\mu}_\star^k)_{k=1}^K$ satisfying*

$$\min_{k \neq k'} \left\| \boldsymbol{\mu}_\star^k - \boldsymbol{\mu}_\star^{k'} \right\| \geq \gamma > 0,$$

*consider the distribution $\pi = (1/K) \sum_{k=1}^K \mathcal{N}(\boldsymbol{\mu}_\star^k, \sigma_\star^2 \boldsymbol{I})$. For each $j \in [N]$, let $\boldsymbol{x}^j \sim \pi$. Fix $i \in [N]$ and $0 \leq t \leq 1$, and let $\boldsymbol{x} \sim \alpha_t \boldsymbol{x}^i + \sigma_t \boldsymbol{g}$, where $\boldsymbol{g} \sim \mathcal{N}(\boldsymbol{0}, \boldsymbol{I})$ is independent from $(x^j)_{j=1}^N$. Then for any $0 \leq \varepsilon \leq 1$ satisfying the coupling conditions*

$$\varepsilon \leq \frac{\alpha_t \gamma}{2\sqrt{d(2\alpha_t^2 \sigma_\star^2 + \sigma_t^2)}}$$

*and*

$$\frac{\varepsilon^2}{1 - \varepsilon} \leq \frac{\alpha_t^2 \sigma_\star^2}{2\sigma_t^2},$$

*one has with probability at least $1 - 4N \exp[-d\varepsilon^2/8]$*

$$\min_{j \notin \mathsf{S}_i} \left\| \alpha_t \boldsymbol{x}^j - \boldsymbol{x} \right\|^2 - \left\| \alpha_t \boldsymbol{x}^i - \boldsymbol{x} \right\|^2 \geq \frac{\alpha_t^2 \gamma^2}{2},$$

*where $\mathsf{S}_i$, where $j \in \mathsf{S}_i$ if $k_j = k_i$, where $k_j$ is the (unique) index of the mean in $(\boldsymbol{\mu}_\star^k)_{k=1}^K$ associated with $\boldsymbol{x}^j$.*

The proof of this lemma is similar to the one of Lemma E.2.

*Proof.* We start by writing $\boldsymbol{x}^j \stackrel{d}{=} \boldsymbol{\mu}_\star^{k_j} + \sigma_\star \boldsymbol{w}^j$ for each $j \in [N]$, where $k_j \in [K]$ is unique and $\boldsymbol{w}^j \sim \mathcal{N}(\boldsymbol{0}, \boldsymbol{I})$ is independent. We have for any $j \in [N]$

$$\alpha_t \boldsymbol{x}^j - \boldsymbol{x} \stackrel{d}{=} \alpha_t \sigma_\star (\boldsymbol{w}^j - \boldsymbol{w}^i) + \alpha_t (\boldsymbol{\mu}_\star^{k_j} - \boldsymbol{\mu}_\star^{k_i}) - \sigma_t \boldsymbol{g},$$

so by rotational invariance

$$\left\| \alpha_t \boldsymbol{x}^j - \boldsymbol{x} \right\|^2 \stackrel{d}{=} \alpha_t^2 \left\| \boldsymbol{\mu}_\star^{k_j} - \boldsymbol{\mu}_\star^{k_i} \right\|^2 + \left\| \alpha_t \sigma_\star (\boldsymbol{w}^j - \boldsymbol{w}^i) - \sigma_t \boldsymbol{g} \right\|^2$$
$$+ 2\alpha_t \left\| \boldsymbol{\mu}_\star^{k_j} - \boldsymbol{\mu}_\star^{k_i} \right\| (\alpha_t \sigma_\star (w_1^j - w_1^i) - \sigma_t g_1). \tag{25}$$

In what follows, we consider the case $j \neq i$. First, suppose that $k_j \neq k_i$. The random variable $\alpha_t \sigma_\star (w_1^j - w_1^i) - \sigma_t g_1$ is equal in distribution to a Gaussian random variable with mean zero and variance $2\alpha_t^2 \sigma_\star^2 + \sigma_t^2$. Call this random variable $X$; by Gaussian concentration, for any $t \geq 0$, we have

$$\mathbb{P}[|X| \geq t] \leq 2 \exp[-t^2/2(2\alpha_t^2 \sigma_\star^2 + \sigma_t^2)],$$

so in particular

$$\mathbb{P}\left[ |X| \geq \frac{\varepsilon \sqrt{d(2\alpha_t^2 \sigma_\star^2 + \sigma_t^2)}}{2} \right] \leq 2 \exp[-d\varepsilon^2/8].$$

Combining this result with (25), we get that with probability at least $1 - 2|\{j \in [N] \mid k_j \neq k_i\}| \exp[-d\varepsilon^2/8]$, we have for every $j \in [N]$ for which $k_j \neq k_i$ that

$$\left\| \alpha_t \boldsymbol{x}^j - \boldsymbol{x} \right\|^2$$
$$\geq \alpha_t^2 \left\| \boldsymbol{\mu}_\star^{k_j} - \boldsymbol{\mu}_\star^{k_i} \right\|^2 + \left\| \alpha_t \sigma_\star (\boldsymbol{w}^j - \boldsymbol{w}^i) - \sigma_t \boldsymbol{g} \right\|^2 - \varepsilon \alpha_t \left\| \boldsymbol{\mu}_\star^{k_j} - \boldsymbol{\mu}_\star^{k_i} \right\| \sqrt{d(2\alpha_t^2 \sigma_\star^2 + \sigma_t^2)}.$$

Given that

$$\min_{k \neq k'} \left\| \boldsymbol{\mu}_\star^k - \boldsymbol{\mu}_\star^{k'} \right\| \geq \gamma,$$

if $\alpha_t \gamma \geq 2\varepsilon \sqrt{d(2\alpha_t^2 \sigma_\star^2 + \sigma_t^2)}$, we have on the previous event

$$\left\| \alpha_t \boldsymbol{x}^j - \boldsymbol{x} \right\|^2 \geq \frac{\alpha_t^2}{2} \left\| \boldsymbol{\mu}_\star^{k_j} - \boldsymbol{\mu}_\star^{k_i} \right\|^2 + \left\| \alpha_t \sigma_\star (\boldsymbol{w}^j - \boldsymbol{w}^i) - \sigma_t \boldsymbol{g} \right\|^2$$
$$\geq \left\| \alpha_t \sigma_\star (\boldsymbol{w}^j - \boldsymbol{w}^i) - \sigma_t \boldsymbol{g} \right\|^2 + \frac{\alpha_t^2 \gamma^2}{2}.$$

Therefore, we have that with probability at least $1 - 2N \exp[-d\varepsilon^2/8]$ for any $j \notin \mathsf{S}_i$

$$\left\| \alpha_t \boldsymbol{x}^j - \boldsymbol{x} \right\|^2 \geq \left\| \alpha_t \sigma_\star (\boldsymbol{w}^j - \boldsymbol{w}^i) - \sigma_t \boldsymbol{g} \right\|^2 + \frac{\alpha_t^2 \gamma^2}{2}. \tag{26}$$

**Lower bound on the difference (first stochasticity level).** We are going to give a lower bound for $j \neq i$ on the quantity

$$\left\| \alpha_t \sigma_\star (\boldsymbol{w}^j - \boldsymbol{w}^i) - \sigma_t \boldsymbol{g} \right\|^2 - \left\| \alpha_t \boldsymbol{x}^i - \boldsymbol{x} \right\|^2.$$

We have for $j \neq i$

$$\begin{aligned}
\left\| \alpha_t \sigma_\star (\boldsymbol{w}^j - \boldsymbol{w}^i) - \sigma_t \boldsymbol{g} \right\|^2 - \left\| \alpha_t \boldsymbol{x}^i - \boldsymbol{x} \right\|^2 &\stackrel{d}{=} \left\| \alpha_t \sigma_\star (\boldsymbol{w}^j - \boldsymbol{w}^i) - \sigma_t \boldsymbol{g} \right\|^2 - \left\| \sigma_t \boldsymbol{g} \right\|^2 \\
&= \alpha_t^2 \sigma_\star^2 \left\| \boldsymbol{w}^j - \boldsymbol{w}^i \right\|^2 - 2\sigma_t \alpha_t \sigma_\star \langle \boldsymbol{w}^j - \boldsymbol{w}^i, \boldsymbol{g} \rangle \\
&\stackrel{d}{=} \alpha_t^2 \sigma_\star^2 \left\| \boldsymbol{w}^j - \boldsymbol{w}^i \right\|^2 - 2\sigma_t \alpha_t \sigma_\star g_1 \left\| \boldsymbol{w}^j - \boldsymbol{w}^i \right\|
\end{aligned}$$

where the last line uses rotational invariance of the Gaussian distribution. Once again using Gaussian concentration, we have that with probability at least $1 - 2\exp[-d\varepsilon^2/8]$ that $|g_1| \leq \frac{\varepsilon\sqrt{d}}{2}$. It follows from a union bound that, on an event of probability at least $1 - 2N \exp[-d\varepsilon^2/8]$, it holds

$$\left\| \alpha_t \sigma_\star (\boldsymbol{w}^j - \boldsymbol{w}^i) - \sigma_t \boldsymbol{g} \right\|^2 - \left\| \alpha_t \boldsymbol{x}^i - \boldsymbol{x} \right\|^2 \geq \alpha_t^2 \sigma_\star^2 \left\| \boldsymbol{w}^j - \boldsymbol{w}^i \right\|^2 - \varepsilon\sqrt{d}\sigma_t \alpha_t \sigma_\star \left\| \boldsymbol{w}^j - \boldsymbol{w}^i \right\|.$$

**Lower bound on the difference (second stochasticity level).** Now, as before, if it holds for all such $j$

$$\left\| \boldsymbol{w}^j - \boldsymbol{w}^i \right\| \geq \frac{2\varepsilon\sigma_t\sqrt{d}}{\alpha_t \sigma_\star},$$

then the preceding bound can be simplified to

$$\left\| \alpha_t \sigma_\star (\boldsymbol{w}^j - \boldsymbol{w}^i) - \sigma_t \boldsymbol{g} \right\|^2 - \left\| \alpha_t \boldsymbol{x}^i - \boldsymbol{x} \right\|^2 \geq \frac{\alpha_t^2 \sigma_\star^2}{2} \left\| \boldsymbol{w}^j - \boldsymbol{w}^i \right\|^2.$$

This leads us to consider the lower tail of the random variable $\min_{j \neq i} \| \boldsymbol{w}^j - \boldsymbol{w}^i \|$, which was studied in Appendix C. A coarser approach will be sufficient for our purposes: when $j \neq i$, the random variable $\frac{1}{2} \| \boldsymbol{w}^j - \boldsymbol{w}^i \|^2$ is distributed as a $\chi_2(d)$ random variable, so Theorem C.1 implies that for any $0 \leq \varepsilon' \leq 1$,

$$\mathbb{P}\left[ \left\| \boldsymbol{w}^j - \boldsymbol{w}^i \right\|^2 \geq 2(1 - \varepsilon')d \right] \geq 1 - \exp\left[ \frac{-d(\varepsilon')^2}{8} \right].$$

We can for simplicity simply enforce $\varepsilon' = \varepsilon$. In this case, if we add the additional condition

$$\frac{\varepsilon^2}{1 - \varepsilon} \leq \frac{\alpha_t^2 \sigma_\star^2}{2\sigma_t^2},$$

then by a union bound, it holds with probability at least $1 - N \exp[-d\varepsilon^2/8]$ that for all such $j$,

$$\left\| \alpha_t \sigma_\star (\boldsymbol{w}^j - \boldsymbol{w}^i) - \sigma_t \boldsymbol{g} \right\|^2 - \left\| \alpha_t \boldsymbol{x}^i - \boldsymbol{x} \right\|^2 \geq \frac{\alpha_t^2 \sigma_\star^2}{2} \left\| \boldsymbol{w}^j - \boldsymbol{w}^i \right\|^2 \geq \alpha_t^2 \sigma_\star^2 (1 - \varepsilon)d, \tag{27}$$

Finally combining (27), (26) and a union bound, we get that with probability at least $1 - 4N \exp[-d\varepsilon^2/8]$, we have

$$\min_{j \notin \mathsf{S}_i} \left\| \alpha_t \boldsymbol{x}^j - \boldsymbol{x} \right\|^2 - \left\| \alpha_t \boldsymbol{x}^i - \boldsymbol{x} \right\|^2 \geq \alpha_t^2 \sigma_\star^2 (1 - \varepsilon)d + \frac{\alpha_t^2 \gamma^2}{2},$$

which concludes the proof. $\square$

### E.2 Denoiser Approximations

Finally, we prove Lemma E.4 and Lemma E.5. Those results control the approximation of the true denoiser in Lemma E.4 and the memorizing denoiser in Lemma E.5. Those approximations express that under certain conditions the true denoiser can be replaced by a Gaussian denoiser and that under certain conditions the memorizing denoiser can be replaced by a point in the dataset.

**Lemma E.4:** *Given vectors $(\boldsymbol{\mu}_\star^k)_{k=1}^K$ satisfying*

$$\min_{k \neq k'} \left\| \boldsymbol{\mu}_\star^k - \boldsymbol{\mu}_\star^{k'} \right\| \geq \gamma > 0,$$

*consider the distribution $\pi = (1/K) \sum_{k=1}^K \mathcal{N}(\boldsymbol{\mu}_\star^k, \sigma_\star^2 \boldsymbol{I})$. For $i \in [N]$, let $\boldsymbol{x}^i \sim \pi$, fix $0 \leq t \leq 1$, and let $\boldsymbol{x}_t^i \sim \alpha_t \boldsymbol{x}^i + \sigma_t \boldsymbol{g}$, where $\boldsymbol{g} \sim \mathcal{N}(\boldsymbol{0}, \boldsymbol{I})$ is independent from $\boldsymbol{x}^i$. Let $k_i$ denote the index of the (uniquely defined) cluster centroid $\boldsymbol{\mu}_\star^k$ associated to $\boldsymbol{x}^i$. Define the nearest-neighbor (one sparse) denoiser $\bar{\boldsymbol{x}}^{\mathrm{nom}}(t, \boldsymbol{x}_t^i)$ associated to $\bar{\boldsymbol{x}}_\star(t, \boldsymbol{x}_t^i)$ (recall Lemma 2.1) by*

$$\bar{\boldsymbol{x}}^{\mathrm{nom}}(t, \boldsymbol{x}_t^i) = \frac{\alpha_t \sigma_\star^2}{\alpha_t^2 \sigma_\star^2 + \sigma_t^2} \boldsymbol{x}_t^i + \frac{\sigma_t^2}{\alpha_t^2 \sigma_\star^2 + \sigma_t^2} \boldsymbol{\mu}_\star^{k_i}.$$

*Then for any $0 \leq \varepsilon \leq 1$ satisfying the coupling condition*

$$\varepsilon \leq \frac{\alpha_t \gamma}{4\sqrt{d(\alpha_t^2 \sigma_\star^2 + \sigma_t^2)}},$$

*one has with probability at least $1 - 4K \exp[-d\varepsilon^2/8]$*

$$\left\| \bar{\boldsymbol{x}}_\star(t, \boldsymbol{x}_t^i) - \bar{\boldsymbol{x}}^{\mathrm{nom}}(t, \boldsymbol{x}_t^i) \right\| \leq \frac{2K \sigma_t^2 \max_{k \in [K]} \|\boldsymbol{\mu}_\star^k\|}{\alpha_t^2 \sigma_\star^2 + \sigma_t^2} \exp\left[ -\frac{\gamma^2 \alpha_t^2}{4(\alpha_t^2 \sigma_\star^2 + \sigma_t^2)} \right],$$

*Proof.* We apply Lemmas D.1 and E.1. Our hypotheses let us apply Lemma E.1, which gives that with probability at least $1 - 4K \exp[-d\varepsilon^2/8]$

$$\min_{k \neq k_i} \left\| \alpha_t \boldsymbol{\mu}_\star^k - \boldsymbol{x}_t^i \right\|^2 - \left\| \alpha_t \boldsymbol{\mu}_\star^{k_i} - \boldsymbol{x}_t^i \right\|^2 \geq \frac{\gamma^2 \alpha_t^2}{2}.$$

Recall the expression for the denoiser (Lemma 2.1, (12)). The weight vector

$$w_k(\boldsymbol{x}_t^i) = -\frac{1}{2} \|\alpha_t \boldsymbol{\mu}_\star^k - \boldsymbol{x}_t^i\|^2 / (\alpha_t^2 \sigma_\star^2 + \sigma_t^2)$$

is related to the gap condition we have asserted: in particular with probability at least $1 - 4K \exp[-d\varepsilon^2/8]$

$$\min_{k \neq k_i} w_{k_i}(\boldsymbol{x}_t^i) - w_k(\boldsymbol{x}_t^i) \geq \frac{\gamma^2 \alpha_t^2}{4(\alpha_t^2 \sigma_\star^2 + \sigma_t^2)}.$$

Hence, an application of Lemma D.1 gives that for any $p \geq 1$,

$$\left\| \mathrm{softmax}(\boldsymbol{w}(\boldsymbol{x}_t^i)) - \boldsymbol{e}_{k_i} \right\|_p \leq 2(K-1) \exp\left[ -\frac{\gamma^2 \alpha_t^2}{4(\alpha_t^2 \sigma_\star^2 + \sigma_t^2)} \right].$$

Using this result, we get

$$\left\| \bar{\boldsymbol{x}}_\star(t, \boldsymbol{x}_t^i) - \bar{\boldsymbol{x}}^{\mathrm{nom}}(t, \boldsymbol{x}_t^i) \right\| = \frac{\sigma_t^2}{\alpha_t^2 \sigma_\star^2 + \sigma_t^2} \left\| \boldsymbol{M}(\boldsymbol{e}_{k_i} - \boldsymbol{w}(\boldsymbol{x}_t^i)) \right\|$$

$$\leq \frac{\sigma_t^2}{\alpha_t^2 \sigma_\star^2 + \sigma_t^2} \Gamma \left\| \boldsymbol{e}_{k_i} - \boldsymbol{w}(\boldsymbol{x}_t^i) \right\|_1$$

$$\leq \frac{2\Gamma(K-1)\sigma_t^2}{\alpha_t^2 \sigma_\star^2 + \sigma_t^2} \exp\left[ -\frac{\gamma^2 \alpha_t^2}{4(\alpha_t^2 \sigma_\star^2 + \sigma_t^2)} \right],$$

where we have defined

$$\boldsymbol{M} = \begin{bmatrix} \boldsymbol{\mu}_\star^1 & \cdots & \boldsymbol{\mu}_\star^K \end{bmatrix} \in \mathbb{R}^{d \times K}, \quad \Gamma = \max_{k \in [K]} \|\boldsymbol{\mu}_\star^k\|,$$

which concludes the proof.

$\square$

**Lemma E.5:** *Given vectors $(\boldsymbol{\mu}_\star^k)_{k=1}^K$ satisfying*

$$\min_{k \neq k'} \left\| \boldsymbol{\mu}_\star^k - \boldsymbol{\mu}_\star^{k'} \right\| \geq \gamma > 0,$$

*consider the distribution $\pi = (1/K) \sum_{k=1}^K \mathcal{N}(\boldsymbol{\mu}_\star^k, \sigma_\star^2 \boldsymbol{I})$. For $i \in [N]$, let $\boldsymbol{x}^i \sim \pi$, fix $0 \leq t \leq 1$, and let $\boldsymbol{x}_t^i \sim \alpha_t \boldsymbol{x}^i + \sigma_t \boldsymbol{g}$, where $\boldsymbol{g} \sim \mathcal{N}(\boldsymbol{0}, \boldsymbol{I})$ is independent from $\boldsymbol{x}^i$. Then for any $0 \leq \varepsilon \leq 1$ satisfying the coupling conditions*

$$\varepsilon \leq \frac{\alpha_t \gamma}{2\sqrt{d(2\alpha_t^2 \sigma_\star^2 + \sigma_t^2)}},$$

*and*

$$\frac{\varepsilon^2}{1 - \varepsilon} \leq \frac{\alpha_t^2 \sigma_\star^2}{2\sigma_t^2},$$

*one has with probability at least $1 - 8N \exp[-d\varepsilon^2/8]$*

$$\left\| \bar{\boldsymbol{x}}_{\mathrm{mem}}(t, \boldsymbol{x}_t^i) - \boldsymbol{x}^i \right\| \leq 2N\Gamma_\star \exp\left[ -\frac{\alpha_t^2 \sigma_\star^2(1 - \varepsilon)d}{2\sigma_t^2} \right],$$

*with*

$$\Gamma_\star = \max_{k \in [K]} \sqrt{\|\boldsymbol{\mu}_\star^k\|^2 + \varepsilon \sigma_\star \sqrt{d}\|\boldsymbol{\mu}_\star^k\| + (1 + \varepsilon)\sigma_\star^2 d}.$$

*Proof.* The proof is similar to that of Lemma E.4: we apply Lemmas D.1 and E.2. Our hypotheses let us apply Lemma E.2, which gives that with probability at least $1 - 4N \exp[-d\varepsilon^2/8]$

$$\min_{j \neq i} \left\| \alpha_t \boldsymbol{x}^j - \boldsymbol{x}_t^i \right\|^2 - \left\| \alpha_t \boldsymbol{x}^i - \boldsymbol{x}_t^i \right\|^2 \geq \alpha_t^2 \sigma_\star^2 (1 - \varepsilon)d.$$

Recall the expression for the denoiser (Lemma 2.1, (12)). The memorizing denoiser corresponds to $\sigma_\star = 0$, so the weight vector

$$w_j(\boldsymbol{x}_t^i) = -\frac{1}{2}\|\alpha_t \boldsymbol{x}^j - \boldsymbol{x}_t^i\|^2/\sigma_t^2,$$

is related to the gap condition we have asserted: in particular with probability at least $1 - 4N \exp[-d\varepsilon^2/8]$

$$\min_{j \neq i} w_j(\boldsymbol{x}_t^i) - w_i(\boldsymbol{x}_t^i) \geq \frac{\alpha_t^2 \sigma_\star^2(1 - \varepsilon)d}{2\sigma_t^2}.$$

Hence, an application of Lemma D.1 gives that for any $p \geq 1$,

$$\left\| \mathrm{softmax}(\boldsymbol{w}(\boldsymbol{x}_t^i)) - \boldsymbol{e}_i \right\|_p \leq 2(N - 1) \exp\left[ -\frac{\alpha_t^2 \sigma_\star^2(1 - \varepsilon)d}{2\sigma_t^2} \right].$$

Using this result, we get

$$
\begin{aligned}
\left\| \bar{\boldsymbol{x}}_{\mathrm{mem}}(t, \boldsymbol{x}_t^i) - \boldsymbol{x}^i \right\| &= \left\| \boldsymbol{M}(\boldsymbol{e}_i - \boldsymbol{w}(\boldsymbol{x}_t^i)) \right\| \\
&\leq \Gamma \left\| \boldsymbol{e}_{k_i} - \boldsymbol{w}(\boldsymbol{x}_t^i) \right\|_1 \\
&\leq 2\Gamma(N - 1) \exp\left[ -\frac{\alpha_t^2 \sigma_\star^2(1 - \varepsilon)d}{2\sigma_t^2} \right],
\end{aligned}
$$

where we have defined

$$\boldsymbol{M} = \begin{bmatrix} \boldsymbol{x}^1 & \cdots & \boldsymbol{x}^N \end{bmatrix} \in \mathbb{R}^{d \times N}, \quad \Gamma = \max_{j \in [N]} \|\boldsymbol{x}^j\|.$$

Finally, using Lemma C.2, we have that for any $0 \leq \varepsilon \leq 1$

$$\mathbb{P}\left[ \left| \|\boldsymbol{x}^j\|^2 - (\|\boldsymbol{\mu}_\star^{k_j}\|^2 + \sigma_\star^2 d) \right| \geq \varepsilon \sigma_\star \sqrt{d}(\sigma_\star \sqrt{d} + \|\boldsymbol{\mu}_\star^{k_j}\|) \right] \leq 3 \exp[-d\varepsilon^2/8],$$

so with probability at least $1 - 4N \exp[-d\varepsilon^2/8]$ we have

$$\Gamma \leq \max_{k \in [K]} \sqrt{\|\boldsymbol{\mu}_\star^k\|^2 + \varepsilon \sigma_\star \sqrt{d}\|\boldsymbol{\mu}_\star^k\| + (1 + \varepsilon)\sigma_\star^2 d}.$$

By a union bound, we conclude that with probability at least $1 - 8N \exp[-d\varepsilon^2/8]$ that

$$\left\| \bar{\boldsymbol{x}}_{\mathrm{mem}}(t, \boldsymbol{x}_t^i) - \boldsymbol{x}^i \right\| \leq 2\Gamma(N-1) \exp\left[ -\frac{\alpha_t^2 \sigma_\star^2 (1-\varepsilon) d}{2\sigma_t^2} \right],$$

for each $i$, which concludes the proof.

$\square$

Finally, we derive similar results for the partially memorizing denoiser. We recall that $\bar{\boldsymbol{x}}_{\mathrm{pmem},M}(t, \boldsymbol{x}_t^i)$ is given by

$$\bar{\boldsymbol{x}}_{\mathrm{pmem},M}(t, \boldsymbol{x}_t^i) = \sum_{j \in [\ell]} \mathrm{softmax}(\boldsymbol{w})_j \boldsymbol{x}^j,$$

where $\boldsymbol{w}_j = -\frac{1}{2\sigma_t^2} \|\alpha_t \boldsymbol{x}^j - \boldsymbol{x}_t^i\|^2$.

**Lemma E.6:** *Given vectors $(\boldsymbol{\mu}_\star^k)_{k=1}^K$ satisfying*

$$\min_{k \neq k'} \left\| \boldsymbol{\mu}_\star^k - \boldsymbol{\mu}_\star^{k'} \right\| \geq \gamma > 0,$$

*consider the distribution $\pi = (1/K) \sum_{k=1}^K \mathcal{N}(\boldsymbol{\mu}_\star^k, \sigma_\star^2 \boldsymbol{I})$. For $i \in [N]$, let $\boldsymbol{x}^i \sim \pi$, fix $0 \leq t \leq 1$, and let $\boldsymbol{x}_t^i \sim \alpha_t \boldsymbol{x}^i + \sigma_t \boldsymbol{g}$, where $\boldsymbol{g} \sim \mathcal{N}(\boldsymbol{0}, \boldsymbol{I})$ is independent from $\boldsymbol{x}^i$. Then for any $0 \leq \varepsilon \leq 1$ satisfying the coupling conditions*

$$\varepsilon \leq \frac{\alpha_t \gamma}{2\sqrt{d(2\alpha_t^2 \sigma_\star^2 + \sigma_t^2)}},$$

*and*

$$\frac{\varepsilon^2}{1-\varepsilon} \leq \frac{\alpha_t^2 \sigma_\star^2}{2\sigma_t^2}.$$

*We consider two cases, first assume that $i \in [\ell]$ then one has with probability at least $1 - 8N \exp[-d\varepsilon^2/8]$*

$$\left\| \bar{\boldsymbol{x}}_{\mathrm{pmem},M}(t, \boldsymbol{x}_t^i) - \boldsymbol{x}^i \right\| \leq 2N\Gamma_\star \exp\left[ -\frac{\alpha_t^2 \sigma_\star^2 (1-\varepsilon) d}{2\sigma_t^2} \right],$$

*with*

$$\Gamma_\star = \max_{k \in [K]} \sqrt{\|\boldsymbol{\mu}_\star^k\|^2 + \varepsilon \sigma_\star \sqrt{d} \|\boldsymbol{\mu}_\star^k\| + (1+\varepsilon)\sigma_\star^2 d}.$$

*Second, assume that $i \notin [\ell]$, with probability $1 - 8N \exp[-d\varepsilon^2/8] - K^{-\log(d)}(1 + \log(K)\log(d))$, $\mathsf{S}_i = \{j \in \ell, \, k_j = k_i\}$ is not empty and*

$$\left\| \bar{\boldsymbol{x}}_{\mathrm{pmem},M}(t, \boldsymbol{x}_t^i) - \sum_{j \in \mathsf{S}_i} \mathrm{softmax}(\boldsymbol{w}|_{\mathsf{S}_i})_j \boldsymbol{x}^j \right\| \leq \Gamma_\star (1+N)^2 \exp\left[ -\frac{\alpha_t^2 \gamma^2}{2\sigma_t^2} \right].$$

*Proof.* The first part of the proof where $i \in [\ell]$ is identical to Lemma E.5. We now assume that $i \notin [\ell]$. First, using Proposition C.4 we have that on an event with probability $1 - K^{-\log(d)}(1 + \log(K)\log(d))$, $\mathsf{S}_i$ is not empty. In addition, using a union bound we have that with probability at least $1 - 4N \exp[-d\varepsilon^2/8] - K^{-\log(d)}(1 + \log(K)\log(d))$, $\mathsf{S}_i = \{j \in \ell, \, k_j = k_i\}$ is not empty and

$$\min_{j \notin \mathsf{S}_i} \left\| \alpha_t \boldsymbol{x}^j - \boldsymbol{x} \right\|^2 - \left\| \alpha_t \boldsymbol{x}^i - \boldsymbol{x} \right\|^2 \geq \frac{\alpha_t^2 \gamma^2}{2},$$

Therefore, using Lemma D.2, we get that

$$\left\| \mathrm{softmax}(\boldsymbol{w}) - \mathrm{softmax}(\boldsymbol{w}|_{\mathsf{S}_i}) \right\|_p \leq (1 + |\mathsf{S}_i|)^{1/p} (|\mathsf{S}| - |\mathsf{S}_i|) \exp\left[ -\frac{\alpha_t^2 \gamma^2}{2\sigma_t^2} \right]$$

$$\leq (1+N)^{1/p} N \exp\left[ -\frac{\alpha_t^2 \gamma^2}{2\sigma_t^2} \right].$$

Using this result, we get

$$\left\|\bar{\boldsymbol{x}}_{\mathrm{pmem},M}(t,\boldsymbol{x}_t^i) - \sum_{j\in\mathsf{S}_i}\mathrm{softmax}(\boldsymbol{w}|_{\mathsf{S}_i})_j\boldsymbol{x}^j\right\| = \|\boldsymbol{M}(\mathrm{softmax}(\boldsymbol{w}|_{\mathsf{S}_i}) - \mathrm{softmax}(\boldsymbol{w}))\|$$

$$\leq \Gamma\|\mathrm{softmax}(\boldsymbol{w}|_{\mathsf{S}_i}) - \mathrm{softmax}(\boldsymbol{w})\|_1$$

$$\leq (N+1)^2\Gamma\exp\left[-\frac{\alpha_t^2\gamma^2}{2\sigma_t^2}\right],$$

where we have defined

$$\boldsymbol{M} = \begin{bmatrix}\boldsymbol{x}^1 & \ldots & \boldsymbol{x}^\ell\end{bmatrix} \in \mathbb{R}^{d\times\ell}, \quad \Gamma = \max_{j\in[N]}\|\boldsymbol{x}^j\|.$$

Finally, using Lemma C.2, we have that for any $0 \leq \varepsilon \leq 1$

$$\mathbb{P}\left[\left|\|\boldsymbol{x}^j\|^2 - (\|\boldsymbol{\mu}_\star^{k_j}\|^2 + \sigma_\star^2 d)\right| \geq \varepsilon\sigma_\star\sqrt{d}(\sigma_\star\sqrt{d} + \|\boldsymbol{\mu}_\star^{k_j}\|)\right] \leq 3\exp[-d\varepsilon^2/8],$$

so with probability at least $1 - 4N\exp[-d\varepsilon^2/8]$ we have

$$\Gamma \leq \Gamma_\star = \max_{k\in[K]}\sqrt{\|\boldsymbol{\mu}_\star^k\|^2 + \varepsilon\sigma_\star\sqrt{d}\|\boldsymbol{\mu}_\star^k\| + (1+\varepsilon)\sigma_\star^2 d}.$$

By a union bound, we conclude that with probability at least $1 - K^{-\log(d)}(1 + \log(K)\log(d)) - 8N\exp[-d\varepsilon^2/8]$ that

$$\left\|\bar{\boldsymbol{x}}_{\mathrm{pmem},M}(t,\boldsymbol{x}_t^i) - \sum_{j\in\mathsf{S}_i}\mathrm{softmax}(\boldsymbol{w}|_{\mathsf{S}_i})_j\boldsymbol{x}^j\right\| \leq \Gamma_\star(1+N)^2\exp\left[-\frac{\alpha_t^2\gamma^2}{2\sigma_t^2}\right],$$

which concludes the proof.

$\square$

## F   Training Loss Approximations

In this section, we give proofs of results that imply our main results stated in the main body, namely Theorem 3.1 and Theorem 3.2. In the proofs, we will use a convenient shorthand for the training loss defined in (6) when specialized to the Gaussian mixture model denoisers of interest. Namely, since in this case the loss is effectively a function of the learnable mean parameters and the shared learnable variance parameter in the model, we will write

$$\mathbb{E}\left[\mathcal{L}_N(\boldsymbol{\mu}_\star^1,\ldots,\boldsymbol{\mu}_\star^K,\sigma_\star^2)\right]$$

to denote the loss of the generalizing denoiser $\mathcal{L}_{N,t}(\bar{\boldsymbol{x}}_\star)$,

$$\mathbb{E}\left[\mathcal{L}_N(\boldsymbol{x}^1,\ldots,\boldsymbol{x}^N,0)\right]$$

to denote the loss of the memorizing denoiser $\mathcal{L}_{N,t}(\bar{\boldsymbol{x}}_{\mathrm{mem}})$, and

$$\mathbb{E}\left[\mathcal{L}_N(\boldsymbol{x}^1,\ldots,\boldsymbol{x}^\ell,0)\right]$$

to denote the loss of the partially memorizing denoiser $\mathcal{L}_{N,t}(\bar{\boldsymbol{x}}_{\mathrm{pmem},M})$.

**Theorem F.1:** *Given vectors* $(\boldsymbol{\mu}_\star^k)_{k=1}^K$ *satisfying*

$$\min_{k \neq k'} \left\| \boldsymbol{\mu}_\star^k - \boldsymbol{\mu}_\star^{k'} \right\| \geq \gamma > 0,$$

*consider the distribution* $\pi = (1/K)\sum_{k=1}^K \mathcal{N}(\boldsymbol{\mu}_\star^k, \sigma_\star^2 \boldsymbol{I})$. *For* $i \in [N]$, *let* $\boldsymbol{x}^i \sim \pi$, *fix* $0 \leq t \leq 1$, *and let* $\boldsymbol{x}_t^i \sim \alpha_t \boldsymbol{x}^i + \sigma_t \boldsymbol{g}$, *where* $\boldsymbol{g} \sim \mathcal{N}(\boldsymbol{0}, \boldsymbol{I})$ *is independent from* $\boldsymbol{x}^i$. *Consider the denoiser* $\bar{\boldsymbol{x}}_\star(t, \boldsymbol{x}_t)$ *associated to the true distribution* $\pi_\star$. *Then for any* $0 \leq \varepsilon \leq 1$ *satisfying the coupling conditions*

$$\varepsilon \leq \frac{\alpha_t \gamma}{2\sqrt{d(\alpha_t^2 \sigma_\star^2 + \sigma_t^2)}},$$

*and*

$$\frac{\varepsilon^2}{1 - \varepsilon} \leq \frac{\alpha_t^2 \sigma_\star^2}{2\sigma_t^2},$$

$$\left| \left[ \mathbb{E}\left[ \mathcal{L}_N(\boldsymbol{\mu}_\star^1, \ldots, \boldsymbol{\mu}_\star^K, \sigma_\star^2) \right] - \mathbb{E}\left[ \mathcal{L}_N(\boldsymbol{x}^1, \ldots, \boldsymbol{x}^N, 0) \right] \right] - \frac{d\sigma_\star^2 \sigma_t^2}{\alpha_t^2 \sigma_\star^2 + \sigma_t^2} \right| \leq \Xi(\varepsilon, t, d),$$

*where the residual* $\Xi$ *is explicit in the proof.*

We denote $\psi_t = \alpha_t^2 / \sigma_t^2$ and $\varsigma_t$ its inverse function.

*Proof.* The proof will be an application of Lemma E.4 and Lemma E.5.

**Nominal value decomposition.** First, we define for any $i \in [N]$

$$\bar{\boldsymbol{x}}^{\mathrm{nom}}(t, \boldsymbol{x}_t^i) = \frac{\alpha_t \sigma_\star^2}{\alpha_t^2 \sigma_\star^2 + \sigma_t^2} \boldsymbol{x}_t^i + \frac{\sigma_t^2}{\alpha_t^2 \sigma_\star^2 + \sigma_t^2} \boldsymbol{\mu}_\star^{k_i}.$$

This nominal value $\bar{\boldsymbol{x}}^{\mathrm{nom}}(t, \boldsymbol{x}_t^i)$ will be crucial for the rest of our analysis. Using (13), we have that

$$\left| \mathbb{E}\left[ \mathcal{L}_N(\boldsymbol{\mu}_\star^1, \ldots, \boldsymbol{\mu}_\star^K, \sigma_\star^2) \right] - \mathbb{E}\left[ \mathcal{L}_N(\boldsymbol{x}^1, \ldots, \boldsymbol{x}^N, 0) \right] - \frac{d\sigma_\star^2 \sigma_t^2}{\alpha_t^2 \sigma_\star^2 + \sigma_t^2} \right|$$

$$= \frac{1}{N} \sum_{i=1}^N \left| \mathbb{E}\left[ \left\| \bar{\boldsymbol{x}}_\star(t, \boldsymbol{x}_t^i) - \bar{\boldsymbol{x}}_{\mathrm{mem}}(t, \boldsymbol{x}_t^i) \right\|^2 \right] - \frac{d\sigma_\star^2 \sigma_t^2}{\alpha_t^2 \sigma_\star^2 + \sigma_t^2} \right|$$

$$= \frac{1}{N} \sum_{i=1}^N \left| \mathbb{E}\left[ \left\| \bar{\boldsymbol{x}}_\star(t, \boldsymbol{x}_t^i) - \bar{\boldsymbol{x}}^{\mathrm{nom}}(t, \boldsymbol{x}_t^i) + \bar{\boldsymbol{x}}^{\mathrm{nom}}(t, \boldsymbol{x}_t^i) - \boldsymbol{x}_i + \boldsymbol{x}_i - \bar{\boldsymbol{x}}_{\mathrm{mem}}(t, \boldsymbol{x}_t^i) \right\|^2 \right] - \frac{d\sigma_\star^2 \sigma_t^2}{\alpha_t^2 \sigma_\star^2 + \sigma_t^2} \right|.$$

In what follows, for simplicity, we denote $\Lambda_t^i$

$$\Lambda_t^i = \bar{\boldsymbol{x}}_\star(t, \boldsymbol{x}_t^i) - \bar{\boldsymbol{x}}^{\mathrm{nom}}(t, \boldsymbol{x}_t^i) + \boldsymbol{x}_i - \bar{\boldsymbol{x}}_{\mathrm{mem}}(t, \boldsymbol{x}_t^i).$$

We have that

$$\left| \mathbb{E}\big[\mathcal{L}_N(\boldsymbol{\mu}_\star^1,\ldots,\boldsymbol{\mu}_\star^K,\sigma_\star^2)\big] - \mathbb{E}\big[\mathcal{L}_N(\boldsymbol{x}^1,\ldots,\boldsymbol{x}^N,0)\big] - \frac{d\sigma_\star^2\sigma_t^2}{\alpha_t^2\sigma_\star^2 + \sigma_t^2} \right| \tag{28}$$

$$= \frac{1}{N}\sum_{i=1}^N \left| \mathbb{E}\Big[\big\|\Lambda_t^i + \bar{\boldsymbol{x}}^{\mathrm{nom}}(t,\boldsymbol{x}_t^i) - \boldsymbol{x}_i\big\|^2\Big] - \frac{d\sigma_\star^2\sigma_t^2}{\alpha_t^2\sigma_\star^2 + \sigma_t^2} \right|$$

$$\leq \frac{1}{N}\sum_{i=1}^N \left| \mathbb{E}\Big[\big\|\bar{\boldsymbol{x}}^{\mathrm{nom}}(t,\boldsymbol{x}_t^i) - \boldsymbol{x}_i\big\|^2\Big] - \frac{d\sigma_\star^2\sigma_t^2}{\alpha_t^2\sigma_\star^2 + \sigma_t^2} \right|$$

$$+ \frac{1}{N}\sum_{i=1}^N \mathbb{E}\Big[\big\|\Lambda_t^i\big\|^2\Big] + \frac{2}{N}\sum_{i=1}^N \mathbb{E}\Big[\big\|\Lambda_t^i\big\|^2\Big]^{1/2}\mathbb{E}\Big[\big\|\bar{\boldsymbol{x}}^{\mathrm{nom}}(t,\boldsymbol{x}_t^i) - \boldsymbol{x}_i\big\|^2\Big]^{1/2}$$

$$\leq \frac{1}{N}\sum_{i=1}^N \left| \mathbb{E}\Big[\big\|\bar{\boldsymbol{x}}^{\mathrm{nom}}(t,\boldsymbol{x}_t^i) - \boldsymbol{x}_i\big\|^2\Big] - \frac{d\sigma_\star^2\sigma_t^2}{\alpha_t^2\sigma_\star^2 + \sigma_t^2} \right|$$

$$+ \frac{2}{N}\sum_{i=1}^N \Big\{ \mathbb{E}\Big[\big\|\bar{\boldsymbol{x}}_\star(t,\boldsymbol{x}_t^i) - \bar{\boldsymbol{x}}^{\mathrm{nom}}(t,\boldsymbol{x}_t^i)\big\|^2\Big] + \mathbb{E}\Big[\big\|\boldsymbol{x}_i - \bar{\boldsymbol{x}}_{\mathrm{mem}}(t,\boldsymbol{x}_t^i)\big\|^2\Big] \Big\} +$$

$$\frac{4}{N}\sum_{i=1}^N \Big\{ \mathbb{E}\Big[\big\|\bar{\boldsymbol{x}}_\star(t,\boldsymbol{x}_t^i) - \bar{\boldsymbol{x}}^{\mathrm{nom}}(t,\boldsymbol{x}_t^i)\big\|^2\Big] + \mathbb{E}\Big[\big\|\boldsymbol{x}_i - \bar{\boldsymbol{x}}_{\mathrm{mem}}(t,\boldsymbol{x}_t^i)\big\|^2\Big] \Big\}^{1/2}$$

$$\times \mathbb{E}\Big[\big\|\bar{\boldsymbol{x}}^{\mathrm{nom}}(t,\boldsymbol{x}_t^i) - \boldsymbol{x}_i\big\|^2\Big]^{1/2}.$$

In the rest of the proof, we control each term in (28).

**Control of generalizing denoiser.** First, we are going to control $\mathbb{E}\Big[\big\|\bar{\boldsymbol{x}}_\star(t,\boldsymbol{x}_t^i) - \bar{\boldsymbol{x}}^{\mathrm{nom}}(t,\boldsymbol{x}_t^i)\big\|^2\Big]$.
First, we recall that by (11)

$$\bar{\boldsymbol{x}}_\star(t,\boldsymbol{x}_t^i) = \frac{\alpha_t\sigma_\star^2}{\alpha_t^2\sigma_\star^2 + \sigma_t^2}\boldsymbol{x}_t^i + \frac{\sigma_t^2}{\alpha_t^2\sigma_\star^2 + \sigma_t^2}\sum_{i=1}^M \boldsymbol{\mu}^i\,\mathrm{softmax}(\boldsymbol{w})_i,$$

In what follows, we denote

$$\Gamma = \max_{i\in[K]}\|\mu_\star^i\|.$$

In addition, we have that

$$\boldsymbol{x}_t^i = \alpha_t\boldsymbol{\mu}_\star^{k_i} + \alpha_t\sigma_\star\boldsymbol{w} + \sigma_t\boldsymbol{g} \overset{d}{=} \alpha_t\boldsymbol{\mu}_\star^{k_i} + (\alpha_t^2\sigma_\star^2 + \sigma_t^2)^{1/2}\boldsymbol{g}.$$

where $k_i$ is the index of the mean corresponding to $\boldsymbol{x}^i$. Finally, we have that for any $p\in\mathbb{N}$

$$\mathbb{E}[\|\boldsymbol{g}\|^p] \leq \mathbb{E}\big[\|\boldsymbol{g}\|^{2p}\big]^{1/2} \leq 2^{p/2}(\Gamma(d/2+p)/\Gamma(d/2))^{1/2} \leq 2^{p/2}(d/2+p)^{p/2}.$$

Combining those results, we get that for any $p\in\mathbb{N}$

$$\mathbb{E}\Big[\big\|\boldsymbol{x}_t^i\big\|^p\Big] \leq 2^{2p}\left(\Gamma^p + (\alpha_t^2\sigma_\star^2 + \sigma_t^2)^{p/2}(d/2+p)^{p/2}\right).$$

Similarly, we have that for any $p\in\mathbb{N}$

$$\mathbb{E}\Big[\big\|\bar{\boldsymbol{x}}^{\mathrm{nom}}(t,\boldsymbol{x}_t^i)\big\|^p\Big] \leq 2^{2p}\left(\Gamma^p + (\alpha_t^2\sigma_\star^2 + \sigma_t^2)^{p/2}(d/2+p)^{p/2}\right).$$

For any event A such that $\big\|\bar{\boldsymbol{x}}_\star(t,\boldsymbol{x}_t^i) - \bar{\boldsymbol{x}}^{\mathrm{nom}}(t,\boldsymbol{x}_t^i)\big\|^2 \leq C$, we have that

$$\mathbb{E}\Big[\big\|\bar{\boldsymbol{x}}_\star(t,\boldsymbol{x}_t^i) - \bar{\boldsymbol{x}}^{\mathrm{nom}}(t,\boldsymbol{x}_t^i)\big\|^2\Big] \leq C\mathbb{P}[\mathsf{A}] + 2\mathbb{P}[\mathsf{A}^{\mathrm{c}}]^{1/2}\left(\mathbb{E}\Big[\big\|\boldsymbol{x}_t^i\big\|^4\Big]^{1/2} + \mathbb{E}\Big[\big\|\bar{\boldsymbol{x}}^{\mathrm{nom}}(t,\boldsymbol{x}_t^i)\big\|^4\Big]^{1/2}\right)$$

$$\leq C + 16\left(\Gamma^2 + (\alpha_t^2\sigma_\star^2 + \sigma_t^2)(d/2+4)\right)\mathbb{P}[\mathsf{A}^{\mathrm{c}}]^{1/2}.$$

Now, combining this result and Lemma E.4 we get that

$$
\mathbb{E}\left[\left\|\bar{\boldsymbol{x}}_\star(t, \boldsymbol{x}_t^i) - \bar{\boldsymbol{x}}^{\mathrm{nom}}(t, \boldsymbol{x}_t^i)\right\|^2\right]
$$

$$
\leq \left(\frac{2K\sigma_t^2 \max_{k\in[K]}\|\boldsymbol{\mu}_\star^k\|}{\alpha_t^2\sigma_\star^2 + \sigma_t^2}\right)^2 \exp\left[-\frac{\gamma^2\alpha_t^2}{2(\alpha_t^2\sigma_\star^2 + \sigma_t^2)}\right]
$$

$$
+ 64\left(\max_{k\in[K]}\|\boldsymbol{\mu}_\star^k\|^2 + (\alpha_t^2\sigma_\star^2 + \sigma_t^2)(d/2+4)\right)K\exp[-d\varepsilon^2/16].
$$

Therefore, there exists a numerical constant $C_0 \geq 0$ such that

$$
\mathbb{E}\left[\left\|\bar{\boldsymbol{x}}_\star(t, \boldsymbol{x}_t^i) - \bar{\boldsymbol{x}}^{\mathrm{nom}}(t, \boldsymbol{x}_t^i)\right\|^2\right]
$$

$$
\leq C_0\left(\max_{k\in[K]}\|\boldsymbol{\mu}_\star^k\|^2 + (1+\sigma_\star^2)d\right)K^2\left(\exp\left[-\frac{\gamma^2\alpha_t^2}{2(\alpha_t^2\sigma_\star^2 + \sigma_t^2)}\right] + \exp[-d\varepsilon^2/16]\right).
$$

**Control of memorizing denoiser.** Second, we are going to control $\mathbb{E}\left[\left\|\boldsymbol{x}_i - \bar{\boldsymbol{x}}_{\mathrm{mem}}(t, \boldsymbol{x}_t^i)\right\|^2\right]$. The proof is similar to the control of $\mathbb{E}\left[\left\|\bar{\boldsymbol{x}}_\star(t, \boldsymbol{x}_t^i) - \bar{\boldsymbol{x}}^{\mathrm{nom}}(t, \boldsymbol{x}_t^i)\right\|^2\right]$. We recall that we have

$$
\bar{\boldsymbol{x}}_{\mathrm{mem}}(t, \boldsymbol{x}_t) = \sum_{i=1}^N \boldsymbol{x}^i \operatorname{softmax}(\boldsymbol{w})_i,
$$

where

$$
w_i(\boldsymbol{x}_t) = -\frac{1}{2}\|\alpha_t\boldsymbol{x}^i - \boldsymbol{x}_t^i\|^2/\sigma_t^2
$$

Similarly as before, we have for any $p \in \mathbb{N}$

$$
\mathbb{E}\left[\max_{i\in[N]}\|\boldsymbol{x}_i\|^p\right] \leq 2^{2p}N(\Gamma^p + (d/2+p)^{p/2}).
$$

Therefore, we get that

$$
\mathbb{E}[\|\boldsymbol{x}_i\|^p] \leq 2^{2p}N(\Gamma^p + \sigma_\star^p(d/2+p)^{p/2}), \qquad \mathbb{E}\left[\|\bar{\boldsymbol{x}}_{\mathrm{mem}}(t, \boldsymbol{x}_t^i)\|^p\right] \leq 2^{2p}N(\Gamma^p + \sigma_\star^p(d/2+p)^{p/2}).
$$

Note that this upper-bound is rather loose but given the rate of growth of $N$ with respect to $d$ that we will consider we won't need a tighter bound. For any event A such that $\left\|\bar{\boldsymbol{x}}_{\mathrm{mem}}(t, \boldsymbol{x}_t^i) - \boldsymbol{x}^i\right\|^2 \leq C$, we have that

$$
\mathbb{E}\left[\left\|\bar{\boldsymbol{x}}_{\mathrm{mem}}(t, \boldsymbol{x}_t^i) - \boldsymbol{x}^i\right\|^2\right] \leq C\mathbb{P}[\mathsf{A}] + 2\mathbb{P}[\mathsf{A}^{\mathrm{c}}]^{1/2}\left(\mathbb{E}\left[\|\boldsymbol{x}_t^i\|^4\right]^{1/2} + \mathbb{E}\left[\|\bar{\boldsymbol{x}}^{\mathrm{nom}}(t, \boldsymbol{x}_t^i)\|^4\right]^{1/2}\right)
$$

$$
\leq C + 16\left(\Gamma^2 + \sigma_\star^2(d/2+4)\right)\mathbb{P}[\mathsf{A}^{\mathrm{c}}]^{1/2}.
$$

Therefore, combining this result with Lemma E.5, there exists a numerical constant $C_1 \geq 0$ such that

$$
\mathbb{E}\left[\left\|\bar{\boldsymbol{x}}_{\mathrm{mem}}(t, \boldsymbol{x}_t^i) - \boldsymbol{x}^i\right\|^2\right]
$$

$$
\leq C_1\left(\max_{k\in[K]}\|\boldsymbol{\mu}_\star^k\|^2 + (1+\sigma_\star^2)d\right)N^2\left(\exp\left[-\frac{\alpha_t^2\sigma_\star^2(1-\varepsilon)d}{2\sigma_t^2}\right] + \exp[-d\varepsilon^2/16]\right).
$$

**Control of the nominal expectation.** For the nominal error $\|\bar{\boldsymbol{x}}^{\mathrm{nom}}(t, \boldsymbol{x}_t^i) - \boldsymbol{x}^i\|$, we have by the nominal denoiser's definition in Lemma E.4 and the distributional assumption $\boldsymbol{x}^i \sim_{\mathrm{i.i.d.}} \pi$

$$
\left\|\bar{\boldsymbol{x}}^{\mathrm{nom}}(t, \boldsymbol{x}_t^i) - \boldsymbol{x}^i\right\|^2 = \left\|\frac{\alpha_t\sigma_\star^2}{\alpha_t^2\sigma_\star^2 + \sigma_t^2}\boldsymbol{x}_t^i + \frac{\sigma_t^2}{\alpha_t^2\sigma_\star^2 + \sigma_t^2}\boldsymbol{\mu}_\star^{k_i} - \boldsymbol{x}^i\right\|^2
$$

$$
= \left\|\frac{\alpha_t^2\sigma_\star^2}{\alpha_t^2\sigma_\star^2 + \sigma_t^2}\sigma_\star\boldsymbol{w}^i + \frac{\alpha_t\sigma_\star^2}{\alpha_t^2\sigma_\star^2 + \sigma_t^2}\sigma_t\boldsymbol{g}^i - \sigma_\star\boldsymbol{w}^i\right\|^2,
$$

where $\boldsymbol{w}^i \sim \mathcal{N}(\mathbf{0}, \boldsymbol{I})$ is independent of $\boldsymbol{g}^i \sim \mathcal{N}(\mathbf{0}, \boldsymbol{I})$. In particular, this is equal in distribution to the squared $\ell^2$ norm of a Gaussian random variable, which has zero mean and isotropic diagonal covariance with diagonal elements

$$\sigma_\star^2 \left( 1 - \frac{\alpha_t^2 \sigma_\star^2}{\alpha_t^2 \sigma_\star^2 + \sigma_t^2} \right)^2 + \sigma_t^2 \left( \frac{\alpha_t \sigma_\star^2}{\alpha_t^2 \sigma_\star^2 + \sigma_t^2} \right)^2 = \frac{\sigma_\star^2 \sigma_t^2}{\alpha_t^2 \sigma_\star^2 + \sigma_t^2}.$$

An application of Theorem C.1 then gives that with probability at least $1 - 2 \exp[-d\varepsilon^2/8]$

$$\left| \left\| \bar{\boldsymbol{x}}^{\mathrm{nom}}(t, \boldsymbol{x}_t^i) - \boldsymbol{x}^i \right\|^2 - \frac{d \sigma_\star^2 \sigma_t^2}{\alpha_t^2 \sigma_\star^2 + \sigma_t^2} \right| \le \varepsilon \frac{d \sigma_\star^2 \sigma_t^2}{\alpha_t^2 \sigma_\star^2 + \sigma_t^2}$$

For any event A such that

$$\left| \left\| \bar{\boldsymbol{x}}^{\mathrm{nom}}(t, \boldsymbol{x}_t^i) - \boldsymbol{x}^i \right\|^2 - \frac{d \sigma_\star^2 \sigma_t^2}{\alpha_t^2 \sigma_\star^2 + \sigma_t^2} \right| \le C,$$

on that event, we have that

$$\left| \mathbb{E}\left[ \left\| \bar{\boldsymbol{x}}^{\mathrm{nom}}(t, \boldsymbol{x}_t^i) - \boldsymbol{x}^i \right\|^2 \right] - \frac{d \sigma_\star^2 \sigma_t^2}{\alpha_t^2 \sigma_\star^2 + \sigma_t^2} \right| \le \mathbb{E}\left[ \left| \left\| \bar{\boldsymbol{x}}^{\mathrm{nom}}(t, \boldsymbol{x}_t^i) - \boldsymbol{x}^i \right\|^2 - \frac{d \sigma_\star^2 \sigma_t^2}{\alpha_t^2 \sigma_\star^2 + \sigma_t^2} \right| \right]$$

$$\le C + \mathbb{P}[\mathrm{A}^c]^{1/2} \left( \mathbb{E}\left[ \left\| \boldsymbol{x}^i \right\|^4 \right]^{1/2} + \mathbb{E}\left[ \left\| \bar{\boldsymbol{x}}^{\mathrm{nom}}(t, \boldsymbol{x}_t^i) \right\|^4 \right]^{1/2} \right)$$

Therefore, there exists $C_2 \ge 0$, a numerical constant, such that

$$\left| \mathbb{E}\left[ \left\| \bar{\boldsymbol{x}}^{\mathrm{nom}}(t, \boldsymbol{x}_t^i) - \boldsymbol{x}^i \right\|^2 \right] - \frac{d \sigma_\star^2 \sigma_t^2}{\alpha_t^2 \sigma_\star^2 + \sigma_t^2} \right|$$

$$\le \varepsilon \frac{d \sigma_\star^2 \sigma_t^2}{\alpha_t^2 \sigma_\star^2 + \sigma_t^2} + C_2 \left( \max_{k \in [K]} \| \boldsymbol{\mu}_\star^k \|^2 + (1 + \sigma_\star^2) d \right) \exp[-\varepsilon^2 d/16].$$

Therefore, there exists $C_3 \ge 0$, a numerical constant, such that

$$\left| \mathbb{E}\left[ \mathcal{L}_N(\boldsymbol{\mu}_\star^1, \dots, \boldsymbol{\mu}_\star^K, \sigma_\star^2) \right] - \mathbb{E}\left[ \mathcal{L}_N(\boldsymbol{x}^1, \dots, \boldsymbol{x}^N, 0) \right] - \frac{d \sigma_\star^2 \sigma_t^2}{\alpha_t^2 \sigma_\star^2 + \sigma_t^2} \right|$$

$$\le \varepsilon \frac{d \sigma_\star^2 \sigma_t^2}{\alpha_t^2 \sigma_\star^2 + \sigma_t^2} + C_3 \left( \max_{k \in [K]} \| \boldsymbol{\mu}_\star^k \|^2 + (1 + \sigma_\star^2) d \right) (K^2 + N^2)$$

$$\times \left\{ \exp\left[ -\frac{\gamma^2 \alpha_t^2}{2(\alpha_t^2 \sigma_\star^2 + \sigma_t^2)} \right] + \exp\left[ -\frac{\alpha_t^2 \sigma_\star^2 (1 - \varepsilon) d}{2 \sigma_t^2} \right] + \exp[-\varepsilon^2 d/16] \right\}.$$

Let us further simplify the bound. We have that

$$\left| \mathbb{E}\left[ \mathcal{L}_N(\boldsymbol{\mu}_\star^1, \dots, \boldsymbol{\mu}_\star^K, \sigma_\star^2) \right] - \mathbb{E}\left[ \mathcal{L}_N(\boldsymbol{x}^1, \dots, \boldsymbol{x}^N, 0) \right] - \frac{d \sigma_\star^2 \sigma_t^2}{\alpha_t^2 \sigma_\star^2 + \sigma_t^2} \right|$$

$$\le \varepsilon \frac{d \sigma_\star^2 \sigma_t^2}{\alpha_t^2 \sigma_\star^2 + \sigma_t^2} + C_3 \left( 1 + \max_{k \in [K]} \| \boldsymbol{\mu}_\star^k \|^2 / d + \sigma_\star^2 \right) \mathrm{Poly}(d) \frac{d \sigma_\star^2 \sigma_t^2}{\alpha_t^2 \sigma_\star^2 + \sigma_t^2} \left( 1/\sigma_\star^2 + \psi_t \right)$$

$$\times \left\{ \exp\left[ -\frac{\gamma^2 \alpha_t^2}{2(\alpha_t^2 \sigma_\star^2 + \sigma_t^2)} \right] + \exp\left[ -\frac{\alpha_t^2 \sigma_\star^2 (1 - \varepsilon) d}{2 \sigma_t^2} \right] + \exp[-\varepsilon^2 d/16] \right\},$$

where we have used that

$$\frac{d \sigma_\star^2 \sigma_t^2}{\alpha_t^2 \sigma_\star^2 + \sigma_t^2} \left( 1/\sigma_\star^2 + \psi_t \right) \ge 1,$$

with $\psi_t = \alpha_t^2 / \sigma_t^2$.

$\square$

**Proposition F.2:** *Assume that* $N = \mathrm{poly}(d)$, $\gamma^2 = \Theta(d)$, $\max_{k \in [K]} \| \boldsymbol{\mu}_\star^k \|^2 = \Theta(d)$ *and* $\sigma_\star^2 = \Theta(1)$. *Let* $\kappa(d) = \varsigma(\Theta(\log(d)^2/d))$. *We have that uniformly on* $t \in [0, \kappa(d)]$

$$\mathbb{E}\left[ \mathcal{L}_N(\boldsymbol{\mu}_\star^1, \dots, \boldsymbol{\mu}_\star^K, \sigma_\star^2) \right] - \mathbb{E}\left[ \mathcal{L}_N(\boldsymbol{x}^1, \dots, \boldsymbol{x}^N, 0) \right] = \Theta\left( \frac{d \sigma_\star^2 \sigma_t^2}{\alpha_t^2 \sigma_\star^2 + \sigma_t^2} \right).$$

*The claim will then follow by an appropriate choice of constant in the definition of* $\kappa(d)$, *which will make the residuals in the previous expression of the correct order of magnitude to be nontrivial, and guarantee the approximation for* $t$ *within the claimed interval.*

*Proof.* First note that $\varepsilon = \Theta(\log(d)/d^{1/2})$ satisfies the coupling condition. Next, if we assume that $d$ is large enough so that $1 - \varepsilon \le 1/2$. We have

$$\left| \mathbb{E}\big[\mathcal{L}_N(\boldsymbol{\mu}_\star^1, \ldots, \boldsymbol{\mu}_\star^K, \sigma_\star^2)\big] - \mathbb{E}\big[\mathcal{L}_N(\boldsymbol{x}^1, \ldots, \boldsymbol{x}^N, 0)\big] - \frac{d\sigma_\star^2 \sigma_t^2}{\alpha_t^2 \sigma_\star^2 + \sigma_t^2} \right|$$

$$\le \varepsilon \frac{d\sigma_\star^2 \sigma_t^2}{\alpha_t^2 \sigma_\star^2 + \sigma_t^2} + C_4 \left( 1 + \max_{k \in [K]} \|\boldsymbol{\mu}_\star^k\|^2/d + \sigma_\star^2 \right) \text{Poly}(d) \frac{d\sigma_\star^2 \sigma_t^2}{\alpha_t^2 \sigma_\star^2 + \sigma_t^2} \left( 1/\sigma_\star^2 + \psi_t \right)$$

$$\times \exp \left[ -\frac{1}{16} \min \left( \log(d)^2, \psi_t \sigma_\star^2 d, \frac{\gamma^2 \psi_t}{1 + \sigma_\star^2 \psi_t} \right) \right].$$

$\square$

**Theorem F.3:** *Given vectors* $(\boldsymbol{\mu}_\star^k)_{k=1}^K$ *satisfying*

$$\min_{k \neq k'} \left\| \boldsymbol{\mu}_\star^k - \boldsymbol{\mu}_\star^{k'} \right\| \ge \gamma > 0,$$

*consider the distribution* $\pi = (1/K) \sum_{k=1}^K \mathcal{N}(\boldsymbol{\mu}_\star^k, \sigma_\star^2 \boldsymbol{I})$. *For* $i \in [N]$, *let* $\boldsymbol{x}^i \sim \pi$, *fix* $0 \le t \le 1$, *and let* $\boldsymbol{x}_t^i \sim \alpha_t \boldsymbol{x}^i + \sigma_t \boldsymbol{g}$, *where* $\boldsymbol{g} \sim \mathcal{N}(\boldsymbol{0}, \boldsymbol{I})$ *is independent from* $\boldsymbol{x}^i$. *Consider the partial memorizing denoiser* $\bar{\boldsymbol{x}}_{\text{pmem},M}(t, \boldsymbol{x}_t)$. *Then for any* $0 \le \varepsilon \le 1$ *satisfying the coupling conditions*

$$\varepsilon \le \frac{\alpha_t \gamma}{2\sqrt{d(\alpha_t^2 \sigma_\star^2 + \sigma_t^2)}},$$

*and*

$$\frac{\varepsilon^2}{1 - \varepsilon} \le \frac{\alpha_t^2 \sigma_\star^2}{2\sigma_t^2},$$

*we have that*

$$\left( 1 - \frac{\ell}{N} \right) d\sigma_\star^2 - \Xi(\varepsilon, t, d)$$

$$\le \mathbb{E}\big[\mathcal{L}_N(\boldsymbol{x}^1, \ldots, \boldsymbol{x}^\ell, 0)\big] - \mathbb{E}\big[\mathcal{L}_N(\boldsymbol{x}^1, \ldots, \boldsymbol{x}^N, 0)\big]$$

$$\le 2 \left( 1 - \frac{\ell}{N} \right) d\sigma_\star^2 + \Xi(\varepsilon, t, d),$$

*where the residual* $\Xi$ *is explicit in the proof.*

*Proof.* The proof will be an application of Lemma E.6 and Lemma E.5.

**Softmax value decomposition.** Using (13), we have that

$$\mathbb{E}\big[\mathcal{L}_N(\boldsymbol{x}^1, \ldots, \boldsymbol{x}^\ell, 0)\big] - \mathbb{E}\big[\mathcal{L}_N(\boldsymbol{x}^1, \ldots, \boldsymbol{x}^N, 0)\big]$$

$$= \frac{1}{N} \sum_{i=1}^N \mathbb{E}\Big[\big\| \bar{\boldsymbol{x}}_{\text{pmem},M}(t, \boldsymbol{x}_t^i) - \bar{\boldsymbol{x}}_{\text{mem}}(t, \boldsymbol{x}_t^i) \big\|^2\Big]$$

$$= \frac{1}{N} \sum_{i=1}^N \mathbb{E}\Big[\big\| \bar{\boldsymbol{x}}_{\text{pmem},M}(t, \boldsymbol{x}_t^i) - \boldsymbol{x}_i + \boldsymbol{x}_i - \bar{\boldsymbol{x}}_{\text{mem}}(t, \boldsymbol{x}_t^i) \big\|^2\Big].$$

We have that

$$\mathbb{E}\big[\mathcal{L}_N(\boldsymbol{x}^1, \ldots, \boldsymbol{x}^\ell, 0)\big] - \mathbb{E}\big[\mathcal{L}_N(\boldsymbol{x}^1, \ldots, \boldsymbol{x}^N, 0)\big]$$

$$= \frac{1}{N} \sum_{i=1}^N \mathbb{E}\Big[\big\| \bar{\boldsymbol{x}}_{\text{pmem},M}(t, \boldsymbol{x}_t^i) - \boldsymbol{x}_i + \boldsymbol{x}_i - \bar{\boldsymbol{x}}_{\text{mem}}(t, \boldsymbol{x}_t^i) \big\|^2\Big]$$

$$= \frac{1}{N} \sum_{i=1}^N \mathbb{E}\Big[\big\| \bar{\boldsymbol{x}}_{\text{pmem},M}(t, \boldsymbol{x}_t^i) - \boldsymbol{x}^i \big\|^2\Big] + \mathbb{E}\Big[\big\| \bar{\boldsymbol{x}}_{\text{mem}}(t, \boldsymbol{x}_t^i) - \boldsymbol{x}^i \big\|^2\Big] + 2\Lambda_t^i,$$

where
$$|\Lambda_t^i| \le \mathbb{E}\Big[\big\|\bar{\boldsymbol{x}}_{\mathrm{mem}}(t, \boldsymbol{x}_t^i) - \boldsymbol{x}^i\big\|^2\Big]^{1/2} \mathbb{E}\Big[\big\|\bar{\boldsymbol{x}}_{\mathrm{pmem},M}(t, \boldsymbol{x}_t^i) - \boldsymbol{x}^i\big\|^2\Big]^{1/2}.$$

In the rest of the proof, we control each term in (28).

**Control of memorizing denoiser.** Second, we are going to control $\mathbb{E}\Big[\big\|\boldsymbol{x}_i - \bar{\boldsymbol{x}}_{\mathrm{mem}}(t, \boldsymbol{x}_t^i)\big\|^2\Big]$. The proof is similar to the one of Theorem F.1 but we reproduce it for completeness. We recall that we have
$$\bar{\boldsymbol{x}}_{\mathrm{mem}}(t, \boldsymbol{x}_t) = \sum_{i=1}^N \boldsymbol{x}^i \,\mathrm{softmax}(\boldsymbol{w})_i,$$

where
$$w_i(\boldsymbol{x}_t) = -\frac{1}{2}\|\alpha_t \boldsymbol{x}^i - \boldsymbol{x}_t^i\|^2/\sigma_t^2$$

Similarly as before, we have for any $p \in \mathbb{N}$
$$\mathbb{E}\Big[\max_{i \in [N]} \|\boldsymbol{x}_i\|^p\Big] \le 2^{2p} N(\Gamma^p + (d/2 + p)^{p/2}).$$

Therefore, we get that
$$\mathbb{E}[\|\boldsymbol{x}_i\|^p] \le 2^{2p} N(\Gamma^p + \sigma_\star^p (d/2 + p)^{p/2}), \qquad \mathbb{E}\big[\|\bar{\boldsymbol{x}}_{\mathrm{mem}}(t, \boldsymbol{x}_t^i)\|^p\big] \le 2^{2p} N(\Gamma^p + \sigma_\star^p (d/2 + p)^{p/2}).$$

Note that this upper-bound is rather loose but given the rate of growth of $N$ with respect to $d$ that we will consider we won't need a tighter bound. For any event A such that $\big\|\bar{\boldsymbol{x}}_{\mathrm{mem}}(t, \boldsymbol{x}_t^i) - \boldsymbol{x}^i\big\|^2 \le C$, we have that
$$\mathbb{E}\Big[\big\|\bar{\boldsymbol{x}}_{\mathrm{mem}}(t, \boldsymbol{x}_t^i) - \boldsymbol{x}^i\big\|^2\Big] \le C\mathbb{P}[\mathsf{A}] + 2\mathbb{P}[\mathsf{A}^c]^{1/2} \left( \mathbb{E}\Big[\big\|\boldsymbol{x}_t^i\big\|^4\Big]^{1/2} + \mathbb{E}\Big[\big\|\bar{\boldsymbol{x}}^{\mathrm{nom}}(t, \boldsymbol{x}_t^i)\big\|^4\Big]^{1/2} \right)$$
$$\le C + 16\left(\Gamma^2 + \sigma_\star^2(d/2 + 4)\right)\mathbb{P}[\mathsf{A}^c]^{1/2}.$$

Therefore, combining this result with Lemma E.5, there exists a numerical constant $C_1 \ge 0$ such that
$$\mathbb{E}\Big[\big\|\bar{\boldsymbol{x}}_{\mathrm{mem}}(t, \boldsymbol{x}_t^i) - \boldsymbol{x}^i\big\|^2\Big]$$
$$\le C_1 \left( \max_{k \in [K]} \|\boldsymbol{\mu}_\star^k\|^2 + (1 + \sigma_\star^2)d \right) N^2 \left( \exp\left[ -\frac{\alpha_t^2 \sigma_\star^2 (1 - \varepsilon)d}{2\sigma_t^2} \right] + \exp[-d\varepsilon^2/16] \right).$$

**Control of the partial memorizing denoiser.** Third, we are going to control $\mathbb{E}\Big[\big\|\boldsymbol{x}_i - \bar{\boldsymbol{x}}_{\mathrm{pmem},M}(t, \boldsymbol{x}_t^i)\big\|^2\Big]$. First, similarly as before, we have that
$$\mathbb{E}[\|\boldsymbol{x}_i\|^p] \le 2^{2p} N(\Gamma^p + \sigma_\star^p (d/2 + p)^{p/2}), \qquad \mathbb{E}\big[\|\bar{\boldsymbol{x}}_{\mathrm{mem}}(t, \boldsymbol{x}_t^i)\|^p\big] \le 2^{2p} N(\Gamma^p + \sigma_\star^p (d/2 + p)^{p/2}).$$

For any event A, we have that
$$\left| \mathbb{E}\Big[\big\|\bar{\boldsymbol{x}}_{\mathrm{pmem},M}(t, \boldsymbol{x}_t^i) - \boldsymbol{x}^i\big\|^2\Big] - \mathbb{E}\Big[\big\|\bar{\boldsymbol{x}}_{\mathrm{pmem},M}(t, \boldsymbol{x}_t^i) - \boldsymbol{x}^i\big\|^2 \mathbf{1}_\mathsf{A}\Big] \right|$$
$$\le 2\mathbb{P}[\mathsf{A}^c]^{1/2} \left( \mathbb{E}\Big[\big\|\boldsymbol{x}_t^i\big\|^4\Big]^{1/2} + \mathbb{E}\Big[\big\|\bar{\boldsymbol{x}}_{\mathrm{pmem},M}(t, \boldsymbol{x}_t^i)\big\|^4\Big]^{1/2} \right)$$
$$\le 16\left(\Gamma^2 + \sigma_\star^2(d/2 + 4)\right)\mathbb{P}[\mathsf{A}^c]^{1/2}. \tag{29}$$

Hence for $i \in [\ell]$, using Lemma E.6 we have
$$\mathbb{E}\Big[\big\|\bar{\boldsymbol{x}}_{\mathrm{pmem},M}(t, \boldsymbol{x}_t^i) - \boldsymbol{x}^i\big\|^2\Big]$$
$$\le 2N\Gamma_\star \exp\left[ -\frac{\alpha_t^2 \sigma_\star^2 (1 - \varepsilon)d}{\sigma_t^2} \right] + 16\left(\Gamma^2 + \sigma_\star^2(d/2 + 4)\right)(8N)^{1/2} \exp[-d\varepsilon^2/16].$$

Now, for $i \notin [\ell]$, we have that

$$\left\| \bar{\boldsymbol{x}}_{\mathrm{pmem},M}(t, \boldsymbol{x}_t^i) - \boldsymbol{x}^i \right\|^2 = \left\| \bar{\boldsymbol{x}}_{\mathrm{pmem},M}(t, \boldsymbol{x}_t^i) - \sum_{j \in \mathsf{S}_i} \mathrm{softmax}(\boldsymbol{w}|_{\mathsf{S}_i})_j \boldsymbol{x}^j + \sum_{j \in \mathsf{S}_i} \mathrm{softmax}(\boldsymbol{w}|_{\mathsf{S}_i})_j \boldsymbol{x}^j - \boldsymbol{x}^i \right\|^2$$

$$= \left\| \bar{\boldsymbol{x}}_{\mathrm{pmem},M}(t, \boldsymbol{x}_t^i) - \sum_{j \in \mathsf{S}_i} \mathrm{softmax}(\boldsymbol{w}|_{\mathsf{S}_i})_j \boldsymbol{x}^j \right\|^2$$

$$+ \left\| \sum_{j \in \mathsf{S}_i} \mathrm{softmax}(\boldsymbol{w}|_{\mathsf{S}_i})_j \boldsymbol{x}^j - \boldsymbol{x}^i \right\|^2$$

$$+ 2 \left\langle \bar{\boldsymbol{x}}_{\mathrm{pmem},M}(t, \boldsymbol{x}_t^i) - \sum_{j \in \mathsf{S}_i} \mathrm{softmax}(\boldsymbol{w}|_{\mathsf{S}_i})_j \boldsymbol{x}^j, \sum_{j \in \mathsf{S}_i} \mathrm{softmax}(\boldsymbol{w}|_{\mathsf{S}_i})_j \boldsymbol{x}^j - \boldsymbol{x}^i \right\rangle$$

Therefore, we have that

$$\left| \left\| \bar{\boldsymbol{x}}_{\mathrm{pmem},M}(t, \boldsymbol{x}_t^i) - \boldsymbol{x}^i \right\|^2 - \left\| \sum_{j \in \mathsf{S}_i} \mathrm{softmax}(\boldsymbol{w}|_{\mathsf{S}_i})_j \boldsymbol{x}^j - \boldsymbol{x}^i \right\|^2 \right|$$

$$\leq \left\| \bar{\boldsymbol{x}}_{\mathrm{pmem},M}(t, \boldsymbol{x}_t^i) - \sum_{j \in \mathsf{S}_i} \mathrm{softmax}(\boldsymbol{w}|_{\mathsf{S}_i})_j \boldsymbol{x}^j \right\|^2$$

$$+ \left\| \bar{\boldsymbol{x}}_{\mathrm{pmem},M}(t, \boldsymbol{x}_t^i) - \sum_{j \in \mathsf{S}_i} \mathrm{softmax}(\boldsymbol{w}|_{\mathsf{S}_i})_j \boldsymbol{x}^j \right\| \left\| \sum_{j \in \mathsf{S}_i} \mathrm{softmax}(\boldsymbol{w}|_{\mathsf{S}_i})_j \boldsymbol{x}^j - \boldsymbol{x}^i \right\|$$

Therefore, we get that

$$\left| \left\| \bar{\boldsymbol{x}}_{\mathrm{pmem},M}(t, \boldsymbol{x}_t^i) - \boldsymbol{x}^i \right\|^2 - \left\| \sum_{j \in \mathsf{S}_i} \mathrm{softmax}(\boldsymbol{w}|_{\mathsf{S}_i})_j \boldsymbol{x}^j - \boldsymbol{x}^i \right\|^2 \right|$$

$$\leq \left\| \bar{\boldsymbol{x}}_{\mathrm{pmem},M}(t, \boldsymbol{x}_t^i) - \sum_{j \in \mathsf{S}_i} \mathrm{softmax}(\boldsymbol{w}|_{\mathsf{S}_i})_j \boldsymbol{x}^j \right\|^2$$

$$+ \left\| \bar{\boldsymbol{x}}_{\mathrm{pmem},M}(t, \boldsymbol{x}_t^i) - \sum_{j \in \mathsf{S}_i} \mathrm{softmax}(\boldsymbol{w}|_{\mathsf{S}_i})_j \boldsymbol{x}^j \right\| \left\| \sum_{j \in \mathsf{S}_i} \mathrm{softmax}(\boldsymbol{w}|_{\mathsf{S}_i})_j \boldsymbol{x}^j - \boldsymbol{x}^i \right\|.$$

Hence, using this result and Lemma E.6 we get that with probability at least $1 - 8N \exp[-d\varepsilon^2/8] - K^{-\log(d)}(1 + \log(K)\log(d))$

$$\mathbb{E}\left[ \left| \left\| \bar{\boldsymbol{x}}_{\mathrm{pmem},M}(t, \boldsymbol{x}_t^i) - \boldsymbol{x}^i \right\|^2 - \left\| \sum_{j \in \mathsf{S}_i} \mathrm{softmax}(\boldsymbol{w}|_{\mathsf{S}_i})_j \boldsymbol{x}^j - \boldsymbol{x}^i \right\|^2 \right| \mathbb{1}_{\mathsf{A}} \right]$$

$$\leq 8\Gamma_\star (1 + N)^2 \exp\left[ -\frac{\alpha_t^2 \gamma^2}{\sigma_t^2} \right] N(\Gamma + \sigma_\star (d/2 + p)^{1/2}),$$

where A is the event of Lemma E.6. Hence, combining this result and (29) we have

$$
\left| \mathbb{E}\left[ \left\| \bar{\boldsymbol{x}}_{\mathrm{pmem},M}(t, \boldsymbol{x}_t^i) - \boldsymbol{x}^i \right\|^2 \right] - \mathbb{E}\left[ \left\| \sum_{j \in \mathsf{S}_i} \mathrm{softmax}(\boldsymbol{w}|_{\mathsf{S}_i})_j \boldsymbol{x}^j - \boldsymbol{x}^i \right\|^2 \right] \right|
$$

$$
\leq \Gamma_\star (1 + N)^2 \exp\left[ -\frac{\alpha_t^2 \gamma^2}{\sigma_t^2} \right]
$$

$$
+ 16 \left( \Gamma^2 + \sigma_\star^2 (d/2 + 4) \right) \left[ (8N)^{1/2} \exp[-d\varepsilon^2/16] + K^{-\log(d)/2}(1 + \log(K)\log(d))^{1/2} \right]
$$

$$
+ 8\Gamma_\star (1 + N)^2 \exp\left[ -\frac{\alpha_t^2 \gamma^2}{\sigma_t^2} \right] N(\Gamma + \sigma_\star (d/2 + p)^{1/2}).
$$

Finally, in order to control $\mathbb{E}\left[ \left\| \sum_{j \in \mathsf{S}_i} \mathrm{softmax}(\boldsymbol{w}|_{\mathsf{S}_i})_j \boldsymbol{x}^j - \boldsymbol{x}^i \right\|^2 \right]$, we use Proposition C.3 which concludes the proof. $\square$

---

**Proposition F.4:** *Assume that $N = \mathrm{poly}(d)$, $\gamma^2 = \Theta(d)$, $\max_{k \in [K]} \|\boldsymbol{\mu}_\star^k\|^2 = \Theta(d)$ and $\sigma_\star^2 = \Theta(1)$. Let $\kappa(d) = \varsigma(\log(d)^2/d)$. We have that uniformly on $t \in [0, \kappa(d)]$*

$$
\mathbb{E}\left[ \mathcal{L}_N(\boldsymbol{x}^1, \ldots, \boldsymbol{x}^\ell, 0) \right] - \mathbb{E}\left[ \mathcal{L}_N(\boldsymbol{x}^1, \ldots, \boldsymbol{x}^N, 0) \right] = \Theta\left( \left( 1 - \frac{\ell}{N} \right) d\sigma_\star^2 \right).
$$

---

*Proof.* The proof of this result is similar to Proposition F.2 by letting $\varepsilon = \Theta(\log(d)/d^{1/2})$. $\square$

## G  Additional Related Work

**Memorization in diffusion models.**   Understanding memorization and generalization properties of diffusion models is crucial for practitioners [Somepalli et al., 2023, Ren et al., 2024, Rahman et al., 2024, Wang et al., 2024a, Chen et al., 2024b, Stein et al., 2024]. Indeed, concerns about privacy [Ghalebikesabi et al., 2023, Carlini et al., 2023, Nasr et al., 2023] and copyright infringement [Cui et al., 2023, Wang et al., 2024a, Vyas et al., 2023, Franceschelli and Musolesi, 2022] are key issues as these models are deployed. Several factors can influence the memorization capabilities of diffusion models. Duplication and out-of-distribution samples have been shown to lead to replication in diffusion models [Carlini et al., 2023, Ross et al., 2024, Webster, 2023] and solutions have been proposed by curating the dataset [Chen et al., 2024a] or introducing dummy data and adapting diffusion models to unseen data [Daras et al., 2024, Yoon et al., 2023]. Neural network architectures have also been shown to play a role in memorization with [Chavhan et al., 2024] identifying specific neurons causing memorization and [Wen et al., 2024, Wang et al., 2024b] analyzing variability in the prediction function to identify problematic prompts in image models. We refer to [Gu et al., 2023] for an in-depth experimental investigation of those issues. Finally, we highlight the work of [Kadkhodaie et al., 2023], in which the authors investigate the inductive bias image denoisers to investigate the generalization properties of state-of-the-art diffusion models.

**Definitions of memorization.**   Note that our definition of memorization (Definition 2.2) is connected to the one of Eidetic Memorization as introduced by Carlini et al. [2023]. Similarly, to [Carlini et al., 2023], we acknowledge that this notion of memorization is *strong* as it implies that an image is memorized if there exists a near perfect copy in the training dataset which does not fully capture *copyright infringement* or *data privacy* issues. We refer to [Elkin-Koren et al., 2023] for an in-depth discussion of those issues.

**Memorization and overfitting.**   *A priori*, it may not be clear why memorization in diffusion models is substantially different from *overfitting* which has been a long-standing and well-studied property of machine learning systems Hastie et al. [2009]. Overfitting is characterized by a large gap between training and validation losses. However, overfitting is not directly connected to memorization. For a small number of parameters, the partially memorizing denoisers defined in Section 3 have large training and validation losses which ultimately have a small gap, meaning that such models are not

significantly overfit; however, every single sample they generate is memorized. Meanwhile, the model corresponding to the fully memorizing denoiser defined in Section 2 is both very overfit and does memorize. It is also worthwhile to discuss the connection of diffusion model memorization with *benign overfitting* [Bartlett et al., 2020] and *double descent* [Belkin et al., 2019]. In particular, we wish to clarify why the setting of diffusion models may be separated from benign overfitting. The core difference is as follows: in the usual double descent setting, one studies an *overparameterized* learning problem, such as regression or classification, and there are many models at a fixed parameter count which obtain the minimal training loss; then it is up to (implicit) regularization to choose the best solution, which benignly interpolates the training data. As the number of parameters increase, the training loss of the trained model never increases, and the validation loss eventually also decreases (after an initial increase for the purpose of overfitting). Meanwhile, in the case of diffusion, there is exactly one model which obtains the minimal training loss, which is the memorizing denoiser, and this does not change no matter how many parameters are allocated. Certainly the memorizing denoiser does not benignly interpolate the training data. As the number of parameters increase past a certain point, the training loss never decreases, and in fact may increase, showing that double descent or benign overfitting indeed do not apply straightforwardly to the case of diffusion.

## H   Experimental Details

We run all experiments on several Nvidia A100 80GB GPUs using Jax 0.6.0 and Equinox 0.12 [Kidger and Garcia, 2021]. Each training/evaluation job occurs on a single A100 and the results are saved to file to be aggregated later. Aggregation, analysis, and visualization occur on a single A100.

### H.1   Datasets, Optimization, and Initialization Details

In our experiments in Section 4.1, we generate synthetic Gaussian mixture models, by generating the $K$ means $\boldsymbol{\mu}_\star^i$ uniformly on the sphere of radius $\sqrt{d}$ and setting the ground truth variance $\sigma_\star^2 = 1$, as prescribed by the results in Section 3. The experiments conducted in Figures 1 and 2 and Appendix H.5 are all under the setting $N = 200$, $d = 50$, and $K = 12$. For the sweep in Figure 3 we take $(N, d, K)$ tuples from $[50, 100, 150, 200] \times [30, 40, 50, 60] \times [3, 6, 9, 12]$, obtaining a total of 64 $(N, d, K)$ tuples. For each setting of $(N, d, K)$ (including the sweep in Figure 3) we train 20 models at different model sizes $M$: starting from $M = \lfloor N/10 \rfloor$ and moving in increments of $\lfloor N/10 \rfloor$ to $M = 10\lfloor N/10 \rfloor$; then, training 10 more models where $M$ is equally spaced between the $M$ where the phase transition starts and the $M$ where it ends (using the empirical criterion in Section 4).

In our experiments in Section 4.2, we generate synthetic colored FashionMNIST data by sampling $K$ FashionMNIST [Xiao et al., 2017] images uniformly at random to use as "templates", then using the PIL (Pillow) utility to reshape them to $15 \times 15$ resolution. For each of the $K$ templates (components) we generate a color vector using a Gaussian with ground truth mean and variance $(\boldsymbol{u}_\star^i, \boldsymbol{\sigma}_\star^2) = (\mathbf{0}, 1) \in \mathbb{R}^3 \times \mathbb{R}$. We take the Kronecker product of the color vector and the template as described in Section 4.2 in order to form the sample. Figure 4 uses the setting $N = K = 8$ while the experiment in Figure 5 uses the setting $N = 100$, $K = 4$, and color dimension $d = 3$. Here we train 10 models at different model sizes $M$, starting from $M = \lfloor N/10 \rfloor$ and moving in increments of $\lfloor N/10 \rfloor$ to $M = 10\lfloor N/10 \rfloor$.

For all experiments, we use the "variance preserving" process which yields $\alpha_t = \sqrt{1 - t^2}$ and $\sigma_t = t$ for $t \in [0, 1]$. We use the objective (5) to train our model denoisers. For training, we always train with the loss weighting $\lambda(t) := \alpha_t^2/\sigma_t^2$, which is equivalent to using *noise prediction* ([Karras et al., 2022]), and $t \sim \mathrm{Unif}((t_\ell)_{\ell=0}^L)$ where we use $L = 25$ decreasing timesteps $t_\ell = 0.01 + 0.998(L-\ell)/L = 0.999 - 0.998\ell/L \in (0, 1)$. In lieu of computing the (obviously intractable) inner expectation in $\mathcal{L}_{N,t}$, we use $N_{\mathrm{dup}} := 100$ Gaussian noise draws for each of the $N$ samples to estimate the expectation. We use full-batch Adam for $N_{\mathrm{epochs}}$ epochs (also, iterations) to optimize the objective; for experiments in Section 4.1 we have $N_{\mathrm{epochs}} = 50,000$ and for experiments in Section 4.2 we have $N_{\mathrm{epochs}} = 100,000$. We use a "warmup-decay" learning rate schedule: for $N_{\mathrm{warmup}} := N_{\mathrm{epochs}}/10$ epochs the learning rate linearly increases from 0 to $10^{-3}$; for the remaining $N_{\mathrm{decay}} := N_{\mathrm{epochs}} - N_{\mathrm{warmup}}$ epochs the learning rate linearly decreases from $10^{-3}$ to $10^{-6}$.

For all models we train, we use a "partial memorization initialization" along with the Adam optimizer We use this initialization because the loss at a truly random initialization is extremely high, in many

cases often *at least eight orders of magnitude* larger than the loss at optimum, and Adam is often unable to learn effectively given this massive conditioning. The partial memorizing initialization, in the context of the isotropic Gaussian mixture model, sets the initial $\sigma^2$ to $10^{-6}$, and each mean $\boldsymbol{\mu}^i$ to a random sample. In the context of the simple image model, it sets each initial template parameter $\boldsymbol{A}_{\boldsymbol{x}^i}$ to the template which generates a sample, the corresponding initial color vector to the unique vector which generates the (not identically zero) sample given that template, and the initial color variance $\sigma^2$ to $10^{-6}$. Notice that in the experiments such as Figure 2, even models initialized with this partial memorizing initialization end up learning (nearly) generalizing solutions — unless of course it is more favorable to memorize, which occurs with very large $M$.

## H.2 Formal Description of Sampling Scheme

We use the implementation of the DDIM sample prescribed in De Bortoli et al. [2025], i.e., using the above notation and given a denoiser $\bar{\boldsymbol{x}}$

$$\hat{\boldsymbol{x}}_{t_{\ell+1}} = \frac{\sigma_{t_{\ell+1}}}{\sigma_{t_\ell}}\hat{\boldsymbol{x}}_{t_\ell} + \left(\alpha_{t_{\ell+1}} - \frac{\sigma_{t_{\ell+1}}}{\sigma_{t_\ell}}\alpha_{t_\ell}\right)\bar{\boldsymbol{x}}(t_\ell, \hat{\boldsymbol{x}}_{t_\ell}), \qquad \hat{\boldsymbol{x}}_{t_0} \sim \mathcal{N}(\boldsymbol{0}, \boldsymbol{I}).$$

As previously stated, we use increasing timesteps $t_\ell = 0.999 - \ell/L$ for $L = 25$ and $\ell \in \{0, 1, \dots, L\}$. Notice that we use the same timesteps for sampling as for training. We implement this iteration using the DiffusionLab PyPI package [Pai, 2025].

## H.3 Formal Description of Loss Weighting Regression

Recall that in Section 4.1 we predicted the phase transition using the loss approximations derived in Section 3. To do this, we solved a regression problem (15). Here, we will discuss how we solve this problem efficiently by smoothing and regularization.

Namely, as a bilevel semi-discrete quadratic program, the problem (15) is hard to optimize outright via gradient methods. As a result we parameterize the distribution over $M$ via a temperature-weighted softmax with high temperature $\tau = 1/20$, placing an entropy penalty ($\beta_{\text{sparsity}} = 10^{-3}$) on the softmax output to ensure that the learned distribution over $M$ is sparse. The overall problem is (using the notation from (15))

$$\min_{\tilde{\lambda}} \sum_{(N,d,K)} \left(\frac{\bar{M}_{\text{pt}}(N, d, K, \tilde{\lambda})}{N} - \frac{M_{\text{pt}}}{N}\right)^2 \tag{30}$$

$$- \beta_{\text{sparsity}} \sum_{(N,d,K)} \sum_{M} \tilde{p}(N, d, K, M, \tilde{\lambda}) \log \tilde{p}(N, d, K, M, \tilde{\lambda})$$

$$\text{where} \quad \bar{M}_{\text{pt}}(N, d, K, \tilde{\lambda}) = \sum_{M} M \cdot \tilde{p}(N, d, K, M, \tilde{\lambda})$$

$$\text{and} \quad \tilde{p}(N, d, K, \cdot, \tilde{\lambda}) = \text{softmax}\left(\left\{-\frac{1}{\tau}\left(\sum_{\ell=0}^{L} \tilde{\lambda}(t_\ell)(\check{L}_{N,t}(\theta_{\text{pmem},M}(N, d, K)) - \check{L}_{N,t}(\theta_\star(N, d, K)))\right)^2\right\}_M\right)$$

In all cases, when we report the loss we refer to just the MSE component described in (30).

## H.4 Simple Image Model Calculations

In this section we simplify the computation of the colored image denoiser in Section 4.2. Recall our setup in Section 4.2 along with the notation in Lemma 2.1. Under the settings $\boldsymbol{\mu}_\star^i = \boldsymbol{A}_\star^i \boldsymbol{u}_\star^i$ and $\boldsymbol{\Sigma}_\star^i = \sigma_\star^2 \boldsymbol{A}_i^\star (\boldsymbol{A}_i^\star)^\top$, it holds that

$$\boldsymbol{\mu}_\star^i = \boldsymbol{A}_\star^i \boldsymbol{u}_\star^i = (\boldsymbol{I} \otimes \boldsymbol{x}_\star^i)\boldsymbol{u}_\star^i = \boldsymbol{u}_\star^i \otimes \boldsymbol{x}_\star^i$$

$$\boldsymbol{\Sigma}_\star^i = \sigma_\star^2(\boldsymbol{I} \otimes \boldsymbol{x}_\star^i)(\boldsymbol{I} \otimes \boldsymbol{x}_\star^i)^\top = \sigma_\star^2(\boldsymbol{I} \otimes \boldsymbol{x}_\star^i)(\boldsymbol{I} \otimes (\boldsymbol{x}_\star^i)^\top) = \sigma_\star^2(\boldsymbol{I} \otimes \boldsymbol{x}_\star^i(\boldsymbol{x}_\star^i)^\top)$$

Also, simplifying the inverse $(\alpha^2 \boldsymbol{\Sigma}_\star^i + \sigma^2 \boldsymbol{I})^{-1}$ using the Sherman-Morrison-Woodbury identity obtains

$$
\begin{aligned}
(\alpha^2 \boldsymbol{\Sigma}_\star^i + \sigma^2 \boldsymbol{I})^{-1} &= \frac{1}{\sigma^2}\left(\boldsymbol{I} + \frac{\alpha^2}{\sigma^2}\boldsymbol{\Sigma}_\star^i\right)^{-1} \\
&= \frac{1}{\sigma^2}\left(\boldsymbol{I} + \frac{\alpha^2\sigma_\star^2}{\sigma^2}\boldsymbol{A}_\star^i(\boldsymbol{A}_\star^i)^\top\right)^{-1} \\
&= \frac{1}{\sigma^2}\left(\boldsymbol{I} - \boldsymbol{A}_\star^i\left[\frac{\sigma^2}{\alpha^2\sigma_\star^2}\boldsymbol{I} + (\boldsymbol{A}_\star^i)^\top \boldsymbol{A}_\star^i\right]^{-1}(\boldsymbol{A}_\star^i)^\top\right).
\end{aligned}
$$

Calculating the interior term first, we obtain

$$
\begin{aligned}
(\boldsymbol{A}_\star^i)^\top \boldsymbol{A}_\star^i &= (\boldsymbol{I} \otimes \boldsymbol{x}_\star^i)^\top(\boldsymbol{I} \otimes \boldsymbol{x}_\star^i) = (\boldsymbol{I} \otimes (\boldsymbol{x}_\star^i)^\top)(\boldsymbol{I} \otimes \boldsymbol{x}_\star^i) = \boldsymbol{I} \otimes [(\boldsymbol{x}_\star^i)^\top(\boldsymbol{x}_\star^i)] \\
&= \boldsymbol{I} \otimes [\|\boldsymbol{x}_\star^i\|^2] = \|\boldsymbol{x}_\star^i\|^2 \boldsymbol{I},
\end{aligned}
$$

which yields

$$
\begin{aligned}
(\alpha^2 \boldsymbol{\Sigma}_\star^i + \sigma^2 \boldsymbol{I})^{-1} &= \frac{1}{\sigma^2}\left(\boldsymbol{I} - \frac{1}{\frac{\sigma^2}{\alpha^2\sigma_\star^2} + \|\boldsymbol{x}_\star^i\|^2}\boldsymbol{A}_\star^i(\boldsymbol{A}_\star^i)^\top\right) \\
&= \frac{1}{\sigma^2}\left(\boldsymbol{I} - \frac{\alpha^2\sigma_\star^2}{\sigma^2 + \alpha^2\sigma_\star^2\|\boldsymbol{x}_\star^i\|^2}\boldsymbol{A}_\star^i(\boldsymbol{A}_\star^i)^\top\right)
\end{aligned}
$$

Another part that requires elaboration is the action of $(\boldsymbol{A}_\star^i)^\top$ on a (block) vector, say $\boldsymbol{\theta}$, which has

$$
(\boldsymbol{A}_\star^i)^\top \boldsymbol{\theta} = (\boldsymbol{I} \otimes \boldsymbol{x}_\star^i)^\top \boldsymbol{\theta} = (\boldsymbol{I} \otimes (\boldsymbol{x}_\star^i)^\top)\boldsymbol{\theta} = \begin{bmatrix} (\boldsymbol{x}_\star^i)^\top \boldsymbol{\theta}_1 \\ \vdots \\ (\boldsymbol{x}_\star^i)^\top \boldsymbol{\theta}_c \end{bmatrix}.
$$

Finally, the last part that can do with simplification is the log-determinant, which obtains

$$
\begin{aligned}
\log\det(\alpha^2 \boldsymbol{\Sigma}_\star^i + \sigma^2 \boldsymbol{I}) &= \log\det\left(\sigma^2\left[\boldsymbol{I} + \frac{\alpha^2}{\sigma^2}\boldsymbol{\Sigma}_\star^i\right]\right) \\
&= \log\det(\sigma^2 \boldsymbol{I}) + \log\det\left(\boldsymbol{I} + \frac{\alpha^2}{\sigma^2}\boldsymbol{\Sigma}_\star^i\right) \\
&= 2cd^2\log(\sigma) + \log\det\left(\boldsymbol{I} + \frac{\alpha^2\sigma_\star^2}{\sigma^2}\boldsymbol{A}_\star^i(\boldsymbol{A}_\star^i)^\top\right) \\
&= 2cd^2\log(\sigma) + \log\det\left(\boldsymbol{I} + \frac{\alpha^2\sigma_\star^2}{\sigma^2}(\boldsymbol{A}_\star^i)^\top \boldsymbol{A}_\star^i\right) \\
&= 2cd^2\log(\sigma) + \log\det\left(\boldsymbol{I} + \frac{\alpha^2\sigma_\star^2}{\sigma^2}\|\boldsymbol{x}_\star^i\|^2\boldsymbol{I}\right) \\
&= 2cd^2\log(\sigma) + \log\det\left(\left\{1 + \frac{\alpha^2\sigma_\star^2}{\sigma^2}\|\boldsymbol{x}_\star^i\|^2\right\}\boldsymbol{I}\right) \\
&= 2cd^2\log(\sigma) + c\log\left(1 + \frac{\alpha^2\sigma_\star^2}{\sigma^2}\|\boldsymbol{x}_\star^i\|^2\right) \\
&= c\left(2d^2\log(\sigma) + \log\left\{1 + \frac{\alpha^2\sigma_\star^2}{\sigma^2}\|\boldsymbol{x}_\star^i\|^2\right\}\right).
\end{aligned}
$$

These terms are all simple to compute, and therefore so is the true denoiser (as well as any we parameterize).

### H.5 More Experimental Results

**Interpreting the learned models.** In this section we take advantage of the white-box nature of our denoiser and specifically how it relates to the data generating process, to try to uncover some mechanistic aspects of memorization in this setup. There are two questions we wish to answer:

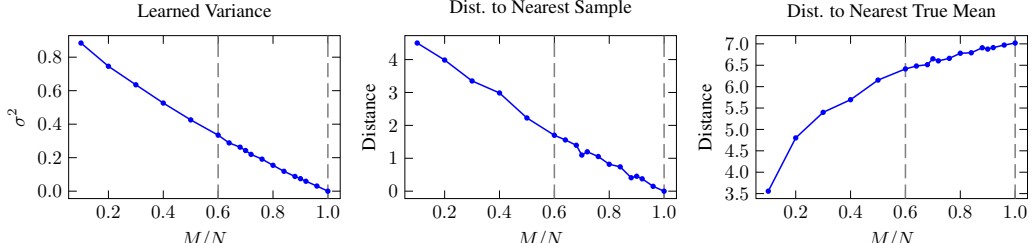

**Figure 6:** *Left:* A plot of the learned variance as a function of the model size $M$. *Middle:* The average distance of a learned mean to its nearest sample in the training data, as a function of $M$. *Right:* The average distance of a learned mean to the nearest ground truth mean, as a function of $M$. All plots include the start and end of the phase transition. While the variance eventually decays to 0, it surprisingly only does so *linearly*, and for every $M$ before the *end* of the phase transition the ground truth variance does not collapse to 0. Similarly, as the memorization ratio increases and the phase transition occurs, the average distance from a learned mean to the nearest sample decreases *linearly* as a function of $M$. Meanwhile, the average distance from a learned mean to the nearest true mean *increases*. Surprisingly, this behavior happens during the generalization phase as well. Note that the provided example is representative among our trained models.

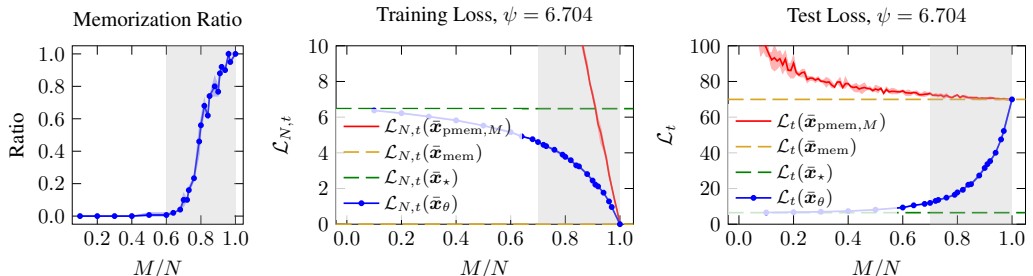

**Figure 7: Our loss calculations and memorization trends are stable under different random seeds.** We observe the same behaviors as Figure 2 when re-attempting the same experiment with three separate random seeds; we provide error bars but note that they are extremely small, indicating a tiny variance.

- Is the learned variance $\sigma^2_{\text{train}}$ an effective proxy for memorization or generalization?
- How do the learned means $\boldsymbol{\mu}^i_{\text{train}}$ behave in memorized and generalized models?

Recall that the ground truth (generalizing) denoiser has variance $\sigma^2_\star$ and means $\boldsymbol{\mu}^i_\star$, and the memorizing denoiser has variance 0 and means $\boldsymbol{x}^i$. Thus, one intuition would be that a smaller variance implies a propensity of the trained model to memorize, and learned means closer to samples would imply the same. To verify this behavior we plot the learned variance and the average distance of the learned means to training samples and ground truth means, respectively, in Appendix H.5. We can confirm that our basic intuition is true, with a twist: while the learned variance $\sigma^2_{\text{train}}$ decreases as the model size $M$ increases, it does so linearly, and *even in the generalization regime*. Similarly the average distance of each learned mean to the closest point in the training dataset decreases linearly to 0. In this sense, these and other mechanistically derived quantities may serve as a continuous proxy for the rapid phase transition, a behavior also observed in large models with far more complicated "circuits" [Nanda et al., 2023, Schaeffer et al., 2023].

**Assessing the variance of different seeds.** In Figure 7, we examine multiple runs using different random seeds to see their effect on the loss and memorization plots (akin to Figure 1). The shading on the memorization plot, which showcases the minimum and maximum value of the quantity across three seeds, amounts to error bars on the regression experiment in Figure 3, since the approximated losses will be the same (as they are computed deterministically), and the only remaining variation is the regression target, i.e., the location of the phase transition.

