# OpenReview forum: "On the Edge of Memorization in Diffusion Models"
_NeurIPS.cc/2025/Conference — NeurIPS 2025 poster_

### Official Review · Reviewer_1zED · 2025-06-10

**Clarity:** 3
**Significance:** 2
**Originality:** 3
**Rating:** 4
**Confidence:** 4

**Summary:**

The paper investigates memorization in (unconditional) diffusion models from a theoretical perspective. More specifically, the paper explores a so-called crossover point, where the training loss landscape of a diffusion model starts to prefer memorizing training data over generalizing across them. Based on this concept, the paper defines the difference between memorization and generalization. It offers a framework to identify the phase transition of a diffusion model's critical capacity when the model starts to favor memorization. This crossover point is theoretically motivated and empirically shown for experiments on Gaussian mixture models and (small) monochromatic images.

**Questions:**

Regarding the 3rd point stated under weaknesses, is it correct that the model always requires multiple training runs with different capacities to identify and predict the crossover point? Or is it possible to predict the number of parameters for, e.g., a model trained on ImageNet?

**Ethical Concerns:**

["NO or VERY MINOR ethics concerns only"]

**Final Justification:**

After reading the paper and going through the reviews and rebuttals, I am not 100% convinced that the findings generalize to more complex and capable diffusion models. However, I think the paper provides interesting insights from a theoretical perspective. Therefore, I support a (weak) acceptance.

**Limitations:**

The paper mainly discusses future work and potential extensions in the conclusion section but lacks a clear statement of the method's limitations.

**Paper Formatting Concerns:**

No concerns identified

**Quality:**

3

**Strengths And Weaknesses:**

**Strengths:**
- The paper lays down a well-motivated theoretical foundation for memorization in diffusion models. The core of the analysis -- the crossover point -- is derived in detail, and important, relevant research and foundations are extensively cited. The proposed framework offers fascinating insights into understanding memorization in diffusion models.
- The paper is, in general, well-written and presents all findings appealingly.

**Weaknesses:**
- My primary concern is that the theoretical concept of the crossover point is only shown for particular synthetic cases. While the paper promises a generalization to more complex datasets, empirical proof is missing. While I understand that training diffusion models can be costly, I would have expected to see at least some experiments on small scales, such as CIFAR-10.
- The paper writing is based heavily on mathematical symbols. While Tables 1 and 2 provide a good lookup, it is sometimes hard to follow without frequently looking up symbols from previous sections. Repeating symbol names or using names instead of symbols would improve the readability of the paper. This also includes figures, e.g., Figures 1 and 3, where stating the name of the symbols on the axis would improve their readability.
- If I understood the method correctly, it seems like the method requires reference models / multiple training runs with different model capacities to identify the phase transition in practice. Assuming one wants to train a larger diffusion model without a valid reference, one cannot a priori define a sufficient number of parameters to avoid memorization.

---

> ### Author Rebuttal · Authors · 2025-07-31
>
> We thank the reviewer for their careful review of our submission, and for recognizing our work’s “well-motivated theoretical foundation” and “fascinating insights”. We respond point-by-point below, and look forward to discussing further.
>
> > My primary concern is that the theoretical concept of the crossover point is only shown for particular synthetic cases. While the paper promises a generalization to more complex datasets, empirical proof is missing. While I understand that training diffusion models can be costly, I would have expected to see at least some experiments on small scales, such as CIFAR-10.
>
> In our view, the memorization-generalization interplay in diffusion models is a holistic phenomenon with numerous interrelated causes (as evidenced by the diversity of explanations proposed in the literature [1-4]), and to develop a mathematical understanding of it, it is necessary to study it in a setting in which these different causes can be disentangled, precisely varied, and their effects examined. In this sense, our work aims to build a *theoretical foundation* for studying memorization in diffusion models. Due to this scope, we do not attempt to rigorously mathematically analyze real data and real networks in our work. However, we note (here and in the paper) that experimental results in our simplified laboratory setting reproduce phenomena observed in previous empirical work studying memorization and generalization in real diffusion models trained on real data (i.e., larger than CIFAR10 scale), see e.g. [5].
>
> > The paper writing is based heavily on mathematical symbols. While Tables 1 and 2 provide a good lookup, it is sometimes hard to follow without frequently looking up symbols from previous sections.
>
> Thank you for the feedback; while our work is intended to be theoretical in nature, and therefore involves significant amounts of mathematics, we agree that the notation indeed needs improvement, and we commit to streamlining it in the revision. With the additional content page, we will also add descriptions of the relevant variables before using them, e.g., “the number of components $K$”.
>
> > If I understood the method correctly, it seems like the method requires reference models / multiple training runs with different model capacities to identify the phase transition in practice. Assuming one wants to train a larger diffusion model without a valid reference, one cannot a priori define a sufficient number of parameters to avoid memorization.
>
> > Regarding the 3rd point stated under weaknesses, is it correct that the model always requires multiple training runs with different capacities to identify and predict the crossover point? Or is it possible to predict the number of parameters for, e.g., a model trained on ImageNet?
>
> Indeed, in our methodology, we need to build a predictive model by learning the loss weighting $\tilde{\lambda}$.  However, as we show in the paper, we can use the learned loss weighting to extrapolate to different (held-out) training configurations, and predict the location of the phase transition from generalization to memorization with high accuracy. Notice that this is similar to the current literature on *scaling laws*, where a simple law relating the size of the model to some property (such as the performance on some benchmark) is fit using a number of trained models, and then used to predict the given property for different model configurations without training those models [6]. In this context we can reframe a contribution of this paper as: within our laboratory, we derive a hypothesized scaling law form for memorization and demonstrate that it holds to high accuracy in practice. We humbly point out that scaling law work conventionally requires this curve-fitting and cannot give a principled hypothesis as to the form of the scaling law. As a result, we do not view this limitation as a major weakness, since in fact it constitutes an improvement (a prediction about the form of the scaling law) compared to numerous works in this domain.
>
> Please let us know if you have other questions or hang-ups, and we will be happy to resolve them. We look forward to a productive discussion during the rebuttal period.
>
> # References
>
> [1] Kadkhodaie, Z., Guth, F., Simoncelli, E. P., & Mallat, S. (2023). Generalization in diffusion models arises from geometry-adaptive harmonic representation. In arXiv [cs.CV]. arXiv.
>
> [2] Kamb, M., & Ganguli, S. (2024). An analytic theory of creativity in convolutional diffusion models. In arXiv [cs.LG]. arXiv.
>
> [3] Finn, E., Keller, T. A., Theodosis, M., & Ba, D. E. (2025). Origins of creativity in attention-based diffusion models. In arXiv [cs.LG]. arXiv.
>
> [4] Niedoba, M., Zwartsenberg, B., Murphy, K., & Wood, F. (2024). Towards a mechanistic explanation of diffusion model generalization. In arXiv [cs.LG]. arXiv.
>
> [5] Zhang, H., Zhou, J., Lu, Y., Guo, M., Shen, L., & Qu, Q. (2023). The Emergence of Reproducibility and Consistency in Diffusion Models. In arXiv [cs.LG]. arXiv.
>
> [6] Kaplan, J., McCandlish, S., Henighan, T., Brown, T.B., Chess, B., Child, R., Gray, S., Radford, A., Wu, J. and Amodei, D., 2020. Scaling laws for neural language models. In arXiv [cs.LG]. arXiv.

---

> > ### Comment · Reviewer_1zED · 2025-08-01
> >
> > I thank the authors for their response and clarifications. I think improving the paper's writing indeed helps with the understanding of it. I am still not sure about how well the findings generalize to more complex or larger datasets. I understand that some insights can be drawn from smaller experiments, and I also acknowledge that training diffusion models is very costly, and running large scale experiments are not feasible for smaller research labs or universities (which is why I do not want to blame the authors for that). However, small scale experiments still limit the greater impact of the paper. Overall, I decided to keep my initial score of rending to accept the paper and looking forward to discussing the paper with the other reviewers.

---

### Official Review · Reviewer_DzmC · 2025-07-03

**Clarity:** 3
**Significance:** 2
**Originality:** 2
**Rating:** 4
**Confidence:** 4

**Summary:**

The paper introduces a theoretical framework to investigate the conditions under which diffusion models either memorize their training data or generalize. The authors model both the data-generating distribution and the denoiser using Gaussian mixtures, which makes the problem analytically tractable. The primary contribution is the theoretical characterization of a crossover point in model underparameterization at which the loss of a simplified, partially memorizing model becomes lower than that of an ideal, generalizing model. The authors demonstrate empirically that this crossover point accurately predicts the onset of a phase transition from generalization to memorization in trained models. Experiments are performed both with a simple isotropic mixture and a low-rank setting designed to emulate the structure of images.

**Questions:**

To strengthen the claim that your framework captures a more general principle, have you considered an experiment with a simple neural network (e.g., an MLP) as the denoiser? While your analytical predictions for the crossover point would not hold, demonstrating that a standard network trained with gradient descent exhibits the same qualitative phase transition as a function of the number of neurons would partially reinforce the paper's relevance.

**Ethical Concerns:**

["NO or VERY MINOR ethics concerns only"]

**Final Justification:**

I am maintaining my rating of borderline accept. The paper is technically sound. However, my primary concern remains the very synthetic nature of the studied setting.

**Limitations:**

Yes.

**Paper Formatting Concerns:**

The paper uses the 2024 template, not the 2025 one.

**Quality:**

3

**Strengths And Weaknesses:**

**Strengths**

1. The paper studies the timely and practically relevant problem of memorization in diffusion models.
2. It provides a sound and rigorous theoretical analysis that appears correct.
3. It is well-written, with clear explanations and figures that support the central claims.
4. Numerical experiments are provided to validate the theoretical predictions.

**Weaknesses**

5. The simplifications required for theoretical tractability (i.e., the GMM assumption) create a substantial gap between the analyzed model and those used in practice.
6. The novelty of the core insight feels somewhat limited. The finding that increased model capacity leads to memorization is well-established. This work's contribution is to formalize this intuition and provide a predictive formula within a specific, very simplified setting, rather than offering a fundamentally new conceptual understanding of the phenomenon itself.

*Minor*

Equation (8) contains an extra $I$.

---

> ### Author Rebuttal · Authors · 2025-07-31
>
> We thank the reviewer for their careful review of our submission, and for recognizing our work’s “sound and rigorous theoretical analysis”. We respond point-by-point below, and look forward to discussing further.
>
> > The simplifications required for theoretical tractability (i.e., the GMM assumption) create a substantial gap between the analyzed model and those used in practice.
>
> This is an unavoidable artifact of developing rigorous theory for a system as complex as diffusion models. Moreover, in spite of the apparent
> simplicity of this data model and its associated denoisers, even basic
> theoretical questions about, for example, the number of parameters at which
> models learned from the class of GMM denoisers tend to memorize, and the relationship between these trained models and the sampling-time behavior of the diffusion sampling algorithm, remain completely open.
>
> More precisely, in the theoretical literature on diffusion models, the Gaussian mixture model (GMM) we study actually sits at the ‘frontier’: almost any algorithmic research question one can formulate around diffusion models remains open for GMMs. Specifically with regards to the behavior of trained models, [1] proves that the minimizers of the training loss over a properly-parameterized class of GMMs generalize (in our notation, analogous to the number of components $M$ being equal to the number of classes $K$), [2] proves the same for models learned by gradient descent for two-point mixtures of Gaussians, and [3–4] establish guarantees for learning properly-parameterized GMMs with non-gradient-descent algorithms. To the best of our knowledge, no prior work has studied the setting where the number of components $M$ may be significantly larger than the number of classes $K$, which is the essential setting to understand memorization in GMM diffusion models as they require $M \gg K$ parameters to memorize with low loss. In addition, because existing theoretical works apply only to the ‘properly parameterized’ setting where $M = K$, they are able to guarantee that trained models generalize at sampling time by applying existing results on diffusion model sampling under accurate score estimates (e.g., [5–6]). In our setting where $M \gg K$, **it is generally intractable to exactly characterize learned models**, which means that characterizing the sampling-time behavior of learned models (i.e., memorizing or generalizing) is a completely open problem – even the Gaussian case (i.e., $K=1$) is highly nontrivial [7]. Our work paves the way towards a rigorous theoretical understanding of this non-trivial question via the hybrid theoretical-empirical approach we employ, which correctly predicts the location of the phase transition between memorization and generalization in terms of simple properties of the loss landscape.
>
> > The novelty of the core insight feels somewhat limited. The finding that increased model capacity leads to memorization is well-established. This work's contribution is to formalize this intuition and provide a predictive formula within a specific, very simplified setting, rather than offering a fundamentally new conceptual understanding of the phenomenon itself.
>
> We respectfully disagree with your characterization of this work’s contributions and novelty, and would phrase it as follows. As a theoretical work, our work’s main contributions are proposing the GMM “laboratory” as an environment for studying emergent phenomena in diffusion models, and demonstrating its utility in rigorously characterizing the phase transition between generalization and memorization. Although this is a “simplified setting”, we note above that the theoretical study of diffusion models in the GMM setting is completely open as far as memorization behavior at intermediate model sizes is concerned. In this sense, our work aims to build a *theoretical foundation* for studying complex phenomena in diffusion models, rather than simply adding another explanation in a general setting, and we demonstrate the viability of this in characterizing the phase transition.
>
> > To strengthen the claim that your framework captures a more general principle, have you considered an experiment with a simple neural network (e.g., an MLP) as the denoiser? While your analytical predictions for the crossover point would not hold, demonstrating that a standard network trained with gradient descent exhibits the same qualitative phase transition as a function of the number of neurons would partially reinforce the paper's relevance.
>
> We thank the reviewer for this suggestion. Firstly, we would highlight that the GMM denoisers we train in our work (eqn.s (9) and (12)) are in fact primitive MLPs, as inspection of these formulae reveals, with a softmax nonlinearity and a specific weight-tying structure. With regards to more practical architectures, we note that this experiment has been performed in prior work in the literature, and the reviewer’s suspicions confirmed. Specifically, in [1, Figure 4], the authors demonstrate under a slightly different metric for memorization (which measures similarity using a fixed neural feature descriptor model instead of comparing $\ell^2$ distances) that U-net models trained on GMM data undergo a rapid phase transition as the number of training data is varied. The existence of these results led us to focus on theoretically characterizing the phase transition, as in the submission.
>
> Please let us know if you have other questions or hang-ups, and we will be happy to resolve them. We look forward to a productive discussion during the rebuttal period.
>
> # References
> [1] Wang, P., Zhang, H., Zhang, Z., Chen, S., Ma, Y., & Qu, Q. (2024). Diffusion Models Learn Low-Dimensional Distributions via Subspace Clustering. arXiv preprint arXiv:2409.02426.
>
> [2] Shah, K., Chen, S., & Klivans, A. (2023). Learning Mixtures of Gaussians Using the DDPM Objective. In Advances in Neural Information Processing Systems 36 (NeurIPS 2023).
>
> [3] Gatmiry, K., Kelner, J., & Lee, H. (2024). Learning Mixtures of Gaussians Using Diffusion Models. arXiv preprint arXiv:2404.18869.
>
> [4] Chen, S., Kontonis, V., & Shah, K. (2024). Learning general Gaussian mixtures with efficient score matching. arXiv preprint arXiv:2404.18893.
>
> [5] Chen, S., Chewi, S., Li, J., Li, Y., Salim, A., & Zhang, A. R. (2023). Sampling is as easy as learning the score: theory for diffusion models with minimal data assumptions. In The Eleventh International Conference on Learning Representations (ICLR 2023).
>
> [6] Benton, J., De Bortoli, V., Doucet, A., & Deligiannidis, G. (2024). Nearly d-Linear Convergence Bounds for Diffusion Models via Stochastic Localization. In The Twelfth International Conference on Learning Representations (ICLR 2024).
>
> [7] Hurault, S., Terris, M., Moreau, T., & Peyré, G. (2025). From score matching to diffusion: A fine-grained error analysis in the Gaussian setting. arXiv preprint arXiv:2503.11615.

---

> ### Author Response · Authors · 2025-08-05
> **Discussion Period Ending**
>
> As we approach the end of the discussion period, we wanted to check back in with you and inquire as to whether you have any remaining questions, comments, or concerns. If so, we are happy to discuss them.

---

> ### Comment · Reviewer_DzmC · 2025-08-06
>
> Thanks for your answers. I acknowledge and appreciate that theory necessitates simplifying assumptions. Yet, my primary concern remains that the theoretical framework is simplified to an extent that it represents a significant departure from diffusion models. Consequently, I am not fully convinced that the derived insights substantially advance our understanding of the behavior of more complex and practically interesting systems.
>
> The paper is technically sound. However, given the concerns about the impact and applicability of its findings, I will retain my borderline acceptance recommendation.

---

### Official Review · Reviewer_vocx · 2025-07-03

**Clarity:** 2
**Significance:** 3
**Originality:** 3
**Rating:** 5
**Confidence:** 2

**Summary:**

This paper studies when diffusion models memorize their training data versus when they properly generalize. The authors study a setting in which the data distribution is a $K$-component Gaussian mixture in $\mathbb{R}^d$. Diffusion models are trained to reverse a standard forward noising process, and the learned denoiser $\tilde{x}_\theta(t, x_t)$ is constrained to be the minimum mean squared error (MMSE) estimator implied by a Gaussian mixture model with $M$ components.

The parameter $M$ controls model capacity. For $M = K$, the model can represent the true data distribution and generalizes. For $M = N$ with zero-variance components at training points, the model memorizes — its denoiser effectively returns a nearest neighbor from the training set.

Two types of training losses are defined and analyzed:

1. $L_N(\mu^1, \dots, \mu^K, \sigma)$: the expected training loss when the denoiser is derived from an assumed $K$-component Gaussian mixture model. The authors derive a closed-form expression for this loss using Tweedie’s identity.

2. $L_N(x^1, \dots, x^N, 0)$: the training loss of a fully memorizing denoiser that places a zero-variance Gaussian at each training sample.

They show that these two losses intersect at a critical model size $M^\star$. This crossover point predicts a transition between generalization and memorization: for $M < M^\star$, the generalizing model has lower loss; for $M > M^\star$, the memorizing model does.

To relate this theoretical crossover to practical training outcomes, they introduce a time-dependent weighting function $\lambda(t)$ over diffusion times. They fit a single weighting function $\tilde{\lambda}(t)$ across multiple training setups to align the theoretical crossover point with the empirically observed memorization threshold. This calibration is used once and held fixed in subsequent evaluations.

The experiments are conducted on two synthetic settings:

1. Isotropic Gaussian Mixture: A dataset drawn from an isotropic $K$-component Gaussian mixture. The authors train denoisers of increasing size $M$ and observe a sharp phase transition in both test loss and a "memorization ratio" (the fraction of generated samples classified as memorized based on proximity to training data). They show that the predicted crossover point matches the empirical onset of memorization.

2. Low-Rank Image-Like Data:  Each image is generated by applying a low-rank color embedding resulting in a mixture of low-rank Gaussians. They repeat the same experimental setup and observe a similar memorization transition.

Together, these experiments demonstrate that the theoretical loss crossover — once calibrated using $\tilde{\lambda}(t)$ — predicts the generalization–memorization phase transition observed in trained diffusion models.

**Questions:**

Can you explain why optimizing the weights $\lambda_t$ actually isn't "cheating"? I'm not claiming that it is, but it does feel fishy when one of the main claims of this paper is that the theory "predicts" a transition point and yet this prediction is made with some sort of explicit training.

Edit: re-evaluated post rebuttal.

**Ethical Concerns:**

["NO or VERY MINOR ethics concerns only"]

**Final Justification:**

This is all clearly laid out in my comment to the authors.

**Limitations:**

Yes.

**Paper Formatting Concerns:**

None.

**Quality:**

3

**Strengths And Weaknesses:**

Strengths: this paper provides a simple model where memorization can be easily characterized. It also gives explicit derivations of the expected losses (rather than bounds) which allows the loss difference comparison (i.e. memorization loss vs loss from using the true data distribution) to be accurate computed.

Weaknesses: this paper claims a "phase transition" without really giving a good definition of what that is. It is sort of obvious that if you increase the memorization capacity of your model that your model will memorize more and generalize less (this is observed ubiquitously in machine learning) so merely providing plots and claiming that it is happening doesn't seem very compelling. Furthermore, claiming that the theory above can "predict" this transition point is a bit less impressive than it sounds because 1. the predictions are in a very restricted theoretical setting and 2. the weights $\tilde{\lambda}_t$ appears to be optimized to make these predictions as accurate as possible.

I also found this paper a bit difficult to read. The notation used throughout the first few sections feels rather messy (there is an excessive use of bolding, subscripts, \tilde, \overline, \hat, etc.) and it took me a while to understand what the Loss $L_n(x^{(1)},\dots x^{(n)}, 0)$ meant. I strongly recommend thoughtfully cleaning up this section and adding some definitions in (like a definition environment) that might help more cleanly explain what is going on here.


Edit: see comment for update on review.

---

> ### Author Rebuttal · Authors · 2025-07-31
>
> We thank the reviewer for their thorough review of our submission. We respond point-by-point below.
>
> Given the clarifications below around the soundness of our methodology, we hope the reviewer will consider raising their score. We look forward to discussing any remaining issues further.
>
> > this paper claims a "phase transition" without really giving a good definition
> > of what that is. It is sort of obvious that if you increase the memorization
> > capacity of your model that your model will memorize more and generalize less
> > (this is observed ubiquitously in machine learning) so merely providing plots
> > and claiming that it is happening doesn't seem very compelling.
>
> We respectfully yet strongly push back on the notion that we are “merely providing plots and claiming that [the phase transition] is happening”. The existence of this phase transition and even its universality across different models and hyperparameters has been very well established by prior empirical work [1–2]. A main contribution in our work is to isolate the factors underlying this phase transition and reproduce it in a synthetic setting amenable to theoretical analysis, and to empirically link the precise point at which it occurs *in trained models* to easily-computable properties of the loss landscape. This sheds light into the mechanisms underlying the phase transition in a novel way with regards to previous empirical studies, and can act as a foundation for further finer-grained studies in the future.
>
> In general, we can clarify the writing to more clearly introduce the concept of a phase transition. This term is prevalent in physics/statistical
> mechanics and their connection to machine learning [5–6], and phase transitions between
> memorization and generalization have been empirically observed in prior
> works [1--2], which we have discussed in the introduction and related work sections of
> the submission. Quoting from [1], a foundational work in this area:
> > We show empirically that diffusion models memorize samples when trained on small sets, but transition to a strong form of generalization as the training set size increases… [t]he amount of data needed to reach this phase transition is very small… and depends on the image size and complexity relative to the neural network capacity.
> >
> > --- [1], p. 9
>
> We respectfully note that the reviewer's objection (“it is sort of obvious…”) could be identically levied
> at any of these foundational prior works.
>
> The fact that this phase transition has caused
> surprise and research interest in the context of diffusion models (in contrast
> to its "[ubiquity] in machine learning") may be attributable to the fact that
> certain diffusion models seem to memorize more than equivalent generative models [3],
> and that the relationship between the trained model (a denoiser for
> different times $t$) and the algorithm for generating samples (a multi-step
> iterative refinement of a noisy image with the trained model) obfuscates the
> link between "overfitting" behavior of the denoisers on a given timestep's loss
> and the ultimate memorization behavior of generated samples.
> On a more abstract level, note that in the context of diffusion
> model training (i.e., Gaussian denoising), *the unique optimizer of the training loss
> is the model that memorizes the training data*, as we describe in
> Section 2 of the submission. This should be
> contrasted with the situation in other machine learning settings, e.g.
> classification, where there are typically an infinite number of models that
> achieve minimum training loss, and the key question is to understand which model
> is learned, given one's specific training algorithm [4].
>
> > Can you explain why optimizing the weights \lambda_t actually isn't
> > "cheating"? I'm not claiming that it is, but it does feel fishy when one of
> > the main claims of this paper is that the theory "predicts" a transition point
> > and yet this prediction is made with some sort of explicit training.
>
> In short: Optimizing the weights $\tilde{\lambda}$ is not “cheating” because it is a standard and accepted scientific methodology. It corresponds to hypothesizing that an empirical phenomenon follows a certain mathematical law, then fitting the free parameters of the law based on empirical data, and verifying whether the resulting law correctly predicts the phenomenon on held-out data. This is the same methodology that underlies the modern practice of scaling laws for training large models [7].
>
> Below, we provide a discussion with full details in order to completely clarify the role of $\tilde{\lambda}$ in our methodology.
>
> In our submission, we hypothesize that the
> memorization-generalization transition occurs when the weighted relative loss
> of two surrogate models (a generalizing model and a memorizing model) switches
> from positive to negative (the "crossover"), for *some* weighting. In this way, our experiments effectively fit a predictive model $f(N, d, K;
> \tilde{\lambda})$ (defined in eqn. 19) with parameters $\tilde{\lambda}$ for the
> number of model parameters $M_{\star}(N, d, K)$ at which memorization becomes prevalent
> (lines 242-251), **and validates the accuracy of this predictive model on a set of
> held-out simulation data** (lines 284–288).
>
> We can also demonstrate that the effective predictive model implied by this parameterization is a simple linear function of the number of samples $N$, so the reviewer should not be worried about any hidden capacity to overfit.
> The two surrogate models’ loss approximations that we use in our predictive model are justified by Theorem 3.1, namely
> $$ \mathcal{L}^{\mathrm{gen}}(t, M) = \mathcal{L}^{\mathrm{mem}}(t) +
>   \frac{d\sigma_\star^2}{\psi_t \sigma_\star^2 + 1},$$
> where $\psi_t = \alpha_t / \sigma_t$ and $\sigma_\star^2$ is the variance of the
> ground-truth model (we use $\sigma_\star^2 = 1$ in our experiments); and Theorem
> 3.2 (and Figure 1), namely
> $$ \mathcal{L}^{\mathrm{pmem}}(t, M) = \mathcal{L}^{\mathrm{mem}}(t) + 2 d\sigma_\star^2 \left( 1 - \frac{M}{N}\right). $$
> For integrable weights $\tilde{\lambda} : [0, 1] \to \mathbb{R}$, the weighted
> crossover point formally
> entails the study of the value of $M$ at which
> $$ \int_{[0, 1]} \tilde{\lambda}(t) \mathcal{L}^{\mathrm{gen}}(t, M) \mathrm{d} t
> -\int_{[0, 1]} \tilde{\lambda}(t) \mathcal{L}^{\mathrm{pmem}}(t, M) \mathrm{d} t = 0. $$
> Because $\mathcal{L}^{\mathrm{pmem}}(t, M)$ is a constant function of $t$,
> when we plug in the concrete expressions for $\mathcal{L}^{\mathrm{gen}}$ and
> $\mathcal{L}^{\mathrm{pmem}}$ above to this expression, we end up with the
> equivalent expression
> $$ \int_{[0, 1]} \frac{\tilde{\lambda}(t)}{\int_{[0, 1]}\tilde{\lambda}(t)\mathrm{d} t } \left( \frac{\tfrac{1}{2} + \psi_t \sigma_\star^2}{1 + \psi_t \sigma_\star^2 } \right) \mathrm{d} t = \frac{M}{N}. $$
> This shows that *fitting weights $\tilde{\lambda}$ to
> minimize the predictive accuracy on a dataset is equivalent to fitting a linear
> model $M_{\star} = w N$*, in the language of the discussion above.
>
> In spite of the simplicity of the model for the location of the phase
> transition, note that our experiments fail to reject this hypothesis.
>
> > Furthermore, claiming that the theory above can "predict" this transition
> > point is a bit less impressive than it sounds because 1. the predictions are
> > in a very restricted theoretical setting…
>
> We emphasize that our setting is no more restrictive than what every theoretical work on diffusion models considers. In the theoretical literature on diffusion models, the Gaussian mixture model (GMM) we study actually sits at the ‘frontier’: almost any algorithmic research question one can formulate around diffusion models remains open for GMMs. More precisely, [8] proves that the minimizers of the training loss over a properly-parameterized class of GMMs generalize (in our notation, analogous to the number of components $M$ being equal to the number of classes $K$), [11] proves the same for models learned by gradient descent for two-point mixtures of Gaussians, and [9–10] establish guarantees for learning properly-parameterized GMMs with non-gradient-descent algorithms. To the best of our knowledge, no prior work has studied the setting where the number of components $M$ may be significantly larger than the number of classes $K$, which is the essential setting to understand memorization in GMM diffusion models as they require $M \gg K$ parameters to memorize with low loss. We provide further references in our response to `DzmC`.
>
> > I strongly recommend thoughtfully cleaning up this section and adding some
> > definitions in (like a definition environment) that might help more cleanly
> > explain what is going on here.
>
> We apologize for the issues of presentation the reviewer mentioned. In the revision, making use of the expanded page limit, we will streamline the presentation in Sections 2–4.
>
> Please let us know if you have other questions or hang-ups, and we will be happy to resolve them. We look forward to a productive discussion during the rebuttal period.
>
> # References
> [1] Kadkhodaie et al., Generalization in diffusion models arises from geometry-adaptive harmonic representations.
>
> [2] Zhang et al., The Emergence of Reproducibility and Consistency in Diffusion Models.
>
> [3] Carlini et al., Extracting Training Data from Diffusion Models.
>
> [4] Zhang et al., Understanding deep learning requires rethinking generalization.
>
> [5] Amelunxen et al., Living on the edge: phase transitions in convex programs with random data.
>
> [6] Maillard et al., Phase retrieval in high dimensions: Statistical and computational phase transitions.
>
> [7] Kaplan et al., Scaling Laws for Neural Language Models.
>
> [8] Wang et al., Diffusion Models Learn Low-Dimensional Distributions via Subspace Clustering.
>
> [9] Gatmiry et al., Learning Mixtures of Gaussians Using Diffusion Models.
>
> [10] Chen et al., Learning general Gaussian mixtures with efficient score matching.
>
> [11] Shah et al., Learning Mixtures of Gaussians Using the DDPM Objective.

---

> > ### Comment · Reviewer_vocx · 2025-08-08
> > **Response to rebuttal**
> >
> > Thanks for your detailed answers. I'd like to also apologize for the somewhat antagonistic phrasing of my review -- I do feel it felt unnecessarily harsh in tone.
> >
> > Regarding my concerns
> >
> > 1. "Triviality of memorization vs genearlization" -- I accept the authors resposne here. While this intuition is well-known, I take the point that diffusion models can have pretty complicated behavior and thus establishing "intuitive" behavior is a significant contribution.
> >
> > 2. "Cheating accusations on \lambda_s" -- I fully retract my objection here. It was based on a poor understanding of the paper.
> >
> > 3. "Significance of theoretical setting" -- it seems that there is wide agreement that this setting is pretty limited but also fairly close to the research frontier. This is hard for me to judge, so I am now agnostic on this issue. If other reviewers agree that this setting is still novel enough to be of significance, then I have no objection.
> >
> > 4. "presentation" -- I'm happy enough with the authors commitment to work on the presentation.
> >
> > Based on this, I will increase my score to an accept as I really have no reason to argue for rejection. however based on the shift from my original review, I will keep a lower confidence.

---

> ### Author Response · Authors · 2025-08-05
> **Discussion Period Ending**
>
> As we approach the end of the discussion period, we wanted to check back in with you and inquire as to whether you have any remaining questions, comments, or concerns. If so, we are happy to discuss them; if not, we would humbly request a re-evaluation of your score.

---

### Decision · Program_Chairs · 2025-09-17

**Decision:**

Accept (poster)

**Comment:**

The paper presents a theoretical framework for understanding memorization in diffusion models, focusing on the point at which these models transition from generalization to memorization. The authors introduce the notion of a “crossover point,” where the training loss landscape begins to favor memorizing training samples rather than generalizing across them. The framework thus provides a theoretical grounding for the phase transition in model capacity that is responsible for memorization. The authors further validate this crossover point empirically, showing that it reliably predicts the onset of memorization in practice.

**Strengths**: The paper provides a well-grounded theoretical analysis of memorization in unconditional diffusion models and provides interesting insights on the memorization-generalization behavior in diffusion models which were previously unstudied theoretically.

**Weaknesses**: The main concerns that remain after the rebuttal are about the transferability of the findings in the simplified setup to state-of-the-art diffusion models, which might limit the actual impact of the work. While it would have been nice to extend the evaluation in the paper towards more real-world datasets, the argument from the rebuttal that these experiments are already conducted in prior (empirical) work seems convincing.

During the rebuttal, concerns about the choice of $\lambda_s$ were resolved and the authors committed to improve the presentation to make it more clear. Additionally, concerns about novelty of the finding were adequately addressed in the rebuttal by arguing that while the phenomenon is "expected", theoretical assessments were missing. Thereby, the paper closes a gap and makes a valuable contribution.